# Convergence Analysis of Split Federated Learning on Heterogeneous Data

**Pengchao Han** [*]
Guangdong University of Technology, China
hanpengchao@gdut.edu.cn

**Chao Huang** [*]
Montclair State University, USA
huangch@montclair.edu

**Geng Tian**
Southern University of Science and Technology, China
12332463@mail.sustech.edu.cn

**Ming Tang** [†]
Southern University of Science and Technology, China
tangm3@sustech.edu.cn

**Xin Liu**
University of California, Davis, USA
xinliu@ucdavis.edu

## Abstract

Split federated learning (SFL) is a recent distributed approach for collaborative model training among multiple clients. In SFL, a global model is typically split into two parts, where clients train one part in a parallel federated manner, and a main server trains the other. Despite the recent research on SFL algorithm development, the convergence analysis of SFL is missing in the literature, and this paper aims to fill this gap. The analysis of SFL can be more challenging than that of federated learning (FL), due to the potential dual-paced updates at the clients and the main server. We provide convergence analysis of SFL for strongly convex and general convex objectives on heterogeneous data. The convergence rates are $O(1/T)$ and $O(1/\sqrt[3]{T})$, respectively, where $T$ denotes the total number of rounds for SFL training. We further extend the analysis to non-convex objectives and the scenario where some clients may be unavailable during training. Experimental experiments validate our theoretical results and show that SFL outperforms FL and split learning (SL) when data is highly heterogeneous across a large number of clients.

## 1 Introduction

### 1.1 Motivation

Federated learning (FL) [18, 9] allows distributed clients to train a global machine learning model collaboratively without sharing raw data. FL leverages the parallel computing capabilities of clients to enhance model training efficiency. However, FL is usually computationally intensive. Clients need to train the entire global model multiple times, which can be infeasible for resource-constrained edge devices. This challenge is further exacerbated as the trend towards increasingly larger model architectures demands more substantial resources [1]. Moreover, FL suffers from the client drift

---

[*]Equal contribution.

[†]Corresponding author.

This work was partially supported by the National Natural Science Foundation of China (Grants 62202214 and 62401161), Guangdong Basic and Applied Basic Research Foundation (Grants 2023A1515012819 and 2022A1515110056), and USDA-020-67021-32855.

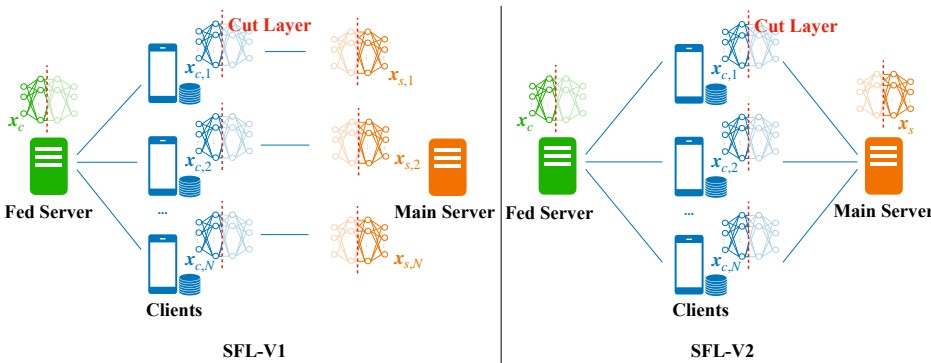

Figure 1: An illustration of SFL framework, and there are two major algorithms, i.e., SFL-V1 (left) and SFL-V2 (right) [27]. More discussions on SFL-V1 and SFL-V2 are given in Sec. 2.

problem when clients' data distributions are heterogeneous, aka non-identically and independently distributed (non-IID). A large number of studies have proposed algorithms to address the client drift issue, e.g., [15, 10, 14, 25].

Split learning (SL) [28] is another distributed approach. By splitting the model across clients and a main server, SL can substantially reduce the computational workload on edge devices. Moreover, recent studies in [34, 17] show that SL can outperform FL when data is highly heterogeneous. However, SL's sequential training among clients can lead to high latency in each training round and potential performance loss (e.g., caused by catastrophic forgetting), which impedes its practical applicability in real-world distributed systems.

In light of above challenges, Thapa et. al in [27] proposed split federated learning (SFL) as a hybrid approach that synergizes the strengths of both FL and SL. SFL combines parallel training of FL with partial model training of SL. They proposed two major SFL algorithms: SFL-V1 and SFL-V2. An illustration of these SFL algorithms are shown in Fig. 1. Specifically, the global model (to be trained) is first split at a cut layer into two parts: a client-side model and a server-side model. Then, the clients are responsible for training only the client-side model under the coordination of a *fed server* (similar to FL). Another server, known as the *main server*, is tasked with training the server-side model by collaborating with the clients (similar to SL). SFL aims to leverage parallel processing to reduce latency, while benefiting from the reduced computational workloads and enhanced data heterogeneity handling of SL.

Following [27], there has been an emerging volume of empirical studies on SFL. e.g., [22, 21, 3, 23, 8, 31, 5]. However, **a convergence analysis of SFL is missing in the literature**, and this paper aims to provide a comprehensive convergence analysis under different conditions. Convergence theory is crucial for understanding the learning performance of SFL, particularly in the context of *heterogeneous data* and *partial participation* scenarios. In practical distributed systems, clients are prone to have different data distributions. Moreover, not all clients may be active or available at all times. These two issues can significantly affect the learning performance of SFL. We aim to provide convergence guarantees for SFL on heterogeneous data (under both full and partial participation). We further compare the results to FL and SL, which provides insights into the practical deployment of various distributed approaches.

## 1.2 Related Work

**Convergence theories of FL and SL**. There are many convergence results on FL. Most studies focus on data heterogeneity, e.g., [30, 16, 11, 10, 12]. Some studies look at partial participation, e.g., [35, 29, 26]. There are also convergence results on Mini-Batch SGD, e.g., [24, 33, 32], where [33] argued that the key difference between FL and Mini-Batch SGD is the communication frequency.

To our best knowledge, there is only one recent study [17] discussing the convergence of SL. The major difference to SL analysis lies in the sequential training manner across clients, while SFL clients perform parallel training.

### 1.3 Challenges and Contributions

**Challenges of SFL convergence analysis**. When data is homogeneous (IID) across clients, the convergence theory in [12] (mainly developed for FL) can be applied to SFL. When data is heterogeneous, however, the theory cannot be directly applied due to the client drift problem. The challenge is intensified with clients' partial participation, which induces bias in the training process. Despite that prior FL theories have handled data heterogeneity [16] and partial participation [29], SFL convergence analysis imposes unique challenges due to the dual-paced model aggregation and model updates at the client-side and server-side. More specifically,

*Dual-paced model aggregation in SFL-V1*: In SFL-V1, the main server maintains one server-side model for each client, and it periodically aggregates the server-side models. When the main server aggregates its models at the same frequency as the clients, the analysis is the same to that of FL. However, FL analysis cannot be applied when aggregations occur at different frequencies, and it is challenging to analyze the impact of such discrepancy on SFL convergence.

*Dual-paced model updates in SFL-V2*: In SFL-V2, the main server only maintains one version of server-side model. The clients update the client-side models in a parallel manner while the main server updates the server-side model in a sequential fashion. Hence, each client's local update depends on the randomness of the previous clients who have interacted with the main server. While [17] handled sequential client training, their theory cannot be applied to SFL-V2 as they did not consider the aggregation of client-side models. This makes our analysis more challenging than FL and SL.

**Contributions**. We summarize our contributions as follows:

- We provide the first comprehensive convergence analysis of SFL. The analysis is more challenging than prior FL analysis due to the dual-paced model aggregation and model updates. To this end, we derive a key decomposition result (Proposition 3.5) that enables us to analyze the convergence from the server-side and client-side separately.

- Based on the decomposition result, we prove that the convergence guarantees of both SFL-V1 and SFL-V2 are $O(1/T)$ for strongly convex objective and $O(1/\sqrt[3]{T})$ for general convex objective, where $T$ denotes the total number of rounds for SFL training. We further extend the analysis to non-convex objectives and more practical scenarios where some clients may be unavailable during training.

- We conduct simulations on various datasets. We show that the results are consistent with our theories. We further show two surprising results: (i) SFL achieves a better performance when clients maintain a larger portion of the global model; (ii) SFL-V2 outperforms FL and SL when clients have highly heterogeneous data and the number of client is large.

The rest of the paper is organized as follows. Sec. 2 formulates the SFL model. Sec. 3 presents the convergence results for SFL. We conduct experiments in Sec. 4 and conclude in Sec. 5.

## 2 Problem Formulation

### 2.1 Model

We consider a set of clients $\mathcal{N} = \{1, 2, \cdots, N\}$, where each client $n \in \mathcal{N}$ has a local private dataset $\mathcal{D}_n$ of size $D_n = |\mathcal{D}_n|$. Suppose the global model parameterized by $\boldsymbol{x}$ has $L$ layers. In SFL, the global model is split at the $L_c$-th layer (i.e., the cut layer) into two segments: a client-side model $\boldsymbol{x}_c$ (from the first layer to layer $L_c$) and a server-side model $\boldsymbol{x}_s$ (from layer $L_c + 1$ to layer $L$), where $\boldsymbol{x} = [\boldsymbol{x}_c; \boldsymbol{x}_s]$. Let $\boldsymbol{x}_{c,n}$ denote the local client-side model of client $n$. The clients train models with the help of two servers: (i) fed server, which periodically aggregates clients' local models $\boldsymbol{x}_{c,n}$ (similar to FL), and (ii) main server, who trains the server-side model $\boldsymbol{x}_s$. In this work, we consider two major SFL algorithms: SFL-V1 and SFL-V2 [27]. In SFL-V1, the main server maintains a separate server-side model $\boldsymbol{x}_{s,n}$ corresponding to each client $n$. In comparison, in SFL-V2, the main server only maintains one model $\boldsymbol{x}_s$.

Let $F_n(\boldsymbol{x}; \zeta_n)$ denote the loss of model $\boldsymbol{x}$ over client $n$'s mini-batch instance $\zeta_n$, which is randomly sampled from client $n$'s dataset $\mathcal{D}_n$. Let $F_n(\boldsymbol{x}) \triangleq \mathbb{E}_{\zeta_n \sim \mathcal{D}_n}[F_n(\boldsymbol{x}; \zeta_n)]$ denote the expected loss of model $\boldsymbol{x}$ over client $n$'s dataset. The goal of SFL is to minimize the expected loss of the model $\boldsymbol{x}$

over the datasets of all clients:

$$\min_{\boldsymbol{x}} f(\boldsymbol{x}) = \sum_{n=1}^{N} a_n F_n(\boldsymbol{x}), \tag{1}$$

where $a_n \in [0, 1]$ is the weight of client $n$ satisfying $\sum_{n \in \mathcal{N}} a_n = 1$. Typically, $a_n = D_n / \sum_{n' \in \mathcal{N}} D_{n'}$, where a client with a larger data size is assigned a larger weight [34].

## 2.2 Algorithm Description

We provide a brief description of SFL. Refer to Appendix B for a more detailed discussion. SFL takes a total number of $T$ rounds to solve (1). At the beginning of each round $t$, clients download the recent global client-side model from the fed server, where the model is an aggregated version of the client-side models of the clients from the previous round $t - 1$. Each round $t$ contains two stages:

**Stage 1: model training**. Clients and the main server train the full global model for $\tau$ iterations in each round. In each iteration $i < \tau$, there are three steps:

*Step 1: client forward propagation*. Each client $n$ samples a mini-batch of data $\zeta_n^{t,i}$ from $\mathcal{D}_n$, computes the intermediate features (e.g., activation values at the cut layer) over its current model $\boldsymbol{x}_{c,n}^{t,i}$, and sends the activation to the main server. The clients perform forward propagation in parallel.

*Step 2: main server training*. Upon receiving the activation of each client $n$,

- SFL-V1: the main server computes the loss using the current server-side model $\boldsymbol{x}_{s,n}^{t,i}$. It then computes the gradients over $\boldsymbol{x}_{s,n}^{t,i}$ to update the model. It also computes the gradient over the activation at the cut layer, and sends it to client $n$.
- SFL-V2: the main server computes the loss $F_n(\{\boldsymbol{x}_{c,n}^{t,i}, \boldsymbol{x}_s^{t,i}\})$, based on which it then updates the server-side model $\boldsymbol{x}_s^{t,i}$. It also computes and sends the gradient over activation at the cut layer to client $n$. Note that the main server sequentially interacts with the clients in a randomized order.

*Step 3: client backward propagation*. Receiving gradient at the cut layer, each client $n$ computes the client-side gradient using the chain rule, and then updates its model $\boldsymbol{x}_{c,n}^{t,i}$.

**Stage 2: model aggregation**. Model aggregation can occur for both client-side and server-side models. For the client side, after $\tau$ iterations of model training (i.e., at the end of round $t$), each client sends its current client-side model to the fed server. The fed server aggregates the clients' models (e.g., weighted averaging), which will be downloaded in the next round $t + 1$:

$$\boldsymbol{x}_c^{t+1} \leftarrow \sum_{n \in \mathcal{N}} a_n \boldsymbol{x}_{c,n}^{t,\tau}. \tag{2}$$

For the server side, (i) in SFL-V1, after $\tilde{\tau}$ iterations of training, the main server aggregates all server-side models. Note that $\tilde{\tau}$ does not necessarily need to equal $\tau$, but when equality holds, SFL-V1 can be regarded as FL (despite the model splitting). (ii) In SFL-V2, no aggregation occurs since the main server only maintains one model.

## 2.3 Client Participation

We consider two cases: (i) *full participation* where all clients are available during training. This can model the scenarios where clients are organizations or companies who likely have sufficient computation and communication resources [7]; (ii) *partial participation* where some clients may be unavailable during training. This can model the cases where clients are edge devices (e.g., mobile phones) that are usually resource-constrained and may be disconnected from the SFL process.

To model partial participation, we consider independent participation probabilities for each client, allowing for arbitrary and heterogeneous participation probabilities. Specifically, we use $q_n \in [0, 1]$ to denote client $n$'s participation level (or probability), and $\boldsymbol{q} = (q_n, n \in \mathcal{N})$. If $q_n = 1$, client $n$ participates in every round of SFL with probability one. If $q_n < 1$, client $n$ is unavailable in some rounds. Denote $\mathcal{P}^t(\boldsymbol{q})$ as the set of participating clients in round $t$. In the presence of partial

participation, we need to modify (2) (and the potential server-side aggregation) to offset the incurred bias:

$$\boldsymbol{x}_c^{t+1} \leftarrow \sum_{n \in \mathcal{P}^t(\boldsymbol{q})} \frac{a_n}{q_n} \boldsymbol{x}_{c,n}^{t,\tau}. \tag{3}$$

## 3  Convergence Analysis

We first make technical assumptions in Sec. 3.1. Then, we present a key technical result in Sec. 3.2 to support the SFL convergence analysis. Finally, we provide the convergence results under full participation and partial participation in Sec. 3.3 and Sec. 3.4, respectively.

### 3.1  Assumptions

We start with some conventional assumptions for convergence analysis in the FL literature.

**Assumption 3.1.** ($S$-*Smoothness*) Each client $n$'s loss function $F_n$ is $S$-smooth. That is, for all $\boldsymbol{x}, \boldsymbol{y} \in \mathbb{R}^d$,

$$F_n(\boldsymbol{y}) \leq F_n(\boldsymbol{x}) + \langle \nabla F_n(\boldsymbol{x}), \boldsymbol{y} - \boldsymbol{x} \rangle + \frac{S}{2}\|\boldsymbol{y} - \boldsymbol{x}\|^2. \tag{4}$$

The smoothness assumption holds for many loss functions in, for example, logistic regression, softmax classifier, and $l_2$-norm regularized linear regression [16].

**Assumption 3.2.** (*Unbiased and bounded stochastic gradients with bounded variance*) The stochastic gradients $\boldsymbol{g}_n(\cdot)$ of $F_n(\cdot)$ is unbiased with the variance bounded by $\sigma_n^2$.

$$\mathbb{E}_{\zeta_n \sim \mathcal{D}_n}\left[\boldsymbol{g}_n\left(\boldsymbol{x}, \zeta_n\right)\right] = \nabla F_n\left(\boldsymbol{x}\right), \tag{5}$$

$$\mathbb{E}_{\zeta_n \sim \mathcal{D}_n}\left[\left\|\boldsymbol{g}_n\left(\boldsymbol{x}, \zeta_n\right) - \nabla F_n\left(\boldsymbol{x}\right)\right\|^2\right] \leq \sigma_n^2. \tag{6}$$

**Assumption 3.3.** (*Bounded gradients*) The expected squared norm of stochastic gradients is bounded by $G^2$.

$$\mathbb{E}_{\zeta_n \sim \mathcal{D}_n}\left\|\boldsymbol{g}_n\left(\boldsymbol{x}, \zeta_n\right)\right\|^2 \leq G^2. \tag{7}$$

The value of $\sigma_n$ measures the level of stochasticity.

**Assumption 3.4.** (*Heterogeneity*) There exists an $\epsilon^2$ such that the divergence between local and global gradients is bounded by $\epsilon^2$.

$$\left\|\nabla F_n\left(\boldsymbol{x}\right) - \nabla f\left(\boldsymbol{x}\right)\right\|^2 \leq \epsilon^2. \tag{8}$$

A larger $\epsilon^2$ indicates a larger degree of data heterogeneity.

### 3.2  Decomposition

As discussed in Sec. 1.3, analyzing the performance bound of SFL can be more challenging than that of conventional FL counterparts due to the dual-paced model aggregation and model updates. To address this challenge, we decompose the convergence analysis into the server-side and client-side updates, respectively. We give the decomposition below.

**Proposition 3.5.** (*Convergence decomposition*) *Let* $\boldsymbol{x}^* \triangleq [\boldsymbol{x}_c^*; \boldsymbol{x}_s^*]$ *denote the optimal global model that minimizes* $f(\cdot)$, *and* $\boldsymbol{x}^T \triangleq [\boldsymbol{x}_c^T; \boldsymbol{x}_s^T]$ *is the global model obtained after* $T$ *rounds of SFL training. Under Assumption 3.1, we have*

$$\mathbb{E}\left[f(\boldsymbol{x}^T)\right] - f(\boldsymbol{x}^*) \leq \frac{S}{2}\left(\mathbb{E}\|\boldsymbol{x}_s^T - \boldsymbol{x}_s^*\|^2 + \mathbb{E}\|\boldsymbol{x}_c^T - \boldsymbol{x}_c^*\|^2\right). \tag{9}$$

The proof is given in Appendix C.4. Proposition 3.5 is particularly useful. It shows that despite the challenging dual-paced updates, to bound the SFL performance gap, it suffices to separately bound the gap at the server-side and client-side models. Note that our decomposition can be easily applied to other distributed approaches such as SL. In addition, such a decomposition is not necessarily loose, as our derived bounds for SFL achieve the same order as in FL (see Appendix H.2 for details).

## 3.3 Results under Full Participation

Built upon Proposition 3.5, we first present the convergence results under full participation. For convenience, define

$$I^{\text{err}} \triangleq \left\| \boldsymbol{x}^0 - \boldsymbol{x}^* \right\|^2, \quad \gamma \triangleq 8S/\mu - 1, \quad \tau_{\min} \triangleq \min\{\tau, \tilde{\tau}\}, \quad \tau_{\max} \triangleq \max\{\tau, \tilde{\tau}\}, \quad (10)$$

and let $\eta^t$ represent the learning rate at round $t$. Let $f^*$ denotes the optimal global loss, i.e., $f^* \triangleq f(\boldsymbol{x}^*)$. All results are obtained based on Assumptions 3.1-3.4. The convergence results for SFL-V1 and SFL-V2 are summarized in Theorems 3.6 and 3.7, respectively[1].

**Theorem 3.6.** ( *SFL-V1: full participation)*

$\mu$-*strongly convex:* Let Assumptions 3.1 - 3.3 hold, and $\eta^t = \frac{4}{\mu\tilde{\tau}(\gamma+t)}$ for client-side model and $\eta^t = \frac{4}{\mu\tau(\gamma+t)}$ for server-side model,

$$\mathbb{E}\left[f(\boldsymbol{x}^T)\right] - f^* \leq \frac{8SN\sum_{n=1}^N a_n^2 \left(2\sigma_n^2 + G^2\right)}{\mu^2 \left(\gamma + T\right)} + \frac{768S^2 \sum_{n=1}^N a_n \left(2\sigma_n^2 + G^2\right)}{\mu^3 \left(\gamma + T\right)\left(\gamma + 1\right)} + \frac{S(\gamma + 1)I^{\text{err}}}{2(\gamma + T)}. \quad (11)$$

*General convex:* Let Assumptions 3.1 - 3.3 hold, and $\eta^t \leq \frac{1}{2S\tau_{\max}}$,

$$\begin{aligned}
\mathbb{E}\left[f\left(\boldsymbol{x}^T\right)\right] - f^* \leq &\frac{SI^{\text{err}}}{2(T+1)} + \frac{1}{2}\left(\frac{(\tilde{\tau}^2 + \tau^2)I^{\text{err}}N}{\tau_{\min}^2(T+1)} \sum_{n=1}^N a_n^2 \left(2\sigma_n^2 + G^2\right)\right)^{\frac{1}{2}} \\
&+ \frac{1}{2}\left(\frac{24(\tilde{\tau}^2 + \tau^2)SI^{\text{err}}}{\tau_{\min}^2(T+1)} \sum_{n=1}^N a_n \left(2\sigma_n^2 + G^2\right)\right)^{\frac{1}{3}}.
\end{aligned} \quad (12)$$

*Non-convex:* Let Assumptions 3.1, 3.2, and 3.4 hold, and $\eta^t \leq \min\left\{\frac{1}{16S\tau_{\max}}, \frac{\tau_{\min}}{8SN\tau_{\max}^2 \sum_{n=1}^N a_n^2}\right\}$,

$$\frac{1}{T}\sum_{t=0}^{T-1} \eta^t \mathbb{E}\left[\left\|\nabla_{\boldsymbol{x}} f\left(\boldsymbol{x}^t\right)\right\|^2\right] \leq \frac{4}{T\tau_{\min}}\left(f(\boldsymbol{x}^0) - f^*\right) + \frac{8NS(\tau^2 + \tilde{\tau}^2)}{T\tau_{\min}} \sum_{n=1}^N a_n^2 (\sigma_n^2 + \epsilon^2) \sum_{t=0}^{T-1} \left(\eta^t\right)^2. \quad (13)$$

**Theorem 3.7.** ( *SFL-V2: full participation)*

$\mu$-*strongly convex:* Let Assumptions 3.1 - 3.3 hold, and $\eta^t = \frac{4}{\mu\tilde{\tau}(\gamma+t)}$ for client-side model and $\eta^t = \frac{4}{\mu\tau(\gamma+t)}$ for server-side model,

$$\mathbb{E}\left[f(\boldsymbol{x}^T)\right] - f^* \leq \frac{8SN\sum_{n=1}^N (a_n^2 + 1)\left(2\sigma_n^2 + G^2\right)}{\mu^2 \left(\gamma + T\right)} + \frac{768S^2 \sum_{n=1}^N (a_n + 1)\left(2\sigma_n^2 + G^2\right)}{\mu^3 \left(\gamma + T\right)\left(\gamma + 1\right)} + \frac{S(\gamma + 1)I^{\text{err}}}{2(\gamma + T)}. \quad (14)$$

*General convex:* Let Assumptions 3.1 - 3.3 hold, and $\eta^t \leq \frac{1}{2S\tau}$,

$$\mathbb{E}\left[f(\boldsymbol{x}^T)\right] - f^* \leq \frac{SI^{\text{err}}}{2(T+1)} + \frac{1}{2}\left(\frac{NI^{\text{err}}}{T+1}\sum_{n=1}^N (a_n^2 + 1)\left(2\sigma_n^2 + G^2\right)\right)^{\frac{1}{2}} + \frac{1}{2}\left(\frac{24SI^{\text{err}}}{T+1}\sum_{n=1}^N (a_n + 1)\left(2\sigma_n^2 + G^2\right)\right)^{\frac{1}{3}}. \quad (15)$$

*Non-convex:* Let Assumptions 3.1, 3.2, and 3.4 hold, and $\eta^t \leq \min\left\{\frac{1}{16S\tau}, \frac{1}{8SN^2\tau}\right\}$,

$$\frac{1}{T}\sum_{t=0}^{T-1} \eta^t \mathbb{E}\left[\left\|\nabla_{\boldsymbol{x}} f\left(\boldsymbol{x}^t\right)\right\|^2\right] \leq \frac{4}{T\tau}\left(f\left(\boldsymbol{x}^0\right) - f^*\right) + \frac{8NS\tau}{T}\sum_{n=1}^N (a_n^2 + 1)(\sigma_n^2 + \epsilon^2)\sum_{t=0}^{T-1}\left(\eta^t\right)^2. \quad (16)$$

---

[1]Following many existing works in FL (e.g., [10]), we consider $\mathbb{E}\left[f(\boldsymbol{x}^T)\right] - f^*$ and $\frac{1}{T}\sum_{t=0}^{T-1} \eta^t \mathbb{E}[\|\nabla_{\boldsymbol{x}} f(\boldsymbol{x}^t)\|^2]$ as the performance metrics for (strongly) convex and non-convex objectives, respectively.

Proofs of Theorems 3.6-3.7 are given in Appendices D-E, respectively. We summarize the key findings below.

**Convergence rate**. The convergence bounds of both SFL-V1 and SFL-V2 achieve an order of $O(1/T)$ on strongly convex (and non-convex) objectives. For general convex objectives, the convergence rate becomes $O(1/\sqrt[3]{T})$.[2] Note that our bounds match the existing bounds for FL and SL (in terms of the order of $T$) on heterogeneous data for strongly convex objectives. For a more detailed comparison, please refer to Appendix H.2.[3]

**Impact of data heterogeneity**. The convergence bounds increase as the level of data heterogeneity increases. For example, in (13), the bound increases in $\epsilon^2$ (see Assumption 3.4). This means that SFL tends to perform worse when clients' data are more heterogeneous, which is a commonly observed phenomenon in distributed learning, e.g., FL.

**Choice of learning rate.** One should use a smaller learning rate when the number of local iteration $\tau$ increases. This bears a similar spirit to [16]. In addition, our results indicate that a proper choice of constant learning rate suffices for SFL convergence. It would be an interesting direction to investigate whether diminishing learning rates are able to achieve faster convergence.

**Comparison between SFL-V1 and SFL-V2**. The convergence results between the two SFL versions are very similar, except that $a_n^2$ (and $a_n$) in SFL-V1 are replaced by $a_n^2 + 1$ (and $a_n + 1$) in SFL-V2. See (11) and (14) for an inspection. We will show in Sec. 4 that SFL-V1 and SFL-V2 achieve similar accuracy (except under highly heterogeneous data).

### 3.4 Results under Partial Participation

Now, we present the results under partial participation.

**Theorem 3.8.** ( *SFL-V1: partial participation*)

*$\mu$-strongly convex: Let Assumptions 3.1 - 3.3 hold, and $\eta^t = \frac{4}{\mu\tilde{\tau}(\gamma+t)}$ for client-side model and $\eta^t = \frac{4}{\mu\tau(\gamma+t)}$ for server-side model,*

$$\mathbb{E}\left[f(\boldsymbol{x}^T)\right]-f^* \le \frac{8SN\sum_{n=1}^N a_n^2\left(2\sigma_n^2+G^2+\frac{G^2}{q_n}\right)}{\mu^2\left(\gamma+T\right)}+\frac{768S^2\sum_{n=1}^N a_n\left(2\sigma_n^2+G^2\right)}{\mu^3\left(\gamma+T\right)\left(\gamma+1\right)}+\frac{S(\gamma+1)I^{\text{err}}}{2(\gamma+T)}. \quad (17)$$

*General convex: Let Assumptions 3.1 - 3.3 hold, and $\eta^t \le \frac{1}{2S\tau_{\max}}$,*

$$\mathbb{E}\left[f\left(\boldsymbol{x}^T\right)\right] - f^* \le \frac{SI^{\text{err}}}{2(T+1)} + \frac{1}{2}\left(\frac{(\tilde{\tau}^2+\tau^2)I^{\text{err}}N}{\tau_{\min}^2(T+1)}\sum_{n=1}^N a_n^2\left(2\sigma_n^2+G^2+\frac{G^2}{q_n}\right)\right)^{\frac{1}{2}}$$
$$+\frac{1}{2}\left(\frac{24(\tilde{\tau}^2+\tau^2)SI^{\text{err}}}{\tau_{\min}^2(T+1)}\sum_{n=1}^N a_n\left(2\sigma_n^2+G^2\right)\right)^{\frac{1}{3}}. \quad (18)$$

*Non-convex: Let Assumptions 3.1, 3.2, and 3.4 hold, and $\eta^t \le \min\{\frac{1}{16S\tau_{\max}}, \frac{\tau_{\min}}{8SN\tau_{\max}^2\sum_{n=1}^N\frac{a_n^2}{q_n}}\}$,*

$$\frac{1}{T}\sum_{t=0}^{T-1}\eta^t\mathbb{E}\left[\left\|\nabla_{\boldsymbol{x}}f\left(\boldsymbol{x}^t\right)\right\|^2\right] \le \frac{4}{T\tau_{\min}}\left(f\left(\boldsymbol{x}^0\right)-f^*\right)+\frac{8NS(\tau^2+\tilde{\tau}^2)}{T\tau_{\min}}\sum_{n=1}^N\frac{a_n^2}{q_n}\left(\sigma_n^2+\epsilon^2\right)\sum_{t=0}^{T-1}\left(\eta^t\right)^2. \quad (19)$$

**Theorem 3.9.** ( *SFL-V2: partial participation*)

*$\mu$-strongly convex: Let Assumptions 3.1 - 3.3 hold, and $\eta^t = \frac{4}{\mu\tilde{\tau}(\gamma+t)}$ for client-side model and $\eta^t = \frac{4}{\mu\tau(\gamma+t)}$ for server-side model,*

$$\mathbb{E}\left[f\left(\boldsymbol{x}^T\right)\right]-f^* \le \frac{8SN\sum_{n=1}^N(a_n^2+1)\left(2\sigma_n^2+G^2+\frac{G^2}{q_n}\right)}{\mu^2\left(\gamma+T\right)}+\frac{768S^2\sum_{n=1}^N(a_n+1)\left(2\sigma_n^2+G^2\right)}{\mu^3\left(\gamma+T\right)\left(\gamma+1\right)}+\frac{S(\gamma+1)I^{\text{err}}}{2(\gamma+T)}. \quad (20)$$

---

[2] Note that it might be counter-intuitive to observe looser bounds on general convex objectives than on non-convex objectives. This is associated with different performance metrics used in the analysis, e.g., see the left hand side of (12) and (13).

[3] We also compared SFL to FL and SL in terms of communication/computation overheads in Appendix H.3.

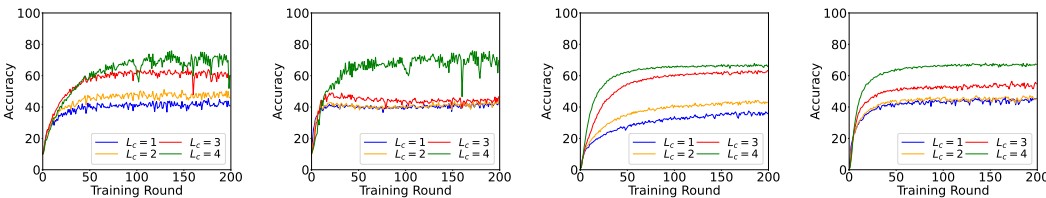

(a) SFL-V1 on CIFAR-10. (b) SFL-V2 on CIFAR-10. (c) SFL-V1 on CIFAR-100. (d) SFL-V2 on CIFAR-100.

Figure 2: Impact of the choice of cut layer on SFL performance.

***General convex:*** *Let Assumptions 3.1 - 3.3 hold, and $\eta^t \leq \frac{1}{2S\tau}$,*

$$
\begin{aligned}
\mathbb{E}\left[f\left(\boldsymbol{x}^T\right)\right] - f^* \leq \; & \frac{SI^{\mathrm{err}}}{2(T+1)} + \frac{1}{2}\left(\frac{NI^{\mathrm{err}}}{T+1}\sum_{n=1}^{N}(a_n^2+1)\left(2\sigma_n^2+G^2+\frac{G_n^2}{q_n}\right)\right)^{\frac{1}{2}} \\
& + \frac{1}{2}\left(\frac{24SI^{\mathrm{err}}}{T+1}\sum_{n=1}^{N}(a_n+1)\left(2\sigma_n^2+G^2\right)\right)^{\frac{1}{3}}.
\end{aligned}
\tag{21}
$$

***Non-convex:*** *Let Assumptions 3.1, 3.2, and 3.4 hold, and $\eta^t \leq \min\left\{\frac{1}{16S\tau}, \frac{1}{8SN^2\tau\sum_{n=1}^{N}\frac{a_n^2}{q_n}}\right\}$,*

$$
\frac{1}{T}\sum_{t=0}^{T-1}\eta^t\mathbb{E}\left[\left\|\nabla_{\boldsymbol{x}}f\left(\boldsymbol{x}^t\right)\right\|^2\right] \leq \frac{4}{T\tau}\left(f\left(\boldsymbol{x}_0\right)-f^*\right) + \frac{8NS\tau}{T}\sum_{n=1}^{N}\frac{a_n^2+1}{q_n}\left(\sigma_n^2+\epsilon^2\right)\sum_{t=0}^{T-1}\left(\eta^t\right)^2. \tag{22}
$$

The proofs are given in Appendices F-G.

**Impact of partial participation**. In practical cross-device settings, some clients may not participate in all rounds of training, i.e., $q_n < 1$ for some $n$. This brings an additional term $G^2/q_n$ to the convergence bound (e.g., see (12) and (18)), meaning that partial participation worsens SFL performance. This is also observed in FL literature (e.g., [29]) and is consistent with our experimental results.

## 4 Experimental Results

### 4.1 Setup

We conduct experiments on CIFAR-10 and CIFAR-100 [13].[4] To simulate data heterogeneity, we adopt the widely used Dirichlet distribution [6] with a controlling parameter $\beta$. Here, a smaller $\beta$ corresponds to a higher level of data heterogeneity across clients. We use ResNet-18, which contains four blocks, as the model structure and consider four types of model splitting represented by $L_c = \{1, 2, 3, 4\}$, where $L_c = n$ means the model is split after the $n$-th residual block. We consider two major distributed approaches as the benchmark, i.e., FL (in particular FedAvg [18]) and SL [28]. The learning rates for SFL-V1, SFL-V2, FL, and SL are set as $0.01$. The batch-size $b_s$ is 128, and we run experiments for $T = 200$ rounds. Unless stated otherwise, we use $N = 10$, $\beta = 0.1$, $E = 5$, where $E$ is the number of local epochs for client-side model aggregation (i.e., every $E$ times of training performed over each client's dataset, their client-side models are aggregated at the fed server), and hence $\tau = \lceil\frac{D_n}{b_s}\rceil \times E$. We set $\tau = \tilde{\tau}$ for the fair comparison to vanilla FL. The experiments are run on a CPU (Intel(R) Xeon(R) Gold 5320 at 2.20GHz) and a GPU (A100-PCIE-80GB). **Our codes are provided in** `https://github.com/TIANGeng708/Convergence-Analysis-of-Split-Federated-Learning-on-Heterogeneous-Data`.

### 4.2 Impact of system parameters on SFL performance

**Impact of cut layer**. We first investigate how the choice of the cut layer $L_c$ affects the SFL performance. The results are reported in Fig. 2. We observe that for both SFL-V1 and SFL-V2,

---

[4]More experiments on FEMNIST are given in Appendix I.5.

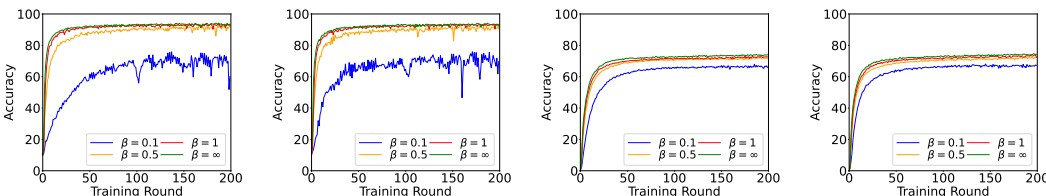

(a) SFL-V1 on CIFAR-10.  (b) SFL-V2 on CIFAR-10.  (c) SFL-V1 on CIFAR-100. (d) SFL-V2 on CIFAR-100.

Figure 3: Impact of data heterogeneity on SFL performance.

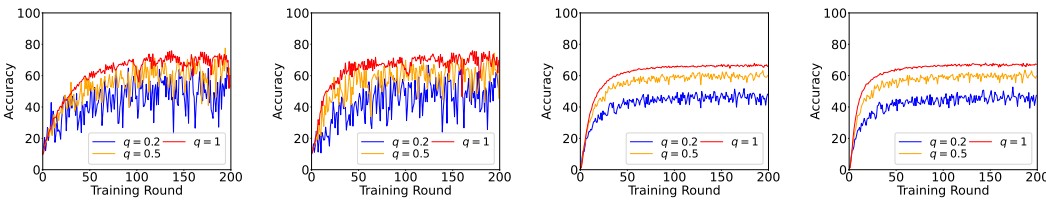

(a) SFL-V1 on CIFAR-10.  (b) SFL-V2 on CIFAR-10.  (c) SFL-V1 on CIFAR-100. (d) SFL-V2 on CIFAR-100.

Figure 4: Impact of client participation on SFL performance.

the performance increases in $L_c$ (i.e., clients have a larger proportion of the global model). This is associated with our empirical observation that the average client gradient variance gets smaller with $L_c$. Intuitively, a smaller gradient variance implies a lower degree of the client drift issue, which leads to a better algorithm performance.[5] Based on this observation, we use $L_c = 4$ for SFL (and SL) for the following experiments.

**Impact of data heterogeneity**. We study the impact of data heterogeneity on SFL performance, where we use $\beta \in \{0.1, 0.5, 1, \infty\}$, and $\beta = \infty$ means clients have IID data. The results are reported in Fig. 3. We observe that a higher level of data heterogeneity (i.e., a smaller $\beta$) leads to slower algorithm convergence and a lower accuracy for both SFL-V1 and SFL-V2. The observation is consistent with our convergence bound, e.g., in (16), the performance bound increases in $\epsilon^2$. Note that the negative impact of heterogeneity is commonly observed in distributed learning literature including FL [7] and SL [21].

**Impact of partial participation**. We study the impact of client participation and let $q_n = q \in \{0.2, 0.5, 1\}, \forall n$. The results are reported in Fig. 4. We observe that a lower level of participation leads to less stable convergence and also a smaller accuracy. This is consistent with our convergence results, e.g., in (20), the bound decreases in clients' participation level $q_n$. Partial participation is expected in practical cross-device scenarios where clients are resourced-constrained edge devices. It is important to develop efficient algorithms as well as effective incentive mechanisms to encourage clients' participation in SFL.

### 4.3 Comparison among SFL, FL, and SL.

We now compare SFL to FL and SL. We consider different combinations of data heterogeneity $\beta \in \{0.1, 0.5\}$ and cohort sizes $N \in \{10, 50, 100\}$. The results are reported in Fig. 5. When data is mildly heterogeneous (i.e., $\beta = 0.5$), SFL and FL have similar convergence rates and accuracy performance. Note that SL seems to under-perform SFL and FL. We think this is mainly due to the catastrophic forgetting issue, which has been observed in [21, 2].

**SFL outperforms FL and SL under highly heterogeneous data and a large client number**. When data becomes more non-IID (i.e., $\beta = 0.1$), SFL-V2 tends to outperform FL and SL. The improvement becomes more significant as the cohort size gets larger. The bottleneck of FL is the client drift issue caused by data heterogeneity. The bottleneck of SL is associated with the catastrophic forgetting. SFL-V2 is a hybrid combination of FL and SL, which can lead to a better tradeoff between client drift and forgetting. By appropriately choosing the cut layer, SFL-V2 outperforms

---

[5]See Appendix I.4 for more detailed discussions on this point.

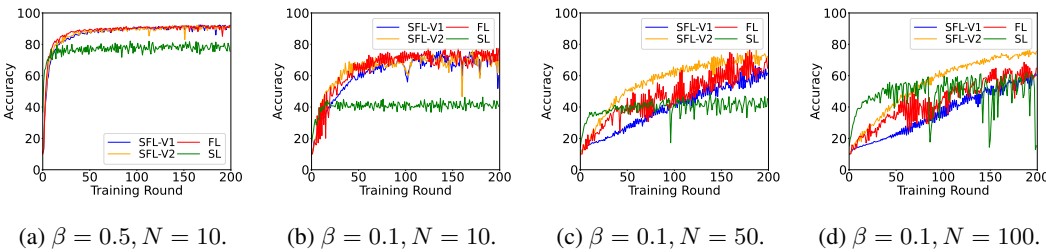

(a) $\beta = 0.5, N = 10$.     (b) $\beta = 0.1, N = 10$.     (c) $\beta = 0.1, N = 50$.     (d) $\beta = 0.1, N = 100$.

Figure 5: Performance comparison on CIFAR-10.

FL and SL. This observation also indicates that SFL-V2 can be a more appealing solution than FL for practical cross-device systems, as it achieves a better performance while requiring smaller computation overheads from edge devices.

## 5 Conclusion

In this work, we provided the first comprehensive convergence analysis of SFL for strongly convex, general-convex, and non-convex objectives on heterogeneous data. One key challenge is the dual-paced model updates. We get around this issue by decomposing the performance gap of the global model into the client-side and server-side gaps. We further extend our analysis to the more practical scenario with partial client participation. Experimental experiments validate our theories and further show that SFL can outperform FL and SL under highly heterogeneous data and a large client number. One limitation of our work is that our bounds for SFL achieve the same order (in terms of training rounds) as in FL, yet the experiments showed that SFL outperforms FL under high heterogeneity. This is possibly due to that tighter bounds for SFL are to be derived, which is an important future work. For future work, one can apply our derived bounds to optimize SFL system performance, considering model accuracy, communication overhead, and computational workload of clients. It is also interesting to theoretically analyze how the choice of the cut layer affects the SFL performance.

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

# A  Appendix / supplemental material

We organize the entire appendix file as follows:

**In Sec. B, we provide detailed algorithmic descriptions.**

**In Sec. C, we provide notations and some technical lemmas.**

- In Sec. C.1, we provide some notations
- In Sec. C.2, we recall SFL-V1 and SFL-V2
- In Sec. C.3, we recall the assumptions
- In Sec. C.4, we provide some useful technical lemmas together with their proofs

**In Sec. D, we prove Theorem 3.6, i.e., convergence of SFL-V1 under full participation.**

- In Sec. D.1, we prove the strongly convex case
- In Sec. D.2, we prove the general convex case
- In Sec. D.3, we prove the non-convex case

**In Sec. E, we prove Theorem 3.7, i.e., convergence of SFL-V2 under full participation.**

- In Sec. E.1, we prove the strongly convex case
- In Sec. E.2, we prove the general convex case
- In Sec. E.3, we prove the non-convex case

**In Sec. F, we prove Theorem 3.8, i.e., convergence of SFL-V1 under partial participation.**

- In Sec. F.1, we prove the strongly convex case
- In Sec. F.2, we prove the general convex case
- In Sec. F.3, we prove the non-convex case

**In Sec. G, we prove Theorem 3.9, i.e., convergence of SFL-V2 under partial participation.**

- In Sec. G.1, we prove the strongly convex case
- In Sec. G.2, we prove the general convex case
- In Sec. G.3, we prove the non-convex case

**In Sec. H, we SFL to other distributed approaches, i.e., FL, SL, and Mini-Batch SGD.**

- In Sec. H.2, we compare their convergence bounds
- In Sec. H.3, we compare their overheads in terms of communication and computation

**In Sec. I, we provide more experimental results.**

# B Algorithm description

For version 1, the client-side model parameter and M-server-side model parameter are aggregated every $\tau$ and $\tilde{\tau}$ iterations, respectively. In iteration $i$ of round $t$, each client $n$ samples a mini-batch of data $\zeta_n^{t,i}$ from $\mathcal{D}_n$, computes the intermediate features $h(\boldsymbol{x}_{c,n}^{t,i}; \zeta_n^{t,i})$ (e.g., activation values at the cut layer) over its current model $\boldsymbol{x}_{c,n}^{t,i}$, and sends $h(\boldsymbol{x}_{c,n}^{t,i}; \zeta_n^{t,i})$ to the M-server. For each client $n$, the M-server computes the loss $F_n(h(\boldsymbol{x}_{c,n}^{t,i}; \zeta_n^{t,i}), \boldsymbol{x}_{s,n}^{t,i})$ based on $\boldsymbol{x}_{s,n}^{t,i}$. Let $\nabla$ denote a gradient operator and $\nabla_{\boldsymbol{w}} F$ represents the gradient of $F$ w.r.t. $\boldsymbol{w}$. The M-server computes the M-server-side gradient $\boldsymbol{g}_{s,n}^{t,i}(\boldsymbol{x}_{s,n}^{t,i}; \zeta_n^{t,i}) = \nabla_{\boldsymbol{x}_s} F_n(h(\boldsymbol{x}_{c,n}^{t,i}; \zeta_n^{t,i}), \boldsymbol{x}_{s,n}^{t,i})$, the gradient over the intermediate features (activations) at the cut layer $\boldsymbol{r}_{c,n}^{t,i}(\boldsymbol{x}_{s,n}^{t,i}; \zeta_n^{t,i}) = \nabla_h F_n(h(\boldsymbol{x}_{c,n}^{t,i}; \zeta_n^{t,i}), \boldsymbol{x}_{s,n}^{t,i})$, and sends $\boldsymbol{r}_{c,n}^{t,i}(\boldsymbol{x}_{s,n}^{t,i}; \zeta_n^{t,i})$ to client $n$. Each client $n$ computes the client-side gradient $\boldsymbol{g}_{c,n}^{t,i}(\boldsymbol{x}_{c,n}^{t,i}; \zeta_n^{t,i})$ based on $\boldsymbol{r}_{c,n}^{t,i}(\boldsymbol{x}_{s,n}^{t,i}; \zeta_n^{t,i})$ using the chain rule.

For version 2, the client-side model is aggregated every $\tau$ iterations, while the M-server trains only one version of the M-server-side model.

---

**Algorithm 1:** SFL-V1 under clients' partial participation

---

**Input:** $\tau, \tilde{\tau}, T$, and learning rate $\eta_t$
**Output:** Global model $\boldsymbol{x}^T = \{\boldsymbol{x}_c^T, \boldsymbol{x}_s^T\}$

1   Initialize $\boldsymbol{x}^0 = \{\boldsymbol{x}_c^0, \boldsymbol{x}_s^0\}$;
2   **for** $i = 0, \ldots, (T-1)\tau_{\max}$ **do**
3     Determine participating client set $\mathcal{P}^t \subseteq \mathcal{N}$ according to $q_n$;

    **Phase 1: model training.**
4     **each client** $n \in \mathcal{P}^t$:
5       Sample a mini-batch $\zeta_n^{t,i}$;
6       Send $h(\boldsymbol{x}_{c,n}^{t,i}; \zeta_n^{t,i})$ to the M-server;
7       **the M-server:**
8         Compute $F_n(h(\boldsymbol{x}_{c,n}^{t,i}; \zeta_n^{t,i}), \boldsymbol{x}_s^{t,i})$, $\boldsymbol{g}_{s,n}^{t,i}(\boldsymbol{x}_{s,n}^{t,i}; \zeta_n^{t,i})$, and $\boldsymbol{r}_{c,n}^{t,i}(\boldsymbol{x}_{s,n}^{t,i}; \zeta_n^{t,i})$;
9         Send $\boldsymbol{r}_{c,n}^{t,i}(\boldsymbol{x}_{s,n}^{t,i}; \zeta_n^{t,i})$ to client $n \in \mathcal{P}^t$;
10         $\boldsymbol{x}_{s,n}^{t,i+1} \leftarrow \boldsymbol{x}_{s,n}^{t,i} - \eta_t \boldsymbol{g}_{s,n}^{t,i}(\boldsymbol{x}_{s,n}^{t,i}; \zeta_n^{t,i})$;
11       Compute $\boldsymbol{g}_{c,n}^{t,i}(\boldsymbol{x}_{c,n}^{t,i}; \zeta_n^{t,i})$;
12       $\boldsymbol{x}_{c,n}^{t,i+1} \leftarrow \boldsymbol{x}_{c,n}^{t,i} - \eta_t \boldsymbol{g}_{c,n}^{t,i}(\boldsymbol{x}_{c,n}^{t,i}; \zeta_n^{t,i})$;

    **Phase 2: model aggregation.**
13     **if** $i \% \tau = 0$ **then**
14       **each client** $n \in \mathcal{P}^t$:
15         Send $\boldsymbol{x}_{c,n}^{t,\tau}$ to the F-server;
16       **the F-server:**
17         $\boldsymbol{x}_c^{t+1} \leftarrow \sum_{n \in \mathcal{P}^t} \frac{a_n}{q_n} \boldsymbol{x}_{c,n}^{t,\tau}$;
18       **each client** $n \in \mathcal{N}$:
19         $\boldsymbol{x}_{c,n}^{t,0} \leftarrow \boldsymbol{x}_c^t$;

20     **if** $i \% \tilde{\tau} = 0$ **then**
21       **the M-server:**
22         $\boldsymbol{x}_s^{t+1} \leftarrow \sum_{n \in \mathcal{P}^t} \frac{a_n}{q_n} \boldsymbol{x}_{s,n}^{t,\tilde{\tau}}$.
23         $\boldsymbol{x}_{s,n}^{t,0} \leftarrow \boldsymbol{x}_s^t, \forall n \in \mathcal{N}$;

---

---

**Algorithm 2:** SFL-V2 under clients' partial participation

---

**Input:** $\tau, T$, and learning rate $\eta_t$
**Output:** Global model $\boldsymbol{x}^T = \{\boldsymbol{x}_c^T, \boldsymbol{x}_s^T\}$

1   Initialize $\boldsymbol{x}^0 = \{\boldsymbol{x}_c^0, \boldsymbol{x}_s^0\}$;
2   **for** $t = 0, \ldots, T-1$ **do**
3      Determine participating client set $\mathcal{P}^t \subseteq \mathcal{N}$ according to $q_n$;

     **Phase 1: model training.**
4      **each client** $n \in \mathcal{P}^t$**:**
5         $\boldsymbol{x}_{c,n}^{t,0} \leftarrow \boldsymbol{x}_c^t$;
6         **for** $i = 0, \ldots, \tau-1$ **do**
7             Sample a mini-batch $\zeta_n^{t,i}$;
8             Send $h(\boldsymbol{x}_{c,n}^{t,i}; \zeta_n^{t,i})$ to the M-server;
9             **the M-server:**
10                Compute $F_n(h(\boldsymbol{x}_{c,n}^{t,i}; \zeta_n^{t,i}), \boldsymbol{x}_s^{t,i}), \boldsymbol{g}_{s,n}^{t,i}(\boldsymbol{x}_s^{t,i}; \zeta_n^{t,i})$, and
                   $\boldsymbol{r}_{c,n}^{t,i}(\boldsymbol{x}_s^{t,i}; \zeta_n^{t,i})$;
11                Send $\boldsymbol{r}_{c,n}^{t,i}(\boldsymbol{x}_s^{t,i}; \zeta_n^{t,i})$ to client $n \in \mathcal{P}^t$;
12                $\boldsymbol{x}_s^{t,i+1} \leftarrow \boldsymbol{x}_s^{t,i} - \frac{\eta_t}{q_n} \boldsymbol{g}_{s,n}^{t,i}(\boldsymbol{x}_s^{t,i}; \zeta_n^{t,i})$;
13             Compute $\boldsymbol{g}_{c,n}^{t,i}(\boldsymbol{x}_{c,n}^{t,i}; \zeta_n^{t,i})$;
14             $\boldsymbol{x}_{c,n}^{t,i+1} \leftarrow \boldsymbol{x}_{c,n}^{t,i} - \eta_t \boldsymbol{g}_{c,n}^{t,i}(\boldsymbol{x}_{c,n}^{t,i}; \zeta_n^{t,i})$;

15      **the M-server:**
16         $\boldsymbol{x}_s^{t+1,0} \leftarrow \boldsymbol{x}_s^{t,\tau}$;

     **Phase 2: model aggregation.**
17      **each client** $n \in \mathcal{P}^t$**:**
18         Send $\boldsymbol{x}_{c,n}^{t,\tau}$ to the F-server;
19      **the F-server:**
20         $\boldsymbol{x}_c^{t+1} \leftarrow \sum_{n \in \mathcal{P}^t} \frac{a_n}{q_n} \boldsymbol{x}_{c,n}^{t,\tau}$.

---

# C  Notations and technical lemmas

## C.1  Notations

Recall that the objective of SFL is given by

$$\min_{\boldsymbol{x}} f(\boldsymbol{x}) := \sum_{n=1}^{N} a_n F_n(\boldsymbol{x}) \tag{23}$$

We define

- $\boldsymbol{x}_c$ and $\boldsymbol{x}_s$: global model parameter on the clients and server sides, respectively.
- $\boldsymbol{x}_{c,n}$ and $\boldsymbol{x}_{s,n}$: local forms of parameter on client $n$ and on the main server corresponding to client $n$ (in SFL-V1).
- $\nabla F_{c,n}(\cdot)$ and $\nabla F_{s,n}(\cdot)$: the gradients of $F_n(\cdot)$ over $\boldsymbol{x}_c$ and $\boldsymbol{x}_s$, respectively.
- $\boldsymbol{g}_{c,n}(\cdot)$ and $\boldsymbol{g}_{s,n}(\cdot)$: the stochastic gradients of $F_n(\cdot)$ over $\boldsymbol{x}_c$ and $\boldsymbol{x}_s$, respectively.

For convenience, we omit the notation for mini-batch training data when referring to stochastic gradients.

Further, we recall how SFL-V1 and SFL-V2 update models below.

## C.2  SFL-V1 and SFL-V2 model updates

Let $q_n$ denote the participating probability of client $n$ and define $\boldsymbol{q} := \{q_1, \ldots, q_N\}$. We denote $\mathbf{I}_n^t$ as a binary variable, taking 1 if client $n$ participates in model training in round $t$, and 0 otherwise. $\mathbf{I}_n^t$ follows a Bernoulli distribution with an expectation of $q_n$. Denote $\mathcal{P}^t(\boldsymbol{q})$ as the set of participating clients in round $t$.

**Parameter update for SFL-V1:**

- Local training of client $n$: $\boldsymbol{x}_{c,n}^{t,0} \leftarrow \boldsymbol{x}_c^t$, $\boldsymbol{x}_{c,n}^{t,i+1} \leftarrow \boldsymbol{x}_{c,n}^{t,i} - \eta^t \boldsymbol{g}_{c,n}^{t,i}\left(\boldsymbol{x}_{c,n}^{t,i}\right)$, $\boldsymbol{x}_{c,n}^{t+1} \leftarrow \boldsymbol{x}_{c,n}^{t,\tau}$;
- Client-side global aggregation:
  - Full participation: $\boldsymbol{x}_c^{t+1} \leftarrow \boldsymbol{x}_c^t - \eta^t \sum_{n \in \mathcal{N}} a_n \sum_{i=0}^{\tau} \boldsymbol{g}_{c,n}^{t,i}\left(\boldsymbol{x}_{c,n}^{t,i}\right)$;
  - Partial participation: $\boldsymbol{x}_c^{t+1} \leftarrow \boldsymbol{x}_c^t - \eta^t \sum_{n \in \mathcal{P}^t(\boldsymbol{q})} \frac{a_n}{q_n} \sum_{i=0}^{\tau} \boldsymbol{g}_{c,n}^{t,i}\left(\boldsymbol{x}_{c,n}^{t,i}\right)$;
- M-server-side model update:
  - Full participation: $\boldsymbol{x}_s^{t+1} \leftarrow \boldsymbol{x}_s^t - \eta^t \sum_{n \in \mathcal{N}} a_n \sum_{i=0}^{\tilde{\tau}-1} \boldsymbol{g}_{s,n}^{t,i}\left(\boldsymbol{x}_{s,n}^{t,i}\right)$;
  - Partial participation: $\boldsymbol{x}_s^{t+1} \leftarrow \boldsymbol{x}_s^t - \eta^t \sum_{n \in \mathcal{P}^t(\boldsymbol{q})} \frac{a_n}{q_n} \sum_{i=0}^{\tilde{\tau}-1} \boldsymbol{g}_{s,n}^{t,i}\left(\boldsymbol{x}_{s,n}^{t,i}\right)$.

**Parameter update for SFL-V2:**

- Local training of client $n$: $\boldsymbol{x}_{c,n}^{t,0} \leftarrow \boldsymbol{x}_c^t$, $\boldsymbol{x}_{c,n}^{t,i+1} \leftarrow \boldsymbol{x}_{c,n}^{t,i} - \eta^t \boldsymbol{g}_{c,n}^{t,i}\left(\boldsymbol{x}_{c,n}^{t,i}\right)$, $\boldsymbol{x}_{c,n}^{t+1} \leftarrow \boldsymbol{x}_{c,n}^{t,\tau}$;
- Client-side global aggregation:
  - Full participation: $\boldsymbol{x}_c^{t+1} \leftarrow \boldsymbol{x}_c^t - \eta^t \sum_{n \in \mathcal{N}} a_n \sum_{i=0}^{\tau} \boldsymbol{g}_{c,n}^{t,i}\left(\boldsymbol{x}_{c,n}^{t,i}\right)$;
  - Partial participation: $\boldsymbol{x}_c^{t+1} \leftarrow \boldsymbol{x}_c^t - \eta^t \sum_{n \in \mathcal{P}^t(\boldsymbol{q})} \frac{a_n}{q_n} \sum_{i=0}^{\tau} \boldsymbol{g}_{c,n}^{t,i}\left(\boldsymbol{x}_{c,n}^{t,i}\right)$;
- M-server-side model update:
  - Full participation: $\boldsymbol{x}_s^{t+1} \leftarrow \boldsymbol{x}_s^t - \eta^t \sum_{n \in \mathcal{N}} \sum_{i=0}^{\tau-1} \boldsymbol{g}_{s,n}^{t,i}\left(\boldsymbol{x}_{s,n}^{t,i}\right)$;
  - Partial participation: $\boldsymbol{x}_s^{t+1} \leftarrow \boldsymbol{x}_s^t - \eta^t \sum_{n \in \mathcal{P}^t(\boldsymbol{q})} \frac{1}{q_n} \sum_{i=0}^{\tau-1} \boldsymbol{g}_{s,n}^{t,i}\left(\boldsymbol{x}_{s,n}^{t,i}\right)$.

## C.3  Assumptions

We further recall the following assumptions for clients' loss functions in the proof.

**Assumption C.1.** For each client $n \in \mathcal{N}$:

- The loss $F_n(\cdot)$ is $S$-smooth:

$$\|\nabla F_n(\boldsymbol{x}) - \nabla F_n(\boldsymbol{y})\| \leq S\|\boldsymbol{x} - \boldsymbol{y}\|, \forall \boldsymbol{x}, \boldsymbol{y}, \tag{24}$$

$$F_n(\boldsymbol{y}) \leq F_n(\boldsymbol{x}) + \langle \nabla F_n(\boldsymbol{x}), \boldsymbol{y} - \boldsymbol{x}\rangle + \frac{S}{2}\|\boldsymbol{y} - \boldsymbol{x}\|^2, \forall \boldsymbol{x}, \boldsymbol{y} \in \mathbb{R}^d. \tag{25}$$

- The stochastic gradients of $F_n(\cdot)$ are unbiased with the variance bounded by $\sigma_n^2$:

$$\mathbb{E}[\boldsymbol{g}_n(\boldsymbol{x})] = \nabla F_n(\boldsymbol{x}), \tag{26}$$

$$\mathbb{E}\left[\|\boldsymbol{g}_n(\boldsymbol{x}) - \nabla F_n(\boldsymbol{x})\|^2\right] \leq \sigma_n^2. \tag{27}$$

- The expected squared norm of stochastic gradients is bounded by $G^2$:

$$\mathbb{E}\|\boldsymbol{g}_n(\boldsymbol{x})\|^2 \leq G^2. \tag{28}$$

- (Bounded gradient divergence) There exists a constant $\epsilon > 0$, such that the divergence between local and global gradients is bounded by $\epsilon^2$:

$$\|\nabla F_n(\mathbf{x}) - \nabla f(\mathbf{x})\|^2 \leq \epsilon^2. \tag{29}$$

**Assumption C.2.** For each client $n \in \mathcal{N}$:

- The loss $F_n(\cdot)$ is $\mu$-strongly convex for some $\mu \geq 0$:

$$F_n(\boldsymbol{y}) \geq F_n(\boldsymbol{x}) + \langle \nabla F_n(\boldsymbol{x}), \boldsymbol{y} - \boldsymbol{x}\rangle + \frac{\mu}{2}\|\boldsymbol{y} - \boldsymbol{x}\|^2, \forall \boldsymbol{x}, \boldsymbol{y} \in \mathbb{R}^d. \tag{30}$$

Here, we allow that $\mu = 0$, referring to this case of the general convex.

## C.4 Technical Lemmas

**Lemma C.3.** *[Lemma 5 in [10]] The following holds for any $S$-smooth and $\mu$-strongly convex function $h$, and any $x, y, z$ in the domain of $h$:*

$$\langle \nabla h(\boldsymbol{x}), \boldsymbol{z} - \boldsymbol{y}\rangle \geq h(\boldsymbol{z}) - h(\boldsymbol{y}) + \frac{\mu}{4}\|\boldsymbol{y} - \boldsymbol{z}\|^2 - S\|\boldsymbol{z} - \boldsymbol{x}\|^2. \tag{31}$$

**Proof of Proposition 3.5**

**Proposition 3.5** (Convergence decomposition) Let $\boldsymbol{x}^* \triangleq [\boldsymbol{x}_c^*; \boldsymbol{x}_s^*]$ denote the optimal global model that minimizes $f(\cdot)$, and $\boldsymbol{x}^T \triangleq [\boldsymbol{x}_c^T; \boldsymbol{x}_s^T]$ is the global model obtained after $T$ rounds of SFL training. Under Assumption 3.1, we have

$$\mathbb{E}\left[f(\boldsymbol{x}^T)\right] - f(\boldsymbol{x}^*) \leq \frac{S}{2}\left(\mathbb{E}\|\boldsymbol{x}_s^T - \boldsymbol{x}_s^*\|^2 + \mathbb{E}\|\boldsymbol{x}_c^T - \boldsymbol{x}_c^*\|^2\right). \tag{32}$$

*Proof.* Since $F_n$'s are $S$-smooth, it is easy to show that the global loss function $f(\cdot)$ is also $S$-smooth. Thus, we have

$$\mathbb{E}\left[f(\boldsymbol{x}^T)\right] - f(\boldsymbol{x}^*) \leq \mathbb{E}\left[\langle \boldsymbol{x}^T - \boldsymbol{x}^*, \nabla f(\boldsymbol{x}^*)\rangle\right] + \frac{S}{2}\mathbb{E}\left[\|\boldsymbol{x}^T - \boldsymbol{x}^*\|^2\right] = \frac{S}{2}\mathbb{E}\left[\|\boldsymbol{x}^T - \boldsymbol{x}^*\|^2\right]. \tag{33}$$

Since $\boldsymbol{x}^T \triangleq [\boldsymbol{x}_c^T; \boldsymbol{x}_s^T]$, and $\boldsymbol{x}^* \triangleq [\boldsymbol{x}_c^*; \boldsymbol{x}_s^*]$, we have

$$
\begin{aligned}
\mathbb{E}\left[\|\boldsymbol{x}^T - \boldsymbol{x}^*\|^2\right] &= \mathbb{E}\left[\|[\boldsymbol{x}_c^T; \boldsymbol{x}_s^T] - [\boldsymbol{x}_c^*; \boldsymbol{x}_s^*]\|^2\right] \\
&= \mathbb{E}\left[\|[\boldsymbol{x}_c^T - \boldsymbol{x}_c^*; \boldsymbol{x}_s^T - \boldsymbol{x}_s^*]\|^2\right] = \mathbb{E}\left[\|\boldsymbol{x}_c^T - \boldsymbol{x}_c^*\|^2\right] + \mathbb{E}\left[\|\boldsymbol{x}_s^T - \boldsymbol{x}_s^*\|^2\right].
\end{aligned} \tag{34}
$$

Substituting (34) into (33), we complete the proof. $\square$

**Proposition C.4** (Decomposition in each round). *Under Assumption C.1, we have*

$$\mathbb{E}\left[f\left(\boldsymbol{x}^{t+1}\right)\right] - f\left(\boldsymbol{x}^t\right)$$

$$\leq \mathbb{E}\left[\langle \nabla_{\boldsymbol{x}_c} f\left(\boldsymbol{x}^t\right), \boldsymbol{x}_c^{t+1} - \boldsymbol{x}_c^t\rangle\right] + \frac{S}{2}\mathbb{E}\left[\|\boldsymbol{x}_c^{t+1} - \boldsymbol{x}_c^t\|^2\right] + \tag{35}$$

$$\mathbb{E}\left[\langle \nabla_{\boldsymbol{x}_s} f\left(\boldsymbol{x}^t\right), \boldsymbol{x}_s^{t+1} - \boldsymbol{x}_s^t\rangle\right] + \frac{S}{2}\mathbb{E}\left[\|\boldsymbol{x}_s^{t+1} - \boldsymbol{x}_s^t\|^2\right]. \tag{36}$$

*Proof.* The proposition can be easily proved by the $S$-smoothness of $f(\cdot)$. $\qquad\square$

**Lemma C.5.** *[Multiple iterations of local training in each round] Under Assumption C.1, if we let $\eta^t \leq \frac{1}{\sqrt{6}S\tau}$ and run client $n$'s local model for $\tau$ iteration continuously in any round $t$, we have*

$$\sum_{i=0}^{\tau-1} \mathbb{E}\left[\left\|\boldsymbol{x}_n^{t,i} - \boldsymbol{x}^t\right\|^2\right] \leq 12\tau^3 \left(\eta^t\right)^2 \left(2\sigma_n^2 + G^2\right). \tag{37}$$

*Proof.* Similar to Lemma 3 in [20], we have

$$\mathbb{E}\left[\left\|\boldsymbol{x}_n^{t,i} - \boldsymbol{x}^t\right\|^2\right]$$

$$\leq \mathbb{E}\left[\left\|\boldsymbol{x}_n^{t,i-1} - \eta^t \boldsymbol{g}_n^{t,i-1} - \boldsymbol{x}^t\right\|^2\right]$$

$$\leq \mathbb{E}\left[\left\|\boldsymbol{x}_n^{t,i-1} - \boldsymbol{x}^t - \eta^t \left(\boldsymbol{g}_n^{t,i-1} - \nabla_{\boldsymbol{x}}F_n\left(\boldsymbol{x}_n^{t,i-1}\right) + \nabla_{\boldsymbol{x}}F_n\left(\boldsymbol{x}_n^{t,i-1}\right) - \nabla_{\boldsymbol{x}}F_n\left(\boldsymbol{x}^t\right) + \nabla_{\boldsymbol{x}}F_n\left(\boldsymbol{x}^t\right)\right)\right\|^2\right]$$

$$\leq \left(1 + \frac{1}{\tau}\right)\mathbb{E}\left[\left\|\boldsymbol{x}_n^{t,i-1} - \boldsymbol{x}^t\right\|^2\right] + 3(1+\tau)\mathbb{E}\left[\left\|\eta^t\left(\boldsymbol{g}_n^{t,i-1} - \nabla_{\boldsymbol{x}}F_n\left(\boldsymbol{x}_n^{t,i-1}\right)\right)\right\|^2\right]$$

$$+ 3(1+\tau)\mathbb{E}\left[\left\|\eta^t\left(\nabla_{\boldsymbol{x}}F_n\left(\boldsymbol{x}_n^{t,i-1}\right) - \nabla_{\boldsymbol{x}}F_n\left(\boldsymbol{x}^t\right)\right)\right\|^2\right] + 3(1+\tau)\mathbb{E}\left[\left\|\eta^t\left(\nabla_{\boldsymbol{x}}F_n\left(\boldsymbol{x}^t\right)\right)\right\|^2\right]$$

$$\leq \left(1 + \frac{1}{\tau}\right)\mathbb{E}\left[\left\|\boldsymbol{x}_n^{t,i-1} - \boldsymbol{x}^t\right\|^2\right] + 3(1+\tau)\left(\eta^t\right)^2 \sigma_n^2$$

$$+ 3(1+\tau)\left(\eta^t\right)^2 S^2 \mathbb{E}\left[\left\|\boldsymbol{x}_n^{t,i-1} - \boldsymbol{x}^t\right\|^2\right] + 3(1+\tau)\left(\eta^t\right)^2 \mathbb{E}\left[\left\|\nabla_{\boldsymbol{x}}F_n\left(\boldsymbol{x}^t\right)\right\|^2\right]$$

$$\leq \left(1 + \frac{1}{\tau} + 6\tau\left(\eta^t\right)^2 S^2\right)\mathbb{E}\left[\left\|\boldsymbol{x}_n^{t,i-1} - \boldsymbol{x}^t\right\|^2\right] + 6\tau\left(\eta^t\right)^2 \sigma_n^2 + 6\tau\left(\eta^t\right)^2 \mathbb{E}\left[\left\|\nabla_{\boldsymbol{x}}F_n\left(\boldsymbol{x}^t\right)\right\|^2\right]$$

$$\leq \left(1 + \frac{2}{\tau}\right)\mathbb{E}\left[\left\|\boldsymbol{x}_n^{t,i-1} - \boldsymbol{x}^t\right\|^2\right] + 6\tau\left(\eta^t\right)^2 \sigma_n^2 + 6\tau\left(\eta^t\right)^2 \mathbb{E}\left[\left\|\nabla_{\boldsymbol{x}}F_n\left(\boldsymbol{x}^t\right)\right\|^2\right],$$

$$\leq \left(1 + \frac{2}{\tau}\right)\mathbb{E}\left[\left\|\boldsymbol{x}_n^{t,i-1} - \boldsymbol{x}^t\right\|^2\right] + 6\tau\left(\eta^t\right)^2 \sigma_n^2 + 6\tau\left(\eta^t\right)^2 \left(\mathbb{E}\left[\left\|\nabla_{\boldsymbol{x}}F_n\left(\boldsymbol{x}^t\right) - \boldsymbol{g}_n^t\right\|^2\right] + \mathbb{E}\left[\left\|\nabla_{\boldsymbol{x}}\boldsymbol{g}_n^t\right\|^2\right]\right),$$

$$\leq \left(1 + \frac{2}{\tau}\right)\mathbb{E}\left[\left\|\boldsymbol{x}_n^{t,i-1} - \boldsymbol{x}^t\right\|^2\right] + 6\tau\left(\eta^t\right)^2 \left(2\sigma_n^2 + G^2\right), \tag{38}$$

where we use Assumption C.1, $(X + Y)^2 \leq (1+a)X^2 + \left(1 + \frac{1}{a}\right)Y^2$ for some positive $a$, and $\eta^t \leq \frac{1}{\sqrt{6}S\tau}$.

Let

$$A^{t,i} := \mathbb{E}\left[\left\|\boldsymbol{x}_n^{t,i} - \boldsymbol{x}^t\right\|^2\right]$$

$$B := 6\tau\left(\eta^t\right)^2 \left(2\sigma_n^2 + G^2\right)$$

$$C := 1 + \frac{2}{\tau}$$

We have

$$A^{t,i} \leq CA^{t,i-1} + B \tag{39}$$

We can show that

$$A^{t,1} \leq CA^t + B$$
$$A^{t,2} \leq CA^{t,1} + B \leq C^2 A^t + CB + B$$
$$A^{t,3} \leq CA^{t,2} + B \leq C^3 A^t + C^2 B + CB + B$$
$$\cdots$$
$$A^{t,i} \leq C^i A^t + B\sum_{j=0}^{i-1} C^j$$

Note that $A^t := A^{t,0} = \mathbb{E}\left[\left\|\boldsymbol{x}^t - \boldsymbol{x}^t\right\|^2\right] = 0$. Accumulate the above for $\tau$ iterations, we have

$$\sum_{i=0}^{\tau-1} \mathbb{E}\left[\left\|\boldsymbol{x}_n^{t,i} - \boldsymbol{x}^t\right\|^2\right] = \sum_{i=0}^{\tau-1} B \sum_{j=0}^{i-1} C^j$$

$$= B\sum_{i=0}^{\tau-1} \frac{C^i - 1}{C - 1} = \frac{B}{C-1} \sum_{i=0}^{\tau-1}\left(C^i - 1\right) = \frac{B}{C-1}\left(\frac{C^\tau - 1}{C-1} - \tau\right)$$

$$= \frac{B}{\frac{2}{\tau}}\left(\frac{\left(1 + \frac{2}{\tau}\right)^\tau - 1}{\frac{2}{\tau}} - \tau\right) \tag{40}$$

$$\leq \frac{\tau^2 B}{2}\left(\frac{e^2 - 1}{2} - 1\right)$$

$$\leq 2\tau^2 B$$

$$\leq 2\tau^2 6\tau\left(\eta^t\right)^2\left(2\sigma_n^2 + G^2\right)$$

$$\leq 12\tau^3\left(\eta^t\right)^2\left(2\sigma_n^2 + G^2\right). \tag{41}$$

The first inequality is due to $\sum_{i=0}^{N-1} x^i = \frac{x^N - 1}{X - 1}$ and the third line results from $\left(1 + \frac{n}{x}\right)^x \leq e^n$. Thus, we finish the proof. $\qquad\square$

**Lemma C.6.** *[Multiple iterations of local training in each round] Under Assumption C.1, if we let $\eta^t \leq \frac{1}{\sqrt{8}S\tau}$ and run client $n$'s local model for $\tau$ iteration continuously in any round $t$, we have*

$$\sum_{i=0}^{\tau-1} \mathbb{E}\left[\left\|\boldsymbol{x}_n^{t,i} - \boldsymbol{x}^t\right\|^2\right] \leq 2\tau^2\left(8\tau\left(\eta^t\right)^2 \sigma_n^2 + 8\tau\left(\eta^t\right)^2 \epsilon^2 + 8\tau\left(\eta^t\right)^2\left\|\nabla_{\boldsymbol{x}} f\left(\boldsymbol{x}^t\right)\right\|^2\right). \tag{42}$$

*Proof.*

$$\mathbb{E}\left[\left\|\boldsymbol{x}_n^{t,i} - \boldsymbol{x}^t\right\|^2\right]$$

$$\leq \mathbb{E}\left[\left\|\boldsymbol{x}_n^{t,i-1} - \eta^t \boldsymbol{g}_n^{t,i-1} - \boldsymbol{x}^t\right\|^2\right]$$

$$\leq \mathbb{E}\left[\left\|\boldsymbol{x}_n^{t,i-1} - \boldsymbol{x}^t - \eta^t\left(\boldsymbol{g}_n^{t,i-1} - \nabla_{\boldsymbol{x}} F_n\left(\boldsymbol{x}_n^{t,i-1}\right)\right.\right.\right.$$

$$\left.\left.\left. + \nabla_{\boldsymbol{x}} F_n\left(\boldsymbol{x}_n^{t,i-1}\right) - \nabla_{\boldsymbol{x}} F_n\left(\boldsymbol{x}^t\right) + \nabla_{\boldsymbol{x}} F_n\left(\boldsymbol{x}^t\right) - \nabla_{\boldsymbol{x}} f\left(\boldsymbol{x}^t\right) + \nabla_{\boldsymbol{x}} f\left(\boldsymbol{x}^t\right)\right)\right\|^2\right]$$

$$\leq \left(1 + \frac{1}{\tau}\right)\mathbb{E}\left[\left\|\boldsymbol{x}_n^{t,i-1} - \boldsymbol{x}^t\right\|^2\right] + 8\tau\mathbb{E}\left[\left\|\eta^t\left(\boldsymbol{g}_n^{t,i-1} - \nabla_{\boldsymbol{x}} F_n\left(\boldsymbol{x}_n^{t,i-1}\right)\right)\right\|^2\right]$$

$$+ 8\tau\mathbb{E}\left[\left\|\eta^t\left(\nabla_{\boldsymbol{x}} F_n\left(\boldsymbol{x}_n^{t,i-1}\right) - \nabla_{\boldsymbol{x}} F_n\left(\boldsymbol{x}^t\right)\right)\right\|^2\right] + 8\tau\mathbb{E}\left[\left\|\eta^t\left(\nabla_{\boldsymbol{x}} F_n\left(\boldsymbol{x}^t\right) - \nabla_{\boldsymbol{x}} f\left(\boldsymbol{x}^t\right)\right)\right\|^2\right]$$

$$+ 8\tau\left\|\eta^t \nabla_{\boldsymbol{x}} f\left(\boldsymbol{x}^t\right)\right\|^2$$

$$\leq \left(1 + \frac{1}{\tau}\right)\mathbb{E}\left[\left\|\boldsymbol{x}_n^{t,i-1} - \boldsymbol{x}^t\right\|^2\right] + 8\tau\left(\eta^t\right)^2 \sigma_n^2 + 8\tau\left(\eta^t\right)^2 S^2\mathbb{E}\left[\left\|\boldsymbol{x}_n^{t,i-1} - \boldsymbol{x}^t\right\|^2\right] + 8\tau\left(\eta^t\right)^2 \epsilon^2$$

$$+ 8\tau\left(\eta^t\right)^2\left\|\nabla_{\boldsymbol{x}} f\left(\boldsymbol{x}^t\right)\right\|^2$$

$$\leq \left(1 + \frac{1}{\tau} + 8\tau\left(\eta^t\right)^2 S^2\right)\mathbb{E}\left[\left\|\boldsymbol{x}_n^{t,i-1} - \boldsymbol{x}^t\right\|^2\right] + 8\tau\left(\eta^t\right)^2 \sigma_n^2 + 8\tau\left(\eta^t\right)^2 \epsilon^2 + 8\tau\left(\eta^t\right)^2\left\|\nabla_{\boldsymbol{x}} f\left(\boldsymbol{x}^t\right)\right\|^2$$

$$\leq \left(1 + \frac{2}{\tau}\right)\mathbb{E}\left[\left\|\boldsymbol{x}_n^{t,i-1} - \boldsymbol{x}^t\right\|^2\right] + 8\tau\left(\eta^t\right)^2 \sigma_n^2 + 8\tau\left(\eta^t\right)^2 \epsilon^2 + 8\tau\left(\eta^t\right)^2\left\|\nabla_{\boldsymbol{x}} f\left(\boldsymbol{x}^t\right)\right\|^2 \tag{43}$$

where we have applied Assumption C.1, $(X + Y)^2 \leq (1 + a) X^2 + \left(1 + \frac{1}{a}\right) Y^2$ for some positive $a$, and $\eta^t \leq \frac{1}{\sqrt{8}S\tau}$.

Let

$$A_{t,i} := \mathbb{E}\left[\left\|\boldsymbol{x}_n^{t,i} - \boldsymbol{x}^t\right\|^2\right]$$

$$B := 8\tau \left(\eta^t\right)^2 \sigma_n^2 + 8\tau \left(\eta^t\right)^2 \epsilon^2 + 8\tau \left(\eta^t\right)^2 \left\|\nabla_{\boldsymbol{x}} f\left(\boldsymbol{x}^t\right)\right\|^2$$

$$C := 1 + \frac{2}{\tau}$$

We have

$$A_{t,i} \leq CA_{t,i-1} + B \tag{44}$$

We can show that

$$A_{t,i} \leq C^i A_t + B \sum_{j=0}^{i-1} C^j$$

Note that $A_t = \mathbb{E}\left[\left\|\boldsymbol{x}^t - \boldsymbol{x}^t\right\|^2\right] = 0$. Accumulate the above for $\tau$ iterations, we have

$$\sum_{i=0}^{\tau-1} \mathbb{E}\left[\left\|\boldsymbol{x}_n^{t,i} - \boldsymbol{x}^t\right\|^2\right] = \sum_{i=0}^{\tau-1} B \sum_{j=0}^{i-1} C^j$$

$$\leq 2\tau^2 B$$

$$\leq 2\tau^2 \left(8\tau \left(\eta^t\right)^2 \sigma_n^2 + 8\tau \left(\eta^t\right)^2 \epsilon^2 + 8\tau \left(\eta^t\right)^2 \left\|\nabla_{\boldsymbol{x}} f\left(\boldsymbol{x}^t\right)\right\|^2\right) \tag{45}$$

where we use $\sum_{i=0}^{N-1} x^i = \frac{x^N - 1}{X-1}$ and $\left(1 + \frac{n}{x}\right)^x \leq e^n$. Therefore, we complete the proof. $\square$

**Lemma C.7.** *[Multiple iterations of local gradient accumulation in each round] Under Assumption C.1, if we let $\eta^t \leq \frac{1}{2S\tau}$ and run client $n$'s local model for $\tau$ iteration continuously in any round $t$, we have*

$$\sum_{i=0}^{\tau-1} \mathbb{E}\left[\left\|\boldsymbol{g}_n^{t,i} - \boldsymbol{g}_n^t\right\|^2\right] \leq 8\tau^3 \left(\eta^t\right)^2 S^2 \left(\left\|\nabla_{\boldsymbol{x}} F_n\left(\boldsymbol{x}^t\right)\right\|^2 + \sigma_n^2\right). \tag{46}$$

*Proof.*

$$\mathbb{E}\left[\left\|\boldsymbol{g}_n^{t,i} - \boldsymbol{g}_n^t\right\|^2\right]$$

$$\leq \mathbb{E}\left[\left\|\boldsymbol{g}_n^{t,i} - \boldsymbol{g}_n^{t,i-1} + \boldsymbol{g}_n^{t,i-1} - \boldsymbol{g}_n^t\right\|^2\right]$$

$$\leq (1+\tau) \mathbb{E}\left[\left\|\boldsymbol{g}_n^{t,i} - \boldsymbol{g}_n^{t,i-1}\right\|^2\right] + \left(1 + \frac{1}{\tau}\right) \mathbb{E}\left[\left\|\boldsymbol{g}_n^{t,i-1} - \boldsymbol{g}_n^t\right\|^2\right]$$

$$\leq (1+\tau) S^2 \mathbb{E}\left[\left\|\boldsymbol{x}_n^{t,i} - \boldsymbol{x}_n^{t,i-1}\right\|^2\right] + \left(1 + \frac{1}{\tau}\right) \mathbb{E}\left[\left\|\boldsymbol{g}_n^{t,i-1} - \boldsymbol{g}_n^t\right\|^2\right]$$

$$\leq (1+\tau) \left(\eta^t\right)^2 S^2 \mathbb{E}\left[\left\|\boldsymbol{g}_n^{t,i-1}\right\|^2\right] + \left(1 + \frac{1}{\tau}\right) \mathbb{E}\left[\left\|\boldsymbol{g}_n^{t,i-1} - \boldsymbol{g}_n^t\right\|^2\right]$$

$$\leq (1+\tau) \left(\eta^t\right)^2 S^2 \mathbb{E}\left[\left\|\boldsymbol{g}_n^{t,i-1} - \boldsymbol{g}_n^t + \boldsymbol{g}_n^t\right\|^2\right] + \left(1 + \frac{1}{\tau}\right) \mathbb{E}\left[\left\|\boldsymbol{g}_n^{t,i-1} - \boldsymbol{g}_n^t\right\|^2\right]$$

$$\leq 2(1+\tau) \left(\eta^t\right)^2 S^2 \mathbb{E}\left[\left\|\boldsymbol{g}_n^{t,i-1} - \boldsymbol{g}_n^t\right\|^2\right] + 2(1+\tau) \left(\eta^t\right)^2 S^2 \mathbb{E}\left[\left\|\boldsymbol{g}_n^t\right\|^2\right]$$

$$+ \left(1 + \frac{1}{\tau}\right) \mathbb{E}\left[\left\|\boldsymbol{g}_n^{t,i-1} - \boldsymbol{g}_n^t\right\|^2\right]$$

$$\leq \left(1 + \frac{2}{\tau}\right) \mathbb{E}\left[\left\|\boldsymbol{g}_n^{t,i-1} - \boldsymbol{g}_n^t\right\|^2\right] + 2(1+\tau) \left(\eta^t\right)^2 S^2 \mathbb{E}\left[\left\|\boldsymbol{g}_n^t\right\|^2\right]. \tag{47}$$

We define the following notation for simplicity:

$$A_{t,i} := \mathbb{E}\left[\left\|\boldsymbol{g}_n^{t,i} - \boldsymbol{g}_n^t\right\|^2\right] \tag{48}$$

$$B := 2 \left( 1 + \tau \right) \left( \eta^t \right)^2 S^2 \mathbb{E} \left[ \left\| \boldsymbol{g}_n^t \right\|^2 \right] \tag{49}$$

$$C := \left( 1 + \frac{2}{\tau} \right) \tag{50}$$

We have

$$A_{t,i} \leq C A_{t,i-1} + B \tag{51}$$

We can show that

$$A_{t,i} \leq C^i A_t + B \sum_{j=0}^{i-1} C^j$$

Note that $A_t = \mathbb{E} \left[ \left\| \boldsymbol{g}_n^t - \boldsymbol{g}_n^t \right\|^2 \right] = 0$. For the second part, we have

$$\begin{aligned}
\sum_{i=0}^{\tau-1} \mathbb{E} \left[ \left\| \boldsymbol{g}_n^{t,i} - \boldsymbol{g}_n^t \right\|^2 \right] &= \sum_{i=0}^{\tau-1} B \sum_{j=0}^{i-1} C^j \leq 2\tau^2 B \\
&\leq 4\tau^2 \left( 1 + \tau \right) \left( \eta^t \right)^2 S^2 \mathbb{E} \left[ \left\| \boldsymbol{g}_n^t \right\|^2 \right] \\
&\leq 8\tau^3 \left( \eta^t \right)^2 S^2 \mathbb{E} \left[ \left\| \boldsymbol{g}_n^t \right\|^2 \right] \\
&\leq 8\tau^3 \left( \eta^t \right)^2 S^2 \left( \left\| \nabla_{\boldsymbol{x}} F_n \left( \boldsymbol{x}^t \right) \right\|^2 + \sigma_n^2 \right).
\end{aligned} \tag{52}$$

$\square$

# D Proof for Theorem 3.6

We organize the proof of Theorem 3.6 as follows:

- In Sec. D.1, we prove the strongly convex case.
- In Sec. D.2, we prove the general convex case.
- In Sec. D.3, we prove the non-convex case.

## D.1 Strongly convex case for SFL-V1

### D.1.1 One-round Parallel Update for M-Server-Side Model

**Lemma D.1.** *Under Assumptions C.1 and C.2, if $\eta^t \leq \frac{1}{2S\tilde{\tau}}$, in round t, the M-server-side model evolves as*

$$
\mathbb{E}\left[\left\|\boldsymbol{x}_s^{t+1} - \boldsymbol{x}_s^*\right\|^2\right]
$$
$$
\leq \left(1 - \frac{\eta^t \tilde{\tau} \mu}{2}\right) \mathbb{E}\left[\left\|\boldsymbol{x}_s^t - \boldsymbol{x}_s^*\right\|^2\right] - 2\eta^t \tilde{\tau} \mathbb{E}\left[f\left(\boldsymbol{x}^t\right) - f\left(\boldsymbol{x}^*\right)\right]
$$
$$
+ \left(\eta^t\right)^2 \left(\tilde{\tau}\right)^2 N \sum_{n=1}^N a_n^2 \left(2\sigma_n^2 + G^2\right) + 24S \left(\tilde{\tau}\right)^3 \left(\eta^t\right)^3 \sum_{n=1}^N a_n \left(2\sigma_n^2 + G^2\right). \tag{53}
$$

We prove Lemma D.1 as follows.

*Proof.* We use $\boldsymbol{x}_{s,n}^{t,i}$ as the M-server-side model when the M-server interacts with client $n$ for the $i$-th iteration of model training at round $t$. Using the (sequential) gradient update rule of $\boldsymbol{x}_s^{t+1} = \boldsymbol{x}_s^t - \eta^t \sum_{n=1}^N \sum_{i=0}^{\tilde{\tau}-1} a_n \boldsymbol{g}_{s,n}^{t,i}\left(\left\{\boldsymbol{x}_{c,n}^{t,i}, \boldsymbol{x}_{s,n}^{t,i}\right\}\right)$, we have

$$
\mathbb{E}\left[\left\|\boldsymbol{x}_s^{t+1} - \boldsymbol{x}_s^*\right\|^2\right]
$$
$$
= \mathbb{E}\left[\left\|\boldsymbol{x}_s^t - \eta^t \sum_{i=0}^{\tilde{\tau}-1} \boldsymbol{g}_s^{t,i} - \boldsymbol{x}_s^* - \eta^t \sum_{i=0}^{\tilde{\tau}-1} \nabla_{\boldsymbol{x}_s} f\left(\left\{\boldsymbol{x}_c^{t,i}, \boldsymbol{x}_s^{t,i}\right\}\right) + \eta^t \sum_{i=0}^{\tilde{\tau}-1} \nabla_{\boldsymbol{x}_s} f\left(\left\{\boldsymbol{x}_c^{t,i}, \boldsymbol{x}_s^{t,i}\right\}\right)\right\|^2\right]
$$
$$
= \mathbb{E}\left[\left\|\boldsymbol{x}_s^t - \boldsymbol{x}_s^* - \eta^t \sum_{i=0}^{\tilde{\tau}-1} \nabla_{\boldsymbol{x}_s} f\left(\left\{\boldsymbol{x}_c^{t,i}, \boldsymbol{x}_s^{t,i}\right\}\right)\right\|^2\right]
$$
$$
+ 2\eta^t \mathbb{E}\left[\left\langle \boldsymbol{x}_s^t - \boldsymbol{x}_s^* - \eta^t \sum_{i=0}^{\tilde{\tau}-1} \nabla_{\boldsymbol{x}_s} f\left(\left\{\boldsymbol{x}_c^{t,i}, \boldsymbol{x}_s^{t,i}\right\}\right), \sum_{i=0}^{\tilde{\tau}-1} \nabla_{\boldsymbol{x}_s} f\left(\left\{\boldsymbol{x}_c^{t,i}, \boldsymbol{x}_s^{t,i}\right\}\right) - \sum_{i=0}^{\tilde{\tau}-1} \boldsymbol{g}_s^{t,i}\right\rangle\right]
$$
$$
+ \mathbb{E}\left[\left(\eta^t\right)^2 \left\|\sum_{i=0}^{\tilde{\tau}-1} \boldsymbol{g}_s^{t,i} - \sum_{i=0}^{\tilde{\tau}-1} \nabla_{\boldsymbol{x}_s} f\left(\left\{\boldsymbol{x}_c^{t,i}, \boldsymbol{x}_s^{t,i}\right\}\right)\right\|^2\right]
$$
$$
\leq \mathbb{E}\left[\left\|\boldsymbol{x}_s^t - \boldsymbol{x}_s^* - \eta^t \sum_{i=0}^{\tilde{\tau}-1} \nabla_{\boldsymbol{x}_s} f\left(\left\{\boldsymbol{x}_c^{t,i}, \boldsymbol{x}_s^{t,i}\right\}\right)\right\|^2\right]
$$
$$
+ \left(\eta^t\right)^2 \mathbb{E}\left[\left\|\sum_{n=1}^N \sum_{i=0}^{\tilde{\tau}-1} a_n \boldsymbol{g}_{s,n}^{t,i} - \sum_{i=0}^{\tilde{\tau}-1} \nabla_{\boldsymbol{x}_s} f\left(\left\{\boldsymbol{x}_c^{t,i}, \boldsymbol{x}_s^{t,i}\right\}\right)\right\|^2\right]. \tag{54}
$$

where the second equality is from $(a+b)^2 = a^2 + 2ab + b^2$ and the last inequality is due to $\mathbb{E}\left[\nabla_{\boldsymbol{x}_s} f\left(\left\{\boldsymbol{x}_c^{t,i}, \boldsymbol{x}_s^{t,i}\right\}\right) - \boldsymbol{g}_s^{t,i}\right] = 0$.

The first part in (54) is

$$
\mathbb{E}\left[\left\|\boldsymbol{x}_s^t - \boldsymbol{x}_s^* - \eta^t \sum_{i=0}^{\tilde{\tau}-1} \nabla_{\boldsymbol{x}_s} f\left(\left\{\boldsymbol{x}_c^{t,i}, \boldsymbol{x}_s^{t,i}\right\}\right)\right\|^2\right]
$$

$$\leq \mathbb{E}\left[\left\|\boldsymbol{x}_s^t - \boldsymbol{x}_s^*\right\|^2\right] + \left(\eta^t\right)^2 \tilde{\tau} N \sum_{n=1}^N \sum_{i=0}^{\tilde{\tau}-1} a_n^2 \mathbb{E}\left[\left\|\nabla_{\boldsymbol{x}_s} F_n\left(\{\boldsymbol{x}_{c,n}^{t,i}, \boldsymbol{x}_{s,n}^{t,i}\}\right)\right\|^2\right]$$

$$- 2\eta^t \mathbb{E}\left[\sum_{n=1}^N \sum_{i=0}^{\tilde{\tau}-1} a_n \left\langle \boldsymbol{x}_s^t - \boldsymbol{x}_s^*, \nabla_{\boldsymbol{x}_s} F_n\left(\{\boldsymbol{x}_c^{t,i}, \boldsymbol{x}_s^{t,i}\}\right)\right\rangle\right], \tag{55}$$

where we use $\nabla_{\boldsymbol{x}_s} f\left(\{\boldsymbol{x}_c, \boldsymbol{x}_s\}\right) = \sum_{n=1}^N a_n \nabla_{\boldsymbol{x}_s} F_n\left(\{\boldsymbol{x}_c, \boldsymbol{x}_s\}\right)$.

For (55), we have

$$\left(\eta^t\right)^2 \tilde{\tau} N \sum_{n=1}^N \sum_{i=0}^{\tilde{\tau}-1} a_n^2 \mathbb{E}\left[\left\|\nabla_{\boldsymbol{x}_s} F_n\left(\{\boldsymbol{x}_{c,n}^{t,i}, \boldsymbol{x}_{s,n}^{t,i}\}\right)\right\|^2\right]$$

$$= \left(\eta^t\right)^2 \tilde{\tau} N \sum_{n=1}^N \sum_{i=0}^{\tilde{\tau}-1} a_n^2 \mathbb{E}\left[\left\|\nabla_{\boldsymbol{x}_s} F_n\left(\{\boldsymbol{x}_{c,n}^{t,i}, \boldsymbol{x}_{s,n}^{t,i}\}\right)\right.\right.$$

$$\left.\left. - \boldsymbol{g}_{s,n}^{t,i}\left(\{\boldsymbol{x}_{c,n}^{t,i}, \boldsymbol{x}_{s,n}^{t,i}\}\right)\right\|^2\right] + \mathbb{E}\left[\left\|\boldsymbol{g}_{s,n}^{t,i}\left(\{\boldsymbol{x}_{c,n}^{t,i}, \boldsymbol{x}_{s,n}^{t,i}\}\right)\right\|^2\right]$$

$$\leq \left(\eta^t\right)^2 (\tilde{\tau})^2 N \sum_{n=1}^N a_n^2 \left(\sigma_n^2 + G^2\right), \tag{56}$$

where the first inequality applies triangle inequality. In the last inequality, we apply the bound of variance and expected squared norm for stochastic gradients in Assumption C.1.

Since $F_n(\boldsymbol{x})$ is $S$-smooth and $\mu$-strongly convex, using Lemma C.3 we have

$$- 2\eta^t \mathbb{E}\left[\sum_{n=1}^N \sum_{i=0}^{\tilde{\tau}-1} a_n \left\langle \boldsymbol{x}_s^t - \boldsymbol{x}_s^*, \nabla_{\boldsymbol{x}_s} F_n\left(\{\boldsymbol{x}_{c,n}^{t,i}, \boldsymbol{x}_{s,n}^{t,i}\}\right)\right\rangle\right]$$

$$\leq -2\eta^t \sum_{n=1}^N \sum_{i=0}^{\tilde{\tau}-1} a_n \mathbb{E}\left[\left(F_n\left(\boldsymbol{x}^t\right) - F_n\left(\boldsymbol{x}^*\right)\right.\right.$$

$$\left.\left. + \frac{\mu}{4}\left\|\boldsymbol{x}_s^t - \boldsymbol{x}_s^*\right\|^2 - S\left\|\boldsymbol{x}_{s,n}^{t,i} - \boldsymbol{x}_s^t\right\|^2\right)\right]. \tag{57}$$

By Lemma C.5, we have

$$\sum_{n=1}^N \sum_{i=0}^{\tilde{\tau}-1} a_n \mathbb{E}\left[\left\|\boldsymbol{x}_{s,n}^{t,i} - \boldsymbol{x}_s^t\right\|^2\right] \leq 12 \sum_{n=1}^N a_n (\tilde{\tau})^3 \left(\eta^t\right)^2 \left(2\sigma_n^2 + G^2\right). \tag{58}$$

From Assumption C.1, the second part in (54) is bounded by

$$\mathbb{E}\left\|\sum_{n=1}^N \sum_{i=0}^{\tilde{\tau}-1} a_n \boldsymbol{g}_{s,n}^{t,i} - \sum_{i=0}^{\tilde{\tau}-1} \nabla_{\boldsymbol{x}_s} f\left(\{\boldsymbol{x}_c^{t,i}, \boldsymbol{x}_s^{t,i}\}\right)\right\|^2$$

$$\leq \tilde{\tau} \sum_{i=0}^{\tilde{\tau}-1} \mathbb{E}\left\|\sum_{n=1}^N a_n\left(\boldsymbol{g}_{s,n}^{t,i} - \nabla_{\boldsymbol{x}_s} F_n\left(\{\boldsymbol{x}_c^{t,i}, \boldsymbol{x}_s^{t,i}\}\right)\right)\right\|^2$$

$$\leq N \sum_{n=1}^N a_n^2 \sigma_n^2 (\tilde{\tau})^2. \tag{59}$$

Thus, by $\sum_{n=1}^N a_n = 1$, (54) becomes

$$\mathbb{E}\left[\left\|\boldsymbol{x}_s^{t+1} - \boldsymbol{x}_s^*\right\|^2\right]$$

$$\leq \mathbb{E}\left[\left\|\boldsymbol{x}_s^t - \boldsymbol{x}_s^*\right\|^2\right] + \left(\eta^t\right)^2 (\tilde{\tau})^2 N \sum_{n=1}^N a_n^2 \left(\sigma_n^2 + G^2\right)$$

$$- 2\eta^t \tilde{\tau} \sum_{n=1}^{N} a_n \mathbb{E}\left[f\left(\boldsymbol{x}^t\right) - f\left(\boldsymbol{x}^*\right)\right]$$

$$- \frac{\mu \tilde{\tau} \eta^t \sum_{n=1}^{N} a_n}{2} \left\|\boldsymbol{x}_s^t - \boldsymbol{x}_s^*\right\|^2 + 2\eta^t \left(12 \sum_{n=1}^{N} a_n S\left(\tilde{\tau}\right)^3 \left(\eta^t\right)^2 \left(2\sigma_n^2 + G^2\right)\right)$$

$$+ N \sum_{n=1}^{N} a_n^2 \left(\eta^t\right)^2 \sigma_n^2 \left(\tilde{\tau}\right)^2$$

$$\leq \left(1 - \frac{\eta^t \tilde{\tau} \mu}{2}\right) \mathbb{E}\left[\left\|\boldsymbol{x}_s^t - \boldsymbol{x}_s^*\right\|^2\right] - 2\eta^t \tilde{\tau} \mathbb{E}\left[f\left(\boldsymbol{x}^t\right) - f\left(\boldsymbol{x}^*\right)\right]$$

$$+ \left(\eta^t\right)^2 \left(\tilde{\tau}\right)^2 N \sum_{n=1}^{N} a_n^2 \left(2\sigma_n^2 + G^2\right) + 24S\left(\tilde{\tau}\right)^3 \left(\eta^t\right)^3 \sum_{n=1}^{N} a_n \left(2\sigma_n^2 + G^2\right). \tag{60}$$

$$\square$$

We now prove the convergence error. Let $\Delta^{t+1} \triangleq \mathbb{E}\left[\left\|\boldsymbol{x}_s^{t+1} - \boldsymbol{x}_s^*\right\|^2\right]$. We can rewrite (60) as:

$$\Delta^{t+1} \leq \left(1 - \frac{\eta^t \tilde{\tau} \mu}{2}\right) \Delta^t - 2\eta^t \tilde{\tau} \mathbb{E}\left[f\left(\boldsymbol{x}^t\right) - f\left(\boldsymbol{x}^*\right)\right],$$

$$+ \left(\eta^t\right)^2 \left(\tilde{\tau}\right)^2 N \sum_{n=1}^{N} \left(2\sigma_n^2 + G^2\right) + 24S\left(\tilde{\tau}\right)^3 \left(\eta^t\right)^3 \sum_{n=1}^{N} \left(2\sigma_n^2 + G^2\right),$$

$$\leq \left(1 - \frac{\eta^t \tilde{\tau} \mu}{2}\right) \Delta^t + \frac{\left(\eta^t\right)^2 \left(\tilde{\tau}\right)^2}{4} B_1 + \frac{\left(\eta^t\right)^3 \left(\tilde{\tau}\right)^3}{8} B_2. \tag{61}$$

where $B_1 := 4N \sum_{n=1}^{N} a_n^2 \left(2\sigma_n^2 + G^2\right)$ and $B := 192S \sum_{n=1}^{N} a_n \left(2\sigma_n^2 + G^2\right)$.

Consider a diminishing stepsize $\eta^t = \frac{2\beta}{\tilde{\tau}(\gamma+t)}$, i.e, $\frac{\eta^t \tilde{\tau}}{2} = \frac{\beta}{\gamma+t}$, where $\beta = \frac{2}{\mu}, \gamma = \frac{8S}{\mu} - 1$. It is easy to show that $\eta^t \leq \frac{1}{2S\tilde{\tau}}$ for all $t$. Next, we will prove that $\Delta^{t+1} \leq \frac{v}{\gamma+t+1}$, where $v = \max\left\{\frac{4B_1}{\mu^2} + \frac{8B_2}{\mu^3(\gamma+1)}, (\gamma+1)\Delta^0\right\}$. We prove this by induction. First, the definition of $v$ ensures that it holds for $t = -1$. Assume the conclusion holds for some $t$, it follows that

$$\Delta^{t+1} \leq \left(1 - \frac{\eta^t \tilde{\tau} \mu}{2}\right) \Delta^t + \frac{\left(\eta^t\right)^2 \left(\tilde{\tau}\right)^2}{4} B_1 + \frac{\left(\eta^t\right)^3 \left(\tilde{\tau}\right)^3}{8} B_2 \tag{62}$$

$$\leq \left(1 - \frac{\mu\beta}{\gamma+t}\right) \frac{v}{\gamma+t} + \frac{\left(\eta^t\right)^2 \left(\tilde{\tau}\right)^2}{4} B_1 + \frac{\left(\eta^t\right)^3 \left(\tilde{\tau}\right)^3}{8} B_2 \tag{63}$$

$$= \frac{\gamma+t-1}{(\gamma+t)^2} v + \left[\frac{\beta^2 B_1}{(\gamma+t)^2} + \frac{\beta^3 B_2}{(\gamma+t)^3} - \frac{\beta\mu-1}{(\gamma+t)^2} v\right] \tag{64}$$

$$= \frac{\gamma+t-1}{(\gamma+t)^2} v + \left[\frac{\beta^2 B_1}{(\gamma+t)^2} + \frac{\beta^3 B_2}{(\gamma+t)^3} - \frac{\beta\mu-1}{(\gamma+t)^2} \max\left\{\frac{4B_1}{\mu^2} + \frac{8B_2}{\mu^3(\gamma+1)}, (\gamma+1)\Delta^0\right\}\right] \tag{65}$$

$$= \frac{\gamma+t-1}{(\gamma+t)^2} v + \left[\frac{\beta^2 B_1}{(\gamma+t)^2} + \frac{\beta^3 B_2}{(\gamma+t)^3} - \frac{\beta\mu-1}{(\gamma+t)^2} \max\left\{\frac{\beta^2 B_1}{\beta\mu-1} + \frac{\beta^3 B_2}{(\beta\mu-1)(\gamma+1)}, (\gamma+1)\Delta^0\right\}\right] \tag{66}$$

$$\leq \frac{\gamma+t-1}{(\gamma+t)^2} v \tag{67}$$

$$\leq \frac{v}{\gamma+t+1}. \tag{68}$$

Hence, we have proven that $\Delta^t \leq \frac{v}{\gamma+t}, \forall t$. Therefore, we have

$$
\begin{aligned}
\mathbb{E}\left[\left\|\boldsymbol{x}_s^t - \boldsymbol{x}_s^*\right\|^2\right] = \Delta^t &\leq \frac{v}{\gamma+t} = \frac{\max\left\{\frac{4B_1}{\mu^2} + \frac{8B_2}{\mu^3(\gamma+1)}, (\gamma+1)\mathbb{E}\left[\left\|\boldsymbol{x}_s^0 - \boldsymbol{x}_s^*\right\|^2\right]\right\}}{\gamma+t} \\
&\leq \frac{16N\sum_{n=1}^N a_n^2\left(2\sigma_n^2 + G^2\right)}{\mu^2(\gamma+t)} + \frac{1536S\sum_{n=1}^N a_n\left(2\sigma_n^2 + G^2\right)}{\mu^3(\gamma+t)(\gamma+1)} + \frac{(\gamma+1)\mathbb{E}\left[\left\|\boldsymbol{x}_s^0 - \boldsymbol{x}_s^*\right\|^2\right]}{\gamma+t}.
\end{aligned}
\tag{69}
$$

### D.1.2   One-round Parallel Update for Client-Side Models

Under Assumptions C.1 and C.2, if $\eta^t \leq \frac{1}{2S\tau}$, in round $t$, Lemma D.1 gives

$$
\begin{aligned}
&\mathbb{E}\left[\left\|\boldsymbol{x}_c^{t+1} - \boldsymbol{x}_c^*\right\|^2\right] \\
&\leq \left(1 - \frac{\eta^t\tau\mu}{2}\right)\mathbb{E}\left[\left\|\boldsymbol{x}_c^t - \boldsymbol{x}_c^*\right\|^2\right] - 2\eta^t\tau\mathbb{E}\left[f\left(\boldsymbol{x}^t\right) - f\left(\boldsymbol{x}^*\right)\right] \\
&\quad + \left(\eta^t\right)^2(\tau)^2 N\sum_{n=1}^N a_n^2\left(2\sigma_n^2 + G^2\right) + 24S(\tau)^3\left(\eta^t\right)^3\sum_{n=1}^N a_n\left(2\sigma_n^2 + G^2\right).
\end{aligned}
\tag{70}
$$

Let $\Delta^{t+1} \triangleq \mathbb{E}\left[\left\|\boldsymbol{x}_c^{t+1} - \boldsymbol{x}_c^*\right\|^2\right]$. We can rewrite (70) as:

$$
\Delta^{t+1} \leq \left(1 - \frac{\eta^t\tau\mu}{2}\right)\Delta^t + \frac{\left(\eta^t\right)^2(\tau)^2}{4}B_1 + \frac{\left(\eta^t\right)^3(\tau)^3}{8}B_2.
\tag{71}
$$

where $B_1 := 4N\sum_{n=1}^N a_n^2\left(2\sigma_n^2 + G^2\right)$ and $B_2 := 192S\sum_{n=1}^N a_n\left(2\sigma_n^2 + G^2\right)$.

Consider a diminishing stepsize $\eta^t = \frac{2\beta}{\tau(\gamma+t)}$, i.e, $\frac{\eta^t\tau}{2} = \frac{\beta}{\gamma+t}$, where $\beta = \frac{2}{\mu}, \gamma = \frac{8S}{\mu} - 1$. It is easy to show that $\eta^t \leq \frac{1}{2S\tau}$ for all $t$. For $v = \max\left\{\frac{4B_1}{\mu^2} + \frac{8B_2}{\mu^3(\gamma+1)}, (\gamma+1)\Delta^0\right\}$, we can prove that $\Delta^t \leq \frac{v}{\gamma+t}, \forall t$. Therefore, we have

$$
\begin{aligned}
\mathbb{E}\left[\left\|\boldsymbol{x}_c^t - \boldsymbol{x}_c^*\right\|^2\right] = \Delta^t &\leq \frac{v}{\gamma+t} = \frac{\max\left\{\frac{4B_1}{\mu^2} + \frac{8B_2}{\mu^3(\gamma+1)}, (\gamma+1)\mathbb{E}\left[\left\|\boldsymbol{x}_c^0 - \boldsymbol{x}_c^*\right\|^2\right]\right\}}{\gamma+t} \\
&\leq \frac{16N\sum_{n=1}^N a_n^2\left(2\sigma_n^2 + G^2\right)}{\mu^2(\gamma+t)} + \frac{1536S\sum_{n=1}^N a_n\left(2\sigma_n^2 + G^2\right)}{\mu^3(\gamma+t)(\gamma+1)} + \frac{(\gamma+1)\mathbb{E}\left[\left\|\boldsymbol{x}_c^0 - \boldsymbol{x}_c^*\right\|^2\right]}{\gamma+t}.
\end{aligned}
\tag{72}
$$

### D.1.3   Superposition of M-Server and Clients

We merge the M-server-side and client-side models in (69) and (72) using Proposition 3.5 by setting $\eta^t \leq \frac{1}{2S\max\{\tau,\tilde{\tau}\}}$. We have

$$
\begin{aligned}
&\mathbb{E}\left[f(\boldsymbol{x}^T)\right] - f(\boldsymbol{x}^*) \\
&\leq \frac{S}{2}\left(\mathbb{E}\|\boldsymbol{x}_s^T - \boldsymbol{x}_s^*\|^2 + \mathbb{E}\|\boldsymbol{x}_c^T - \boldsymbol{x}_c^*\|^2\right) \\
&\leq \frac{8SN\sum_{n=1}^N a_n^2\left(2\sigma_n^2 + G^2\right)}{\mu^2(\gamma+T)} + \frac{768S^2\sum_{n=1}^N a_n\left(2\sigma_n^2 + G^2\right)}{\mu^3(\gamma+T)(\gamma+1)} + \frac{S(\gamma+1)\mathbb{E}\left[\left\|\boldsymbol{x}^0 - \boldsymbol{x}^*\right\|^2\right]}{2(\gamma+T)}.
\end{aligned}
\tag{73}
$$

## D.2 General convex case for SFL-V1

### D.2.1 One-round Parallel Update for M-Server-Side Model

By Lemma D.1 with $\mu = 0$ and $\eta^t \leq \frac{1}{2S\tilde{\tau}}$, we have

$$
\mathbb{E}\left[\left\|\boldsymbol{x}_s^{t+1} - \boldsymbol{x}_s^*\right\|^2\right]
$$
$$
\leq \mathbb{E}\left[\left\|\boldsymbol{x}_s^t - \boldsymbol{x}_s^*\right\|^2\right] - 2\eta^t\tilde{\tau}\mathbb{E}\left[f\left(\boldsymbol{x}^t\right) - f\left(\boldsymbol{x}^*\right)\right]
$$
$$
+ \left(\eta^t\right)^2 \tilde{\tau}^2 N \sum_{n=1}^N a_n^2 \left(2\sigma_n^2 + G^2\right) + 24S\tilde{\tau}^3 \left(\eta^t\right)^3 \sum_{n=1}^N a_n \left(2\sigma_n^2 + G^2\right). \tag{74}
$$

### D.2.2 One-round Parallel Update for Client-Side Models

By Lemma D.1 with $\mu = 0$ and $\eta^t \leq \frac{1}{2S\tau}$, we have

$$
\mathbb{E}\left[\left\|\boldsymbol{x}_c^{t+1} - \boldsymbol{x}_c^*\right\|^2\right]
$$
$$
\leq \mathbb{E}\left[\left\|\boldsymbol{x}_c^t - \boldsymbol{x}_c^*\right\|^2\right] - 2\eta^t\tau\mathbb{E}\left[f\left(\boldsymbol{x}^t\right) - f\left(\boldsymbol{x}^*\right)\right]
$$
$$
+ \left(\eta^t\right)^2 \tau^2 N \sum_{n=1}^N a_n^2 \left(2\sigma_n^2 + G^2\right) + 24S\tau^3 \left(\eta^t\right)^3 \sum_{n=1}^N a_n \left(2\sigma_n^2 + G^2\right). \tag{75}
$$

### D.2.3 Superposition of M-Server and Clients

We merge the M-server-side and client-side models in (74) and (75) as follows

$$
\mathbb{E}\left[\left\|\boldsymbol{x}^{t+1} - \boldsymbol{x}^*\right\|^2\right] \leq \mathbb{E}\left[\left\|\boldsymbol{x}_s^{t+1} - \boldsymbol{x}_s^*\right\|^2\right] + \mathbb{E}\left[\left\|\boldsymbol{x}_c^{t+1} - \boldsymbol{x}_c^*\right\|^2\right],
$$
$$
\leq \mathbb{E}\left[\left\|\boldsymbol{x}_s^t - \boldsymbol{x}_s^*\right\|^2\right] - 2\eta^t\tilde{\tau}\mathbb{E}\left[f\left(\boldsymbol{x}^t\right) - f\left(\boldsymbol{x}^*\right)\right]
$$
$$
+ \left(\eta^t\right)^2 \tilde{\tau}^2 N \sum_{n=1}^N a_n^2 \left(2\sigma_n^2 + G^2\right) + 24S\tilde{\tau}^3 \left(\eta^t\right)^3 \sum_{n=1}^N a_n \left(2\sigma_n^2 + G^2\right)
$$
$$
+ \mathbb{E}\left[\left\|\boldsymbol{x}_c^t - \boldsymbol{x}_c^*\right\|^2\right] - 2\eta^t\tau\mathbb{E}\left[f\left(\boldsymbol{x}^t\right) - f\left(\boldsymbol{x}^*\right)\right]
$$
$$
+ \left(\eta^t\right)^2 \tau^2 N \sum_{n=1}^N a_n^2 \left(2\sigma_n^2 + G^2\right) + 24S\tau^3 \left(\eta^t\right)^3 \sum_{n=1}^N a_n \left(2\sigma_n^2 + G^2\right)
$$
$$
= \mathbb{E}\left[\left\|\boldsymbol{x}^t - \boldsymbol{x}^*\right\|^2\right] - 4\eta^t \min\{\tau, \tilde{\tau}\} \mathbb{E}\left[f\left(\boldsymbol{x}^t\right) - f\left(\boldsymbol{x}^*\right)\right]
$$
$$
+ \left(\eta^t\right)^2 N \sum_{n=1}^N a_n^2(\tilde{\tau}^2 + \tau^2)\left(2\sigma_n^2 + G^2\right) + 24S\left(\eta^t\right)^3 \sum_{n=1}^N a_n(\tilde{\tau}^3 + \tau^3)\left(2\sigma_n^2 + G^2\right). \tag{76}
$$

Then, we can obtain the relation between $\mathbb{E}\left[\left\|\boldsymbol{x}^{t+1} - \boldsymbol{x}^*\right\|^2\right]$ and $\mathbb{E}\left[\left\|\boldsymbol{x}^t - \boldsymbol{x}^*\right\|^2\right]$, which is related to $\mathbb{E}\left[f\left(\boldsymbol{x}^t\right) - f\left(\boldsymbol{x}^*\right)\right]$. Applying Lemma 8 in [17] and let $\tau_{\min} := \min\{\tilde{\tau}, \tau\}$ and $\eta^t \leq \frac{1}{2S\max\{\tau, \tilde{\tau}\}}$, we obtain the performance bound as

$$
\mathbb{E}\left[f\left(\boldsymbol{x}^T\right)\right] - f\left(\boldsymbol{x}^*\right)
$$
$$
\leq \frac{1}{2}\left(\frac{\tilde{\tau}^2 + \tau^2}{\tau_{\min}^2} N \sum_{n=1}^N a_n^2 \left(2\sigma_n^2 + G^2\right)\right)^{\frac{1}{2}} \left(\frac{\left\|\boldsymbol{x}^0 - \boldsymbol{x}^*\right\|^2}{T+1}\right)^{\frac{1}{2}}
$$
$$
+ \frac{1}{2}\left(\frac{\tilde{\tau}^2 + \tau^2}{\tau_{\min}^2} 24S \sum_{n=1}^N a_n \left(2\sigma_n^2 + G^2\right)\right)^{\frac{1}{3}} \left(\frac{\left\|\boldsymbol{x}^0 - \boldsymbol{x}^*\right\|^2}{T+1}\right)^{\frac{1}{3}} + \frac{S\left\|\boldsymbol{x}^0 - \boldsymbol{x}^*\right\|^2}{2(T+1)}. \tag{77}
$$

### D.3 Non-convex case for SFL-V1

#### D.3.1 One-round Parallel Update for M-Server-Side Model

For the server, we have

$$
\mathbb{E}\left[\left\langle \nabla_{\boldsymbol{x}_s} f\left(\boldsymbol{x}^t\right), \boldsymbol{x}_s^{t+1} - \boldsymbol{x}_s^t \right\rangle\right]
$$
$$
\leq \mathbb{E}\left[\left\langle \nabla_{\boldsymbol{x}_s} f\left(\boldsymbol{x}^t\right), \boldsymbol{x}_s^{t+1} - \boldsymbol{x}_s^t + \eta^t \tilde{\tau} \nabla_{\boldsymbol{x}_s} f\left(\boldsymbol{x}^t\right) - \eta^t \tilde{\tau} \nabla_{\boldsymbol{x}_s} f\left(\boldsymbol{x}^t\right) \right\rangle\right]
$$
$$
\leq \mathbb{E}\left[\left\langle \nabla_{\boldsymbol{x}_s} f\left(\boldsymbol{x}^t\right), \boldsymbol{x}_s^{t+1} - \boldsymbol{x}_s^t + \eta^t \tilde{\tau} \nabla_{\boldsymbol{x}_s} f\left(\boldsymbol{x}^t\right) \right\rangle - \left\langle \nabla_{\boldsymbol{x}_s} f\left(\boldsymbol{x}_s^t\right), \eta^t \tilde{\tau} \nabla_{\boldsymbol{x}_s} f\left(\boldsymbol{x}^t\right) \right\rangle\right]
$$
$$
\leq \left\langle \nabla_{\boldsymbol{x}_s} f\left(\boldsymbol{x}^t\right), \mathbb{E}\left[-\eta^t \sum_{n=1}^{N} \sum_{i=0}^{\tilde{\tau}-1} a_n \boldsymbol{g}_{s,n}^{t,i}\right] + \eta^t \tilde{\tau} \nabla_{\boldsymbol{x}_s} f\left(\boldsymbol{x}^t\right) \right\rangle - \eta^t \tilde{\tau} \left\|\nabla_{\boldsymbol{x}_s} f\left(\boldsymbol{x}^t\right)\right\|^2
$$
$$
\leq \left\langle \nabla_{\boldsymbol{x}_s} f\left(\boldsymbol{x}^t\right), \mathbb{E}\left[-\eta^t \sum_{n=1}^{N} \sum_{i=0}^{\tilde{\tau}-1} a_n \nabla_{\boldsymbol{x}_s} F_n\left(\left\{\boldsymbol{x}_{c,n}^{t,i}, \boldsymbol{x}_{s,n}^{t,i}\right\}\right)\right] + \eta^t \tilde{\tau} \nabla_{\boldsymbol{x}_s} f\left(\boldsymbol{x}^t\right) \right\rangle - \eta^t \tilde{\tau} \left\|\nabla_{\boldsymbol{x}_s} f\left(\boldsymbol{x}^t\right)\right\|^2
$$
$$
\leq \left\langle \nabla_{\boldsymbol{x}_s} f\left(\boldsymbol{x}^t\right), \mathbb{E}\left[-\eta^t \sum_{n=1}^{N} \sum_{i=0}^{\tilde{\tau}-1} a_n \nabla_{\boldsymbol{x}_s} F_n\left(\left\{\boldsymbol{x}_{c,n}^{t,i}, \boldsymbol{x}_{s,n}^{t,i}\right\}\right) + \eta^t \sum_{n=1}^{N} \sum_{i=0}^{\tilde{\tau}-1} a_n \nabla_{\boldsymbol{x}_s} F_n\left(\boldsymbol{x}^t\right)\right] \right\rangle
$$
$$
- \eta^t \tilde{\tau} \left\|\nabla_{\boldsymbol{x}_s} f\left(\boldsymbol{x}^t\right)\right\|^2
$$
$$
\leq \eta^t \tilde{\tau} \left\langle \nabla_{\boldsymbol{x}_s} f\left(\boldsymbol{x}^t\right), \mathbb{E}\left[-\frac{1}{\tilde{\tau}} \sum_{n=1}^{N} \sum_{i=0}^{\tilde{\tau}-1} a_n \nabla_{\boldsymbol{x}_s} F_n\left(\left\{\boldsymbol{x}_{c,n}^{t,i}, \boldsymbol{x}_{s,n}^{t,i}\right\}\right) + \frac{1}{\tilde{\tau}} \sum_{n=1}^{N} \sum_{i=0}^{\tilde{\tau}-1} a_n \nabla_{\boldsymbol{x}_s} F_n\left(\boldsymbol{x}^t\right)\right] \right\rangle
$$
$$
- \eta^t \tilde{\tau} \left\|\nabla_{\boldsymbol{x}_s} f\left(\boldsymbol{x}^t\right)\right\|^2
$$
$$
\leq \frac{\eta^t \tilde{\tau}}{2} \left\|\nabla_{\boldsymbol{x}_s} f\left(\boldsymbol{x}^t\right)\right\|^2 + \frac{\eta^t}{2\tilde{\tau}} \mathbb{E}\left[\left\|\sum_{n=1}^{N} \sum_{i=0}^{\tilde{\tau}-1} a_n \nabla_{\boldsymbol{x}_s} F_n\left(\left\{\boldsymbol{x}_{c,n}^{t,i}, \boldsymbol{x}_{s,n}^{t,i}\right\}\right) - \sum_{n=1}^{N} \sum_{i=0}^{\tilde{\tau}-1} a_n \nabla_{\boldsymbol{x}_s} F_n\left(\boldsymbol{x}^t\right)\right\|^2\right]
$$
$$
- \eta^t \tilde{\tau} \left\|\nabla_{\boldsymbol{x}_s} f\left(\boldsymbol{x}^t\right)\right\|^2
$$
$$
\leq -\frac{\eta^t \tilde{\tau}}{2} \left\|\nabla_{\boldsymbol{x}_s} f\left(\boldsymbol{x}^t\right)\right\|^2 + \frac{\eta^t}{2\tilde{\tau}} \mathbb{E}\left[\left\|\sum_{n=1}^{N} a_n \sum_{i=0}^{\tilde{\tau}-1} \left(\nabla_{\boldsymbol{x}_s} F_n\left(\left\{\boldsymbol{x}_{c,n}^{t,i}, \boldsymbol{x}_{s,n}^{t,i}\right\}\right) - \nabla_{\boldsymbol{x}_s} F_n\left(\boldsymbol{x}^t\right)\right)\right\|^2\right]
$$
$$
\leq -\frac{\eta^t \tilde{\tau}}{2} \left\|\nabla_{\boldsymbol{x}_s} f\left(\boldsymbol{x}^t\right)\right\|^2 + \frac{N\eta^t}{2\tilde{\tau}} \sum_{n=1}^{N} a_n^2 \mathbb{E}\left[\left\|\sum_{i=0}^{\tilde{\tau}-1} \left(\nabla_{\boldsymbol{x}_s} F_n\left(\left\{\boldsymbol{x}_{c,n}^{t,i}, \boldsymbol{x}_{s,n}^{t,i}\right\}\right) - \nabla_{\boldsymbol{x}_s} F_n\left(\boldsymbol{x}^t\right)\right)\right\|^2\right]
$$
$$
\leq -\frac{\eta^t \tilde{\tau}}{2} \left\|\nabla_{\boldsymbol{x}_s} f\left(\boldsymbol{x}^t\right)\right\|^2 + \frac{N\eta^t S^2}{2} \sum_{n=1}^{N} a_n^2 \sum_{i=0}^{\tilde{\tau}-1} \mathbb{E}\left[\left\|\boldsymbol{x}_{s,n}^{t,i} - \boldsymbol{x}_s^t\right\|^2\right], \tag{78}
$$

where we apply Assumption C.1, $\nabla_{\boldsymbol{x}_s} f\left(\boldsymbol{x}^t\right) = \sum_{n=1}^{N} a_n \nabla_{\boldsymbol{x}_s} F_n\left(\boldsymbol{x}^t\right)$, and $\langle a, b \rangle \leq \frac{a^2 + b^2}{2}$.

By Lemma C.6 with $\eta^t \leq \frac{1}{\sqrt{8}S\tilde{\tau}}$, we have

$$
\sum_{i=0}^{\tilde{\tau}-1} \mathbb{E}\left[\left\|\boldsymbol{x}_{s,n}^{t,i} - \boldsymbol{x}_s^t\right\|^2\right] \leq 2\tilde{\tau}^2 \left(8\tilde{\tau} \left(\eta^t\right)^2 \sigma_n^2 + 8\tilde{\tau} \left(\eta^t\right)^2 \epsilon^2 + 8\tilde{\tau} \left(\eta^t\right)^2 \left\|\nabla_{\boldsymbol{x}_s} f\left(\boldsymbol{x}_s^t\right)\right\|^2\right). \tag{79}
$$

Thus, (78) becomes

$$
\mathbb{E}\left[\left\langle \nabla_{\boldsymbol{x}_s} f\left(\boldsymbol{x}^t\right), \boldsymbol{x}_s^{t+1} - \boldsymbol{x}_s^t \right\rangle\right]
$$
$$
\leq -\frac{\eta^t \tilde{\tau}}{2} \left\|\nabla_{\boldsymbol{x}_s} f\left(\boldsymbol{x}^t\right)\right\|^2 + \frac{N\eta^t S^2}{2} \sum_{n=1}^{N} a_n^2 2\tilde{\tau}^2 \left(8\tilde{\tau} \left(\eta^t\right)^2 \sigma_n^2 + 8\tilde{\tau} \left(\eta^t\right)^2 \epsilon^2 + 8\tilde{\tau} \left(\eta^t\right)^2 \left\|\nabla_{\boldsymbol{x}_s} f\left(\boldsymbol{x}_s^t\right)\right\|^2\right)
$$
$$
\leq \left(-\frac{\eta^t \tilde{\tau}}{2} + 8N \left(\eta^t\right)^3 \tilde{\tau}^3 S^2 \sum_{n=1}^{N} a_n^2\right) \left\|\nabla_{\boldsymbol{x}_s} f\left(\boldsymbol{x}^t\right)\right\|^2 + 8N\eta^t S^2 \tilde{\tau}^3 \sum_{n=1}^{N} a_n^2 \left(\eta^t\right)^2 \left(\sigma_n^2 + \epsilon^2\right). \tag{80}
$$

Furthermore, we have

$$\frac{S}{2}\mathbb{E}\left[\left\|\boldsymbol{x}_s^{t+1} - \boldsymbol{x}_s^t\right\|^2\right]$$

$$= \frac{SN\left(\eta^t\right)^2}{2}\sum_{n=1}^N \mathbb{E}\left[\left\|\sum_{i=0}^{\tilde{\tau}-1} a_n \boldsymbol{g}_{s,n}^{t,i}\right\|^2\right]$$

$$\leq \frac{SN\left(\eta^t\right)^2}{2}\sum_{n=1}^N a_n^2 \mathbb{E}\left[\left\|\sum_{i=0}^{\tilde{\tau}-1} \boldsymbol{g}_{s,n}^{t,i}\right\|^2\right]$$

$$\leq \frac{SN\left(\eta^t\right)^2 \tilde{\tau}}{2}\sum_{n=1}^N a_n^2 \sum_{i=0}^{\tilde{\tau}-1} \mathbb{E}\left[\left\|\boldsymbol{g}_{s,n}^{t,i}\right\|^2\right]$$

$$\leq \frac{SN\left(\eta^t\right)^2 \tilde{\tau}}{2}\sum_{n=1}^N a_n^2 \sum_{i=0}^{\tilde{\tau}-1} \mathbb{E}\left[\left\|\boldsymbol{g}_{s,n}^{t,i} - \boldsymbol{g}_{s,n}^t + \boldsymbol{g}_{s,n}^t\right\|^2\right]$$

$$\leq \frac{SN\left(\eta^t\right)^2 \tilde{\tau}}{2}\sum_{n=1}^N a_n^2 \sum_{i=0}^{\tilde{\tau}-1} \left(\mathbb{E}\left[\left\|\boldsymbol{g}_{s,n}^{t,i} - \boldsymbol{g}_{s,n}^t\right\|^2\right] + \mathbb{E}\left[\left\|\boldsymbol{g}_{s,n}^t\right\|^2\right]\right)$$

$$\leq \frac{SN\left(\eta^t\right)^2 \tilde{\tau}}{2}\sum_{n=1}^N a_n^2 \sum_{i=0}^{\tilde{\tau}-1} \left(\mathbb{E}\left[\left\|\boldsymbol{g}_{s,n}^{t,i} - \boldsymbol{g}_{s,n}^t\right\|^2\right] + \mathbb{E}\left[\left\|\nabla_{\boldsymbol{x}_s} F_n\left(\boldsymbol{x}^t\right)\right\|^2 + \sigma_n^2\right]\right), \quad (81)$$

where the last line uses Assumption C.1 and $\mathbb{E}\left[\|\mathbf{z}\|^2\right] = \|\mathbb{E}[\mathbf{z}]\|^2 + \mathbb{E}[\|\mathbf{z} - \mathbb{E}[\mathbf{z}]\|^2]$ for any random variable $\mathbf{z}$.

By Lemma C.7 with $\eta^t \leq \frac{1}{2S\tilde{\tau}}$, we have

$$\sum_{i=0}^{\tilde{\tau}-1} \mathbb{E}\left[\left\|\boldsymbol{g}_{s,n}^{t,i} - \boldsymbol{g}_{s,n}^t\right\|^2\right] \leq 8\tilde{\tau}^3 \left(\eta^t\right)^2 S^2 \left(\left\|\nabla_{\boldsymbol{x}_s} F_n\left(\boldsymbol{x}^t\right)\right\|^2 + \sigma_n^2\right). \quad (82)$$

Thus, (81) becomes

$$\frac{S}{2}\mathbb{E}\left[\left\|\boldsymbol{x}_s^{t+1} - \boldsymbol{x}_s^t\right\|^2\right]$$

$$\leq \frac{SN\left(\eta^t\right)^2 \tilde{\tau}}{2}\sum_{n=1}^N a_n^2 \left(8\tilde{\tau}^3 \left(\eta^t\right)^2 S^2 \left(\left\|\nabla_{\boldsymbol{x}_s} F_n\left(\boldsymbol{x}^t\right)\right\|^2 + \sigma_n^2\right) + \tilde{\tau}\mathbb{E}\left[\left\|\nabla_{\boldsymbol{x}_s} F_n\left(\boldsymbol{x}^t\right)\right\|^2 + \sigma_n^2\right]\right)$$

$$\leq \frac{SN\left(\eta^t\right)^2 \tilde{\tau}}{2}\sum_{n=1}^N a_n^2 \left(\tilde{\tau} + 8\tilde{\tau}^3 \left(\eta^t\right)^2 S^2\right)\left(\left\|\nabla_{\boldsymbol{x}_s} F_n\left(\boldsymbol{x}^t\right)\right\|^2 + \sigma_n^2\right)$$

$$\leq \frac{SN\left(\eta^t\right)^2 \tilde{\tau}}{2}\sum_{n=1}^N a_n^2 \left(\tilde{\tau} + 8\tilde{\tau}^3 \left(\eta^t\right)^2 S^2\right)\left(\left\|\nabla_{\boldsymbol{x}_s} F_n\left(\boldsymbol{x}^t\right) - \nabla_{\boldsymbol{x}_s} f\left(\boldsymbol{x}^t\right) + \nabla_{\boldsymbol{x}_s} f\left(\boldsymbol{x}^t\right)\right\|^2 + \sigma_n^2\right)$$

$$\leq \frac{SN\left(\eta^t\right)^2 \tilde{\tau}}{2}\sum_{n=1}^N a_n^2 \left(\tilde{\tau} + 8\tilde{\tau}^3 \left(\eta^t\right)^2 S^2\right)\left(2\left\|\nabla_{\boldsymbol{x}_s} f\left(\boldsymbol{x}^t\right)\right\|^2 + 2\epsilon^2 + \sigma_n^2\right). \quad (83)$$

### D.3.2  One-round Parallel Update for Client-Side Models

The analysis of the client-side model update is similar to the server. Thus, we have

$$\mathbb{E}\left[\left\langle \nabla_{\boldsymbol{x}_c} f\left(\boldsymbol{x}^t\right), \boldsymbol{x}_c^{t+1} - \boldsymbol{x}_c^t\right\rangle\right]$$

$$\leq \left(-\frac{\eta^t \tau}{2} + 8N\left(\eta^t\right)^3 \tau^3 S^2 \sum_{n=1}^N a_n^2\right)\left\|\nabla_{\boldsymbol{x}_c} f\left(\boldsymbol{x}^t\right)\right\|^2 + 8N\eta^t S^2 \tau^3 \sum_{n=1}^N a_n^2 \left(\eta^t\right)^2 \left(\sigma_n^2 + \epsilon^2\right). \quad (84)$$

For $\eta^t \leq \frac{1}{2S\tau}$,

$$\frac{S}{2}\mathbb{E}\left[\left\|\boldsymbol{x}_c^{t+1}-\boldsymbol{x}_c^t\right\|^2\right]$$
$$\leq \frac{SN\left(\eta^t\right)^2\tau}{2}\sum_{n=1}^N a_n^2\left(\tau+8\tau^3\left(\eta^t\right)^2 S^2\right)\left(2\left\|\nabla_{\boldsymbol{x}_c}f\left(\boldsymbol{x}^t\right)\right\|^2+2\epsilon^2+\sigma_n^2\right). \tag{85}$$

### D.3.3 Superposition of M-Server and Clients

Applying (80), (83), (85) and (84) into (36) in Proposition C.4 and define $\tau_{\min}\triangleq\min\{\tau,\tilde{\tau}\}$, $\tau_{\max}\triangleq\max\{\tau,\tilde{\tau}\}$, we have

$$\mathbb{E}\left[f\left(\boldsymbol{x}^{t+1}\right)\right]-f\left(\boldsymbol{x}^t\right)$$

$$\leq \mathbb{E}\left[\left\langle\nabla_{\boldsymbol{x}_c}f\left(\boldsymbol{x}^t\right),\boldsymbol{x}_c^{t+1}-\boldsymbol{x}_c^t\right\rangle\right]+\frac{S}{2}\mathbb{E}\left[\left\|\boldsymbol{x}_c^{t+1}-\boldsymbol{x}_c^t\right\|^2\right]+\mathbb{E}\left[\left\langle\nabla_{\boldsymbol{x}_s}f\left(\boldsymbol{x}^t\right),\boldsymbol{x}_s^{t+1}-\boldsymbol{x}_s^t\right\rangle\right]$$

$$+\frac{S}{2}\mathbb{E}\left[\left\|\boldsymbol{x}_s^{t+1}-\boldsymbol{x}_s^t\right\|^2\right]$$

$$\leq \left(-\frac{\eta^t\min\{\tau,\tilde{\tau}\}}{2}+8N\left(\eta^t\right)^3\left(\max\{\tau,\tilde{\tau}\}\right)^3 S^2\sum_{n=1}^N a_n^2\right)\left\|\nabla_{\boldsymbol{x}}f\left(\boldsymbol{x}^t\right)\right\|^2$$

$$+8N\eta^t S^2\left(\tau^3+\tilde{\tau}^3\right)\sum_{n=1}^N\left(\eta^t\right)^2 a_n^2\left(\sigma_n^2+\epsilon^2\right)$$

$$+\frac{SN\left(\eta^t\right)^2\max\{\tau,\tilde{\tau}\}}{2}\sum_{n=1}^N a_n^2\left(\max\{\tau,\tilde{\tau}\}+8\left(\max\{\tau,\tilde{\tau}\}\right)^3\left(\eta^t\right)^2 S^2\right)2\left\|\nabla_{\boldsymbol{x}}f\left(\boldsymbol{x}^t\right)\right\|^2$$

$$+\frac{SN\left(\eta^t\right)^2\tau}{2}\sum_{n=1}^N a_n^2\left(\tau+8\tau^3\left(\eta^t\right)^2 S^2\right)\left(2\epsilon^2+\sigma_n^2\right)$$

$$+\frac{SN\left(\eta^t\right)^2\tilde{\tau}}{2}\sum_{n=1}^N a_n^2\left(\tilde{\tau}+8\tilde{\tau}^3\left(\eta^t\right)^2 S^2\right)\left(2\epsilon^2+\sigma_n^2\right) \tag{86}$$

$$\leq \left(-\frac{\eta^t\tau_{\min}}{2}+8N\left(\eta^t\right)^3 S^2\tau_{\max}^3\sum_{n=1}^N a_n^2+SN\left(\eta^t\right)^2\tau_{\max}\sum_{n=1}^N a_n^2\left(\tau_{\max}+8\tau_{\max}^3\left(\eta^t\right)^2 S^2\right)\right)\left\|\nabla_{\boldsymbol{x}}f\left(\boldsymbol{x}^t\right)\right\|^2$$

$$+8N\eta^t S^2\left(\tau^3+\tilde{\tau}^3\right)\sum_{n=1}^N a_n^2\left(\eta^t\right)^2\left(\sigma_n^2+\epsilon^2\right)$$

$$+\frac{1}{2}SN\left(\eta^t\right)^2\tau\left(\tau+8\tau^3\left(\eta^t\right)^2 S^2\right)\sum_{n=1}^N a_n^2\left(2\epsilon^2+\sigma_n^2\right)+\frac{1}{2}SN\left(\eta^t\right)^2\tilde{\tau}\left(\tilde{\tau}+8\tilde{\tau}^3\left(\eta^t\right)^2 S^2\right)\sum_{n=1}^N a_n^2\left(2\epsilon^2+\sigma_n^2\right)$$

$$\leq \left(-\frac{\eta^t\tau_{\min}}{2}+SN\left(\eta^t\right)^2\tau_{\max}^2\sum_{n=1}^N a_n^2+8N\left(\eta^t\right)^3 vS^2\tau_{\max}^3\sum_{n=1}^N a_n^2+8S^3N\left(\eta^t\right)^4\tau_{\max}^4\sum_{n=1}^N a_n^2\right)\left\|\nabla_{\boldsymbol{x}}f\left(\boldsymbol{x}^t\right)\right\|^2$$

$$+8N\left(\eta^t\right)^3 S^2\tau^3\sum_{n=1}^N a_n^2\sigma_n^2+8N\left(\eta^t\right)^3 S^2\tau^3\epsilon^2\sum_{n=1}^N a_n^2$$

$$+SN\left(\eta^t\right)^2\tau^2\epsilon^2\sum_{n=1}^N a_n^2+\frac{1}{2}SN\left(\eta^t\right)^2\tilde{\tau}^2\sum_{n=1}^N a_n^2\sigma_n^2+8NS^3\left(\eta^t\right)^4\tau^4\epsilon^2\sum_{n=1}^N a_n^2+4NS^3\left(\eta^t\right)^4\tau^4\sum_{n=1}^N a_n^2\sigma_n^2$$

$$+8N\left(\eta^t\right)^3 S^2\tilde{\tau}^3\sum_{n=1}^N a_n^2\sigma_n^2+8N\left(\eta^t\right)^3 S^2\tilde{\tau}^3\epsilon^2\sum_{n=1}^N a_n^2$$

$$+SN\left(\eta^t\right)^2\tilde{\tau}^2\epsilon^2\sum_{n=1}^N a_n^2+\frac{1}{2}SN\left(\eta^t\right)^2\tilde{\tau}^2\sum_{n=1}^N a_n^2\sigma_n^2+8NS^3\left(\eta^t\right)^4\tilde{\tau}^4\epsilon^2\sum_{n=1}^N a_n^2+4NS^3\left(\eta^t\right)^4\tilde{\tau}^4\sum_{n=1}^N a_n^2\sigma_n^2$$

$$\leq -\frac{\eta^t \tau_{\min}}{2}\left(1 - 2SN\eta^t \frac{\tau_{\max}^2}{\tau_{\min}}\sum_{n=1}^{N} a_n^2 \left(1 + 8S\eta^t\tau + 8S^2\left(\eta^t\right)^2\tau_{\max}^2\right)\right)\left\|\nabla_{\boldsymbol{x}}f\left(\boldsymbol{x}^t\right)\right\|^2$$

$$+\left(\frac{1}{2}NS\left(\eta^t\right)^2\tau^2 + 8N\left(\eta^t\right)^3 S^2\tau^3 + 4NS^3\left(\eta^t\right)^4\tau^4\right)\sum_{n=1}^{N} a_n^2\sigma_n^2$$

$$+\left(NS\left(\eta^t\right)^2\tau^2 + 8N\left(\eta^t\right)^3 S^2\tau^3 + 8NSL^3\left(\eta^t\right)^4\tau^4\right)\sum_{n=1}^{N} a_n^2\epsilon^2$$

$$+\left(\frac{1}{2}NS\left(\eta^t\right)^2\tilde{\tau}^2 + 8N\left(\eta^t\right)^3 S^2\tilde{\tau}^3 + 4NS^3\left(\eta^t\right)^4\tilde{\tau}^4\right)\sum_{n=1}^{N} a_n^2\sigma_n^2$$

$$+\left(NS\left(\eta^t\right)^2\tau^2 + 8N\left(\eta^t\right)^3 S^2\tilde{\tau}^3 + 8NSL^3\left(\eta^t\right)^4\tilde{\tau}^4\right)\sum_{n=1}^{N} a_n^2\epsilon^2$$

$$\leq -\frac{\eta^t\tau_{\min}}{2}\left(1 - 2NS\eta^t\frac{\tau_{\max}^2}{\tau_{\min}}\sum_{n=1}^{N} a_n^2\left(1 + \frac{1}{2} + \frac{1}{32}\right)\right)\left\|\nabla_{\boldsymbol{x}}f\left(\boldsymbol{x}^t\right)\right\|^2$$

$$+NS\left(\eta^t\right)^2\tau^2\left(\frac{1}{2} + \frac{1}{2} + \frac{1}{64}\right)\sum_{n=1}^{N} a_n^2\sigma_n^2 + 2SN\left(\eta^t\right)^2\tau^2\left(\frac{1}{2} + \frac{1}{4} + \frac{1}{64}\right)\sum_{n=1}^{N} a_n^2\epsilon^2$$

$$+NS\left(\eta^t\right)^2\tilde{\tau}^2\left(\frac{1}{2} + \frac{1}{2} + \frac{1}{64}\right)\sum_{n=1}^{N} a_n^2\sigma_n^2 + 2SN\left(\eta^t\right)^2\tilde{\tau}^2\left(\frac{1}{2} + \frac{1}{4} + \frac{1}{64}\right)\sum_{n=1}^{N} a_n^2\epsilon^2$$

$$\leq -\frac{\eta^t\tau_{\min}}{2}\left(1 - 4NS\eta^t\frac{\tau_{\max}^2}{\tau_{\min}}\sum_{n=1}^{N} a_n^2\right)\left\|\nabla_{\boldsymbol{x}}f\left(\boldsymbol{x}^t\right)\right\|^2 + 2NS\left(\eta^t\right)^2\left(\tau^2 + \tilde{\tau}^2\right)\sum_{n=1}^{N} a_n^2\left(\sigma_n^2 + \epsilon^2\right)$$

$$\leq -\frac{\eta^t\tau_{\min}}{4}\left\|\nabla_{\boldsymbol{x}}f\left(\boldsymbol{x}^t\right)\right\|^2 + 2NS\left(\eta^t\right)^2\left(\tau^2 + \tilde{\tau}^2\right)\sum_{n=1}^{N} a_n^2\left(\sigma_n^2 + \epsilon^2\right),\tag{87}$$

where we first let $\eta^t \leq \frac{1}{16S\tau_{\max}}$ and then let $\eta^t \leq \frac{1}{8SN\frac{\tau_{\max}^2}{\tau_{\min}}\sum_{n=1}^{N} a_n^2}$. We also use $\left\|\nabla_{\boldsymbol{x}}f\left(\boldsymbol{x}^t\right)\right\|^2 = \left\|\nabla_{\boldsymbol{x}_c}f\left(\boldsymbol{x}^t\right)\right\|^2 + \left\|\nabla_{\boldsymbol{x}_s}f\left(\boldsymbol{x}^t\right)\right\|^2$.

Rearranging the above we have

$$\eta^t\left\|\nabla_{\boldsymbol{x}}f\left(\boldsymbol{x}^t\right)\right\|^2 \leq \frac{4}{\tau_{\min}}\left(f\left(\boldsymbol{x}^t\right) - \mathbb{E}\left[f\left(\boldsymbol{x}_s^{t+1}\right)\right]\right) + 8NS\left(\eta^t\right)^2\frac{\tau^2 + \tilde{\tau}^2}{\tau_{\min}}\sum_{n=1}^{N} a_n^2\left(\sigma_n^2 + \epsilon^2\right).\tag{88}$$

Taking expectation and averaging over all $t$, we have

$$\frac{1}{T}\sum_{t=0}^{T-1}\eta^t\mathbb{E}\left[\left\|\nabla_{\boldsymbol{x}}f\left(\boldsymbol{x}^t\right)\right\|^2\right] \leq \frac{4}{T\tau_{\min}}\left(f\left(\boldsymbol{x}_0\right) - f^*\right) + \frac{8NS\frac{\tau^2+\tilde{\tau}^2}{\tau_{\min}}}{T}\sum_{n=1}^{N} a_n^2\left(\sigma_n^2 + \epsilon^2\right)\sum_{t=0}^{T-1}\left(\eta^t\right)^2.\tag{89}$$

# E  Proof of Theorem 3.7

- In Sec. E.1, we prove the strongly convex case.
- In Sec. E.2, we prove the general convex case.
- In Sec. E.3, we prove the non-convex case.

## E.1  Strongly convex case for SFL-V2

### E.1.1  One-round Sequential Update for M-Server-Side Model

**Lemma E.1.** *Under Assumptions C.1 and C.2, if $\eta^t \leq \frac{1}{2S\tau}$, in round t, the M-server-side model evolves as*

$$
\mathbb{E}\left[\left\|\boldsymbol{x}_s^{t+1} - \boldsymbol{x}_s^*\right\|^2\right]
$$

$$
\leq \left(1 - \frac{N\eta^t\tau\mu}{2}\right)\mathbb{E}\left[\left\|\boldsymbol{x}_s^t - \boldsymbol{x}_s^*\right\|^2\right] - 2\eta^t\tau\mathbb{E}\left[f\left(\boldsymbol{x}^t\right) - f\left(\boldsymbol{x}^*\right)\right]
$$

$$
+ \left(\eta^t\right)^2\tau^2 N\sum_{n=1}^N\left(2\sigma_n^2 + G^2\right) + 24S\tau^3\left(\eta^t\right)^3\sum_{n=1}^N\left(2\sigma_n^2 + G^2\right). \tag{90}
$$

We prove Lemma E.1 as follows.

*Proof.* We use $\boldsymbol{x}_{s,n}^{t,i}$ as the M-server-side model when the M-server interacts with client $n$ for the $i$-th iteration of model training at round $t$. Using the (sequential) gradient update rule of $\boldsymbol{x}_s^{t+1} = \boldsymbol{x}_s^t - \eta^t\sum_{n=1}^N\sum_{i=0}^{\tau-1}\boldsymbol{g}_s^{t,i}\left(\left\{\boldsymbol{x}_{c,n}^{t,i},\boldsymbol{x}_{s,n}^{t,i}\right\}\right)$, we have

$$
\mathbb{E}\left[\left\|\boldsymbol{x}_s^{t+1} - \boldsymbol{x}_s^*\right\|^2\right]
$$

$$
= \mathbb{E}\left[\left\|\boldsymbol{x}_s^t - \eta^t\sum_{i=0}^{\tau-1}\boldsymbol{g}_s^{t,i} - \boldsymbol{x}_s^* - \eta^t\sum_{i=0}^{\tau-1}\nabla_{\boldsymbol{x}_s}f\left(\left\{\boldsymbol{x}_c^{t,i},\boldsymbol{x}_s^{t,i}\right\}\right) + \eta^t\sum_{i=0}^{\tau-1}\nabla_{\boldsymbol{x}_s}f\left(\left\{\boldsymbol{x}_c^{t,i},\boldsymbol{x}_s^{t,i}\right\}\right)\right\|^2\right]
$$

$$
= \mathbb{E}\left[\left\|\boldsymbol{x}_s^t - \boldsymbol{x}_s^* - \eta^t\sum_{i=0}^{\tau-1}\nabla_{\boldsymbol{x}_s}f\left(\left\{\boldsymbol{x}_c^{t,i},\boldsymbol{x}_s^{t,i}\right\}\right)\right\|^2\right]
$$

$$
+ 2\eta^t\mathbb{E}\left[\left\langle \boldsymbol{x}_s^t - \boldsymbol{x}_s^* - \eta^t\sum_{i=0}^{\tau-1}\nabla_{\boldsymbol{x}_s}f\left(\left\{\boldsymbol{x}_c^{t,i},\boldsymbol{x}_s^{t,i}\right\}\right), \sum_{i=0}^{\tau-1}\nabla_{\boldsymbol{x}_s}f\left(\left\{\boldsymbol{x}_c^{t,i},\boldsymbol{x}_s^{t,i}\right\}\right) - \sum_{i=0}^{\tau-1}\boldsymbol{g}_s^{t,i}\right\rangle\right]
$$

$$
+ \mathbb{E}\left[\left(\eta^t\right)^2\left\|\sum_{i=0}^{\tau-1}\boldsymbol{g}_s^{t,i} - \sum_{i=0}^{\tau-1}\nabla_{\boldsymbol{x}_s}f\left(\left\{\boldsymbol{x}_c^{t,i},\boldsymbol{x}_s^{t,i}\right\}\right)\right\|^2\right]
$$

$$
\leq \mathbb{E}\left[\left\|\boldsymbol{x}_s^t - \boldsymbol{x}_s^* - \eta^t\sum_{i=0}^{\tau-1}\nabla_{\boldsymbol{x}_s}f\left(\left\{\boldsymbol{x}_c^{t,i},\boldsymbol{x}_s^{t,i}\right\}\right)\right\|^2\right]
$$

$$
+ \left(\eta^t\right)^2\mathbb{E}\left[\left\|\sum_{n=1}^N\sum_{i=0}^{\tau-1}\boldsymbol{g}_{s,n}^{t,i} - \sum_{i=0}^{\tau-1}\nabla_{\boldsymbol{x}_s}f\left(\left\{\boldsymbol{x}_c^{t,i},\boldsymbol{x}_s^{t,i}\right\}\right)\right\|^2\right]. \tag{91}
$$

where the first equality is from $(a+b)^2 = a^2 + 2ab + b^2$ and the last inequality is due to $\mathbb{E}\left[\nabla_{\boldsymbol{x}_s}f\left(\left\{\boldsymbol{x}_c^{t,i},\boldsymbol{x}_s^{t,i}\right\}\right) - \boldsymbol{g}_s^{t,i}\right] = 0$.

The first part in (91) is

$$
\mathbb{E}\left[\left\|\boldsymbol{x}_s^t - \boldsymbol{x}_s^* - \eta^t\sum_{i=0}^{\tau-1}\nabla_{\boldsymbol{x}_s}f\left(\left\{\boldsymbol{x}_c^{t,i},\boldsymbol{x}_s^{t,i}\right\}\right)\right\|^2\right]
$$

$$\leq \mathbb{E}\left[\left\|\boldsymbol{x}_s^t - \boldsymbol{x}_s^*\right\|^2\right] + \left(\eta^t\right)^2 \tau N \sum_{n=1}^{N}\sum_{i=0}^{\tau-1}\mathbb{E}\left[\left\|\nabla_{\boldsymbol{x}_s}F_n\left(\left\{\boldsymbol{x}_{c,n}^{t,i},\boldsymbol{x}_{s,n}^{t,i}\right\}\right)\right\|^2\right]$$

$$- 2\eta^t\mathbb{E}\left[\sum_{n=1}^{N}\sum_{i=0}^{\tau-1}\left\langle \boldsymbol{x}_s^t - \boldsymbol{x}_s^*, \nabla_{\boldsymbol{x}_s}F_n\left(\left\{\boldsymbol{x}_c^{t,i},\boldsymbol{x}_s^{t,i}\right\}\right)\right\rangle\right], \tag{92}$$

where we use $\nabla_{\boldsymbol{x}_s}f\left(\left\{\boldsymbol{x}_c,\boldsymbol{x}_s\right\}\right) = \sum_{n=1}^{N}\nabla_{\boldsymbol{x}_s}F_n\left(\left\{\boldsymbol{x}_c,\boldsymbol{x}_s\right\}\right)$.

For (92), we have

$$\left(\eta^t\right)^2 \tau N \sum_{n=1}^{N}\sum_{i=0}^{\tau-1}\mathbb{E}\left[\left\|\nabla_{\boldsymbol{x}_s}F_n\left(\left\{\boldsymbol{x}_{c,n}^{t,i},\boldsymbol{x}_{s,n}^{t,i}\right\}\right)\right\|^2\right]$$

$$= \left(\eta^t\right)^2 \tau N \sum_{n=1}^{N}\sum_{i=0}^{\tau-1}\mathbb{E}\left[\left\|\nabla_{\boldsymbol{x}_s}F_n\left(\left\{\boldsymbol{x}_{c,n}^{t,i},\boldsymbol{x}_{s,n}^{t,i}\right\}\right)\right.\right.$$

$$\left.\left. - \boldsymbol{g}_{s,n}^{t,i}\left(\left\{\boldsymbol{x}_{c,n}^{t,i},\boldsymbol{x}_{s,n}^{t,i}\right\}\right)\right\|^2\right] + \mathbb{E}\left[\left\|\boldsymbol{g}_{s,n}^{t,i}\left(\left\{\boldsymbol{x}_{c,n}^{t,i},\boldsymbol{x}_{s,n}^{t,i}\right\}\right)\right\|^2\right]$$

$$\leq \left(\eta^t\right)^2 \tau^2 N \sum_{n=1}^{N}\left(\sigma_n^2 + G^2\right), \tag{93}$$

where the first inequality applies triangle inequality. In the last inequality, we apply the bound of variance and expected squared norm for stochastic gradients in Assumption C.1.

Since $F_n(\boldsymbol{x})$ is $S$-smooth and $\mu$-strongly convex, using Lemma C.3 we have

$$- 2\eta^t\mathbb{E}\left[\sum_{n=1}^{N}\sum_{i=0}^{\tau-1}\left\langle \boldsymbol{x}_s^t - \boldsymbol{x}_s^*, \nabla_{\boldsymbol{x}_s}F_n\left(\left\{\boldsymbol{x}_{c,n}^{t,i},\boldsymbol{x}_{s,n}^{t,i}\right\}\right)\right\rangle\right]$$

$$\leq -2\eta^t\sum_{n=1}^{N}\sum_{i=0}^{\tau-1}\mathbb{E}\left[\left(F_n\left(\boldsymbol{x}^t\right) - F_n\left(\boldsymbol{x}^*\right)\right.\right.$$

$$\left.\left. + \frac{\mu}{4}\left\|\boldsymbol{x}_s^t - \boldsymbol{x}_s^*\right\|^2 - S\left\|\boldsymbol{x}_{s,n}^{t,i} - \boldsymbol{x}_s^t\right\|^2\right)\right]. \tag{94}$$

By Lemma C.5, we have

$$\sum_{n=1}^{N}\sum_{i=0}^{\tau-1}\mathbb{E}\left[\left\|\boldsymbol{x}_{s,n}^{t,i} - \boldsymbol{x}_s^t\right\|^2\right] \leq 12\sum_{n=1}^{N}\tau^3\left(\eta^t\right)^2\left(2\sigma_n^2 + G^2\right). \tag{95}$$

From Assumption C.1, the second part in (91) is bounded by

$$\mathbb{E}\left\|\sum_{n=1}^{N}\sum_{i=0}^{\tau-1}\boldsymbol{g}_{s,n}^{t,i} - \sum_{i=0}^{\tau-1}\nabla_{\boldsymbol{x}_s}f\left(\left\{\boldsymbol{x}_c^{t,i},\boldsymbol{x}_s^{t,i}\right\}\right)\right\|^2$$

$$\leq \tau\sum_{i=0}^{\tau-1}\mathbb{E}\left\|\sum_{n=1}^{N}\boldsymbol{g}_{s,n}^{t,i} - \nabla_{\boldsymbol{x}_s}F_n\left(\left\{\boldsymbol{x}_c^{t,i},\boldsymbol{x}_s^{t,i}\right\}\right)\right\|^2$$

$$\leq N\sum_{n=1}^{N}\sigma_n^2\tau^2. \tag{96}$$

Thus, (91) becomes

$$\mathbb{E}\left[\left\|\boldsymbol{x}_s^{t+1} - \boldsymbol{x}_s^*\right\|^2\right]$$

$$\leq \mathbb{E}\left[\left\|\boldsymbol{x}_s^t - \boldsymbol{x}_s^*\right\|^2\right] + \left(\eta^t\right)^2\tau^2 N\sum_{n=1}^{N}\left(\sigma_n^2 + G^2\right)$$

$$- 2\eta^t \tau \mathbb{E}\left[f\left(\boldsymbol{x}^t\right) - f\left(\boldsymbol{x}^*\right)\right]$$

$$- \frac{\mu N\tau\eta^t}{2} \left\|\boldsymbol{x}_s^t - \boldsymbol{x}_s^*\right\|^2 + 2\eta^t \left(12\sum_{n=1}^{N} S\tau^3 \left(\eta^t\right)^2 \left(2\sigma_n^2 + G^2\right)\right)$$

$$+ N\sum_{n=1}^{N} \left(\eta^t\right)^2 \sigma_n^2 \tau^2$$

$$\leq \left(1 - \frac{\eta^t N\tau\mu}{2}\right)\mathbb{E}\left[\left\|\boldsymbol{x}_s^t - \boldsymbol{x}_s^*\right\|^2\right] - 2\eta^t\tau\mathbb{E}\left[f\left(\boldsymbol{x}^t\right) - f\left(\boldsymbol{x}^*\right)\right]$$

$$+ \left(\eta^t\right)^2 \tau^2 N\sum_{n=1}^{N}\left(2\sigma_n^2 + G^2\right) + 24S\tau^3\left(\eta^t\right)^3\sum_{n=1}^{N}\left(2\sigma_n^2 + G^2\right). \tag{97}$$

$\square$

Using the above lemma, we can prove the convergence error. Let $\Delta^{t+1} \triangleq \mathbb{E}\left[\left\|\boldsymbol{x}_s^{t+1} - \boldsymbol{x}_s^*\right\|^2\right]$. We can rewrite (97) as:

$$\Delta^{t+1} \leq \left(1 - \frac{\eta^t N\tau\mu}{2}\right)\Delta^t - 2\eta^t\tau\mathbb{E}\left[f\left(\boldsymbol{x}^t\right) - f\left(\boldsymbol{x}^*\right)\right],$$

$$+ \left(\eta^t\right)^2 \tau^2 N\sum_{n=1}^{N}\left(2\sigma_n^2 + G^2\right) + 24S\tau^3\left(\eta^t\right)^3\sum_{n=1}^{N}\left(2\sigma_n^2 + G^2\right),$$

$$\leq \left(1 - \frac{\eta^t N\tau\mu}{2}\right)\Delta^t + \frac{\left(\eta^t\right)^2 \tau^2}{4}B_1 + \frac{\left(\eta^t\right)^3 \tau^3}{8}B_2. \tag{98}$$

where $B_1 := 4N\sum_{n=1}^{N}\left(2\sigma_n^2 + G^2\right)$ and $B := 192S\sum_{n=1}^{N}\left(2\sigma_n^2 + G^2\right)$.

Consider a diminishing stepsize $\eta^t = \frac{2\beta}{N\tau(\gamma_s+t)}$, i.e, $\frac{N\eta^t\tau}{2} = \frac{\beta}{\gamma_s+t}$, where $\beta = \frac{2}{\mu}, \gamma_s = \frac{8S}{N\mu} - 1$. It is easy to show that $\eta^t \leq \frac{1}{2S\tau}$ for all $t$. Next, we will prove that $\Delta^{t+1} \leq \frac{v}{\gamma_s+t+1}$, where $v = \max\left\{\frac{4B_1}{\mu^2} + \frac{8B_2}{\mu^3(\gamma_s+1)}, (\gamma_s+1)\Delta^0\right\}$. We prove this by induction. First, the definition of $v$ ensures that it holds for $t = -1$. Assume the conclusion holds for some $t$, it follows that

$$\Delta^{t+1} \leq \left(1 - \frac{N\eta^t\tau\mu}{2}\right)\Delta^t + \frac{\left(\eta^t\right)^2 \tau^2}{4}B_1 + \frac{\left(\eta^t\right)^3 \tau^3}{8}B_2 \tag{99}$$

$$\leq \left(1 - \frac{\mu\beta}{\gamma_s+t}\right)\frac{v}{\gamma_s+t} + \frac{\left(\eta^t\right)^2 \tau^2}{4}B_1 + \frac{\left(\eta^t\right)^3 \tau^3}{8}B_2 \tag{100}$$

$$= \frac{\gamma_s+t-1}{(\gamma_s+t)^2}v + \left[\frac{\beta^2 B_1}{(\gamma_s+t)^2} + \frac{\beta^3 B_2}{(\gamma_s+t)^3} - \frac{\beta\mu-1}{(\gamma_s+t)^2}v\right] \tag{101}$$

$$= \frac{\gamma_s+t-1}{(\gamma_s+t)^2}v + \left[\frac{\beta^2 B_1}{(\gamma_s+t)^2} + \frac{\beta^3 B_2}{(\gamma_s+t)^3} - \frac{\beta\mu-1}{(\gamma_s+t)^2}\max\left\{\frac{4B_1}{\mu^2} + \frac{8B_2}{\mu^3(\gamma_s+1)}, (\gamma_s+1)\Delta^0\right\}\right] \tag{102}$$

$$= \frac{\gamma_s+t-1}{(\gamma_s+t)^2}v + \left[\frac{\beta^2 B_1}{(\gamma_s+t)^2} + \frac{\beta^3 B_2}{(\gamma_s+t)^3} - \frac{\beta\mu-1}{(\gamma_s+t)^2}\max\left\{\frac{\beta^2 B_1}{\beta\mu-1} + \frac{\beta^3 B_2}{(\beta\mu-1)(\gamma_s+1)}, (\gamma_s+1)\Delta^0\right\}\right] \tag{103}$$

$$\leq \frac{\gamma_s+t-1}{(\gamma_s+t)^2}v \tag{104}$$

$$\leq \frac{v}{\gamma_s+t+1}. \tag{105}$$

Hence, we have proven that $\Delta^t \leq \frac{v}{\gamma_s + t}, \forall t$. Therefore, we have

$$
\mathbb{E}\left[\left\|\boldsymbol{x}_s^t - \boldsymbol{x}_s^*\right\|^2\right] = \Delta^t \leq \frac{v}{\gamma_s + t} = \frac{\max\left\{\frac{4B_1}{\mu^2} + \frac{8B_2}{\mu^3(\gamma_s+1)}, (\gamma_s+1)\mathbb{E}\left[\left\|\boldsymbol{x}_s^0 - \boldsymbol{x}_s^*\right\|^2\right]\right\}}{\gamma_s + t}
$$

$$
\leq \frac{16N\sum_{n=1}^{N}\left(2\sigma_n^2 + G^2\right)}{\mu^2(\gamma_s + t)} + \frac{1536S\ \sum_{n=1}^{N}\left(2\sigma_n^2 + G^2\right)}{\mu^3(\gamma_s + t)(\gamma_s + 1)} + \frac{(\gamma_s + 1)\mathbb{E}\left[\left\|\boldsymbol{x}_s^0 - \boldsymbol{x}_s^*\right\|^2\right]}{\gamma_s + t}. \quad (106)
$$

### E.1.2 One-round Parallel Update for Client-Side Models

Under Assumptions C.1 and C.2, if $\eta^t \leq \frac{1}{2S\tau}$, in round $t$, Lemma D.1 gives

$$
\mathbb{E}\left[\left\|\boldsymbol{x}_c^{t+1} - \boldsymbol{x}_c^*\right\|^2\right]
$$
$$
\leq \left(1 - \frac{\eta^t \tau \mu}{2}\right)\mathbb{E}\left[\left\|\boldsymbol{x}_c^t - \boldsymbol{x}_c^*\right\|^2\right] - 2\eta^t \tau \mathbb{E}\left[f\left(\boldsymbol{x}^t\right) - f\left(\boldsymbol{x}^*\right)\right]
$$
$$
+ \left(\eta^t\right)^2 \tau^2 N \sum_{n=1}^{N} a_n^2\left(2\sigma_n^2 + G^2\right) + 24S\tau^3\left(\eta^t\right)^3 \sum_{n=1}^{N} a_n\left(2\sigma_n^2 + G^2\right). \quad (107)
$$

Let $\Delta^{t+1} \triangleq \mathbb{E}\left[\left\|\boldsymbol{x}_c^{t+1} - \boldsymbol{x}_c^*\right\|^2\right]$. We can rewrite (107) as:

$$
\Delta^{t+1} \leq \left(1 - \frac{\eta^t \tau \mu}{2}\right)\Delta^t + \frac{\left(\eta^t\right)^2 \tau^2}{4}B_1 + \frac{\left(\eta^t\right)^3 \tau^3}{8}B_2. \quad (108)
$$

where $B_1 := 4N\sum_{n=1}^{N} a_n^2\left(2\sigma_n^2 + G^2\right)$ and $B_2 := 192S\sum_{n=1}^{N} a_n\left(2\sigma_n^2 + G^2\right)$.

Consider a diminishing stepsize $\eta^t = \frac{2\beta}{\tau(\gamma_c + t)}$, i.e, $\frac{\eta^t \tau}{2} = \frac{\beta}{\gamma_c + t}$, where $\beta = \frac{2}{\mu}, \gamma_c = \frac{8S}{\mu} - 1$. It is easy to show that $\eta^t \leq \frac{1}{2S\tau}$ for all $t$. For $v = \max\left\{\frac{4B_1}{\mu^2} + \frac{8B_2}{\mu^3(\gamma_c+1)}, (\gamma_c + 1)\Delta^0\right\}$, we can prove that $\Delta^t \leq \frac{v}{\gamma_c + t}, \forall t$. Therefore, we have

$$
\mathbb{E}\left[\left\|\boldsymbol{x}_c^t - \boldsymbol{x}_c^*\right\|^2\right] = \Delta^t \leq \frac{v}{\gamma_c + t} = \frac{\max\left\{\frac{4B_1}{\mu^2} + \frac{8B_2}{\mu^3(\gamma_c+1)}, (\gamma_c+1)\mathbb{E}\left[\left\|\boldsymbol{x}_c^0 - \boldsymbol{x}_c^*\right\|^2\right]\right\}}{\gamma_c + t}
$$

$$
\leq \frac{16N\sum_{n=1}^{N} a_n^2\left(2\sigma_n^2 + G^2\right)}{\mu^2(\gamma_c + t)} + \frac{1536S\ \sum_{n=1}^{N} a_n\left(2\sigma_n^2 + G^2\right)}{\mu^3(\gamma_c + t)(\gamma_c + 1)} + \frac{(\gamma_c + 1)\mathbb{E}\left[\left\|\boldsymbol{x}_c^0 - \boldsymbol{x}_c^*\right\|^2\right]}{\gamma_c + t}. \quad (109)
$$

### E.1.3 Superposition of M-Server and Clients

We merge the M-server-side and client-side models in (106) and (109) using Proposition 3.5. For $\eta^t \leq \frac{1}{2S\tau}$ and $\gamma = \frac{8S}{\mu} - 1$, we have

$$
\mathbb{E}\left[f(\boldsymbol{x}^T)\right] - f(\boldsymbol{x}^*)
$$
$$
\leq \frac{S}{2}\left(\mathbb{E}\|\boldsymbol{x}_s^T - \boldsymbol{x}_s^*\|^2 + \mathbb{E}\|\boldsymbol{x}_c^T - \boldsymbol{x}_c^*\|^2\right)
$$
$$
\leq \frac{8SN\sum_{n=1}^{N}\left(a_n^2+1\right)\left(2\sigma_n^2+G^2\right)}{\mu^2(\gamma + T)} + \frac{768S^2\sum_{n=1}^{N}\left(a_n+1\right)\left(2\sigma_n^2+G^2\right)}{\mu^3(\gamma + T)(\gamma + 1)} + \frac{S(\gamma+1)\mathbb{E}\left[\left\|\boldsymbol{x}^0 - \boldsymbol{x}^*\right\|^2\right]}{2(\gamma + T)} \quad (110)
$$

### E.2 General convex case for SFL-V2

#### E.2.1 One-round Sequential Update for M-Server-Side Model

By Lemma E.1 with $\mu = 0$ and $\eta^t \leq \frac{1}{2S\tau}$, we have

$$
\mathbb{E}\left[\left\|\boldsymbol{x}_s^{t+1} - \boldsymbol{x}_s^*\right\|^2\right]
$$
$$
\leq \mathbb{E}\left[\left\|\boldsymbol{x}_s^t - \boldsymbol{x}_s^*\right\|^2\right] - 2\eta^t\tau\mathbb{E}\left[f\left(\boldsymbol{x}^t\right) - f\left(\boldsymbol{x}^*\right)\right]
$$
$$
+ \left(\eta^t\right)^2\tau^2 N\sum_{n=1}^N\left(2\sigma_n^2 + G^2\right) + 24S\tau^3\left(\eta^t\right)^3\sum_{n=1}^N\left(2\sigma_n^2 + G^2\right). \tag{111}
$$

#### E.2.2 One-round Parallel Update for Client-Side Models

By Lemma D.1 with $\mu = 0$ and $\eta^t \leq \frac{1}{2S\tau}$, we have

$$
\mathbb{E}\left[\left\|\boldsymbol{x}_c^{t+1} - \boldsymbol{x}_c^*\right\|^2\right]
$$
$$
\leq \mathbb{E}\left[\left\|\boldsymbol{x}_c^t - \boldsymbol{x}_c^*\right\|^2\right] - 2\eta^t\tau\mathbb{E}\left[f\left(\boldsymbol{x}^t\right) - f\left(\boldsymbol{x}^*\right)\right]
$$
$$
+ \left(\eta^t\right)^2\tau^2 N\sum_{n=1}^N a_n^2\left(2\sigma_n^2 + G^2\right) + 24S\tau^3\left(\eta^t\right)^3\sum_{n=1}^N a_n\left(2\sigma_n^2 + G^2\right). \tag{112}
$$

#### E.2.3 Superposition of M-Server and Clients

We merge the M-server-side and client-side models in (111) and (112) as follows

$$
\mathbb{E}\left[\left\|\boldsymbol{x}^{t+1} - \boldsymbol{x}^*\right\|^2\right] \leq \mathbb{E}\left[\left\|\boldsymbol{x}_s^{t+1} - \boldsymbol{x}_s^*\right\|^2\right] + \mathbb{E}\left[\left\|\boldsymbol{x}_c^{t+1} - \boldsymbol{x}_c^*\right\|^2\right],
$$
$$
\leq \mathbb{E}\left[\left\|\boldsymbol{x}_s^t - \boldsymbol{x}_s^*\right\|^2\right] - 2\eta^t\tau\mathbb{E}\left[f\left(\boldsymbol{x}^t\right) - f\left(\boldsymbol{x}^*\right)\right]
$$
$$
+ \mathbb{E}\left[\left\|\boldsymbol{x}_c^t - \boldsymbol{x}_c^*\right\|^2\right] - 2\eta^t\tau\mathbb{E}\left[f\left(\boldsymbol{x}^t\right) - f\left(\boldsymbol{x}^*\right)\right]
$$
$$
+ \left(\eta^t\right)^2\tau^2 N\sum_{n=1}^N(a_n^2 + 1)\left(2\sigma_n^2 + G^2\right) + 24S\tau^3\left(\eta^t\right)^3\sum_{n=1}^N(a_n + 1)\left(2\sigma_n^2 + G^2\right)
$$
$$
= \mathbb{E}\left[\left\|\boldsymbol{x}^t - \boldsymbol{x}^*\right\|^2\right] - 4\eta^t\tau\mathbb{E}\left[f\left(\boldsymbol{x}^t\right) - f\left(\boldsymbol{x}^*\right)\right]
$$
$$
+ \left(\eta^t\right)^2\tau^2 N\sum_{n=1}^N(a_n^2 + 1)\left(2\sigma_n^2 + G^2\right) + 24S\tau^3\left(\eta^t\right)^3\sum_{n=1}^N(a_n + 1)\left(2\sigma_n^2 + G^2\right). \tag{113}
$$

Then, we can obtain the relation between $\mathbb{E}\left[\left\|\boldsymbol{x}^{t+1} - \boldsymbol{x}^*\right\|^2\right]$ and $\mathbb{E}\left[\left\|\boldsymbol{x}^t - \boldsymbol{x}^*\right\|^2\right]$, which is related to $\mathbb{E}\left[f\left(\boldsymbol{x}^t\right) - f\left(\boldsymbol{x}^*\right)\right]$. Applying Lemma 8 in [17], we obtain the performance bound as

$$
\mathbb{E}\left[f\left(\boldsymbol{x}^T\right)\right] - f\left(\boldsymbol{x}^*\right)
$$
$$
\leq \frac{1}{2}\left(N\sum_{n=1}^N(a_n^2 + 1)\left(2\sigma_n^2 + G^2\right)\right)^{\frac{1}{2}}\left(\frac{\left\|\boldsymbol{x}^0 - \boldsymbol{x}^*\right\|^2}{T + 1}\right)^{\frac{1}{2}}
$$
$$
+ \frac{1}{2}\left(24S\sum_{n=1}^N(a_n + 1)\left(2\sigma_n^2 + G^2\right)\right)^{\frac{1}{3}}\left(\frac{\left\|\boldsymbol{x}^0 - \boldsymbol{x}^*\right\|^2}{T + 1}\right)^{\frac{1}{3}} + \frac{S\left\|\boldsymbol{x}^0 - \boldsymbol{x}^*\right\|^2}{2(T + 1)}. \tag{114}
$$

### E.3 Non-convex case for SFL-V2

#### E.3.1 One-round Sequential Update for M-Server-Side Model

For the server, we have

$$
\mathbb{E}\left[\langle \nabla_{\boldsymbol{x}_s} f\left(\boldsymbol{x}^t\right), \boldsymbol{x}_s^{t+1} - \boldsymbol{x}_s^t \rangle\right]
$$
$$
\leq \mathbb{E}\left[\langle \nabla_{\boldsymbol{x}_s} f\left(\boldsymbol{x}^t\right), \boldsymbol{x}_s^{t+1} - \boldsymbol{x}_s^t + \eta^t \tau \nabla_{\boldsymbol{x}_s} f\left(\boldsymbol{x}^t\right) - \eta^t \tau \nabla_{\boldsymbol{x}_s} f\left(\boldsymbol{x}^t\right) \rangle\right]
$$
$$
\leq \mathbb{E}\left[\langle \nabla_{\boldsymbol{x}_s} f\left(\boldsymbol{x}^t\right), \boldsymbol{x}_s^{t+1} - \boldsymbol{x}_s^t + \eta^t \tau \nabla_{\boldsymbol{x}_s} f\left(\boldsymbol{x}^t\right) \rangle - \langle \nabla_{\boldsymbol{x}_s} f\left(\boldsymbol{x}_s^t\right), \eta^t \tau \nabla_{\boldsymbol{x}_s} f\left(\boldsymbol{x}^t\right) \rangle\right]
$$
$$
\leq \left\langle \nabla_{\boldsymbol{x}_s} f\left(\boldsymbol{x}^t\right), \mathbb{E}\left[-\eta^t \sum_{n=1}^{N}\sum_{i=0}^{\tau-1} \boldsymbol{g}_{s,n}^{t,i}\right] + \eta^t \tau \nabla_{\boldsymbol{x}_s} f\left(\boldsymbol{x}^t\right) \right\rangle - \eta^t \tau \left\|\nabla_{\boldsymbol{x}_s} f\left(\boldsymbol{x}^t\right)\right\|^2
$$
$$
\leq \left\langle \nabla_{\boldsymbol{x}_s} f\left(\boldsymbol{x}^t\right), \mathbb{E}\left[-\eta^t \sum_{n=1}^{N}\sum_{i=0}^{\tau-1} \nabla_{\boldsymbol{x}_s} F_n\left(\{\boldsymbol{x}_{c,n}^{t,i}, \boldsymbol{x}_{s,n}^{t,i}\}\right)\right] + \eta^t \tau \nabla_{\boldsymbol{x}_s} f\left(\boldsymbol{x}^t\right) \right\rangle - \eta^t \tau \left\|\nabla_{\boldsymbol{x}_s} f\left(\boldsymbol{x}^t\right)\right\|^2
$$
$$
\leq \left\langle \nabla_{\boldsymbol{x}_s} f\left(\boldsymbol{x}^t\right), \mathbb{E}\left[-\eta^t \sum_{n=1}^{N}\sum_{i=0}^{\tau-1} \nabla_{\boldsymbol{x}_s} F_n\left(\{\boldsymbol{x}_{c,n}^{t,i}, \boldsymbol{x}_{s,n}^{t,i}\}\right) + \eta^t \sum_{n=1}^{N}\sum_{i=0}^{\tau-1} \nabla_{\boldsymbol{x}_s} F_n\left(\boldsymbol{x}^t\right)\right] \right\rangle
$$
$$
- \eta^t \tau \left\|\nabla_{\boldsymbol{x}_s} f\left(\boldsymbol{x}^t\right)\right\|^2
$$
$$
\leq \eta^t \tau \left\langle \nabla_{\boldsymbol{x}_s} f\left(\boldsymbol{x}^t\right), \mathbb{E}\left[-\frac{1}{\tau} \sum_{n=1}^{N}\sum_{i=0}^{\tau-1} \nabla_{\boldsymbol{x}_s} F_n\left(\{\boldsymbol{x}_{c,n}^{t,i}, \boldsymbol{x}_{s,n}^{t,i}\}\right) + \frac{1}{\tau} \sum_{n=1}^{N}\sum_{i=0}^{\tau-1} \nabla_{\boldsymbol{x}_s} F_n\left(\boldsymbol{x}^t\right)\right] \right\rangle
$$
$$
- \eta^t \tau \left\|\nabla_{\boldsymbol{x}_s} f\left(\boldsymbol{x}^t\right)\right\|^2
$$
$$
\leq \frac{\eta^t \tau}{2} \left\|\nabla_{\boldsymbol{x}_s} f\left(\boldsymbol{x}^t\right)\right\|^2 + \frac{\eta^t}{2\tau} \mathbb{E}\left[\left\|\sum_{n=1}^{N}\sum_{i=0}^{\tau-1} \nabla_{\boldsymbol{x}_s} F_n\left(\{\boldsymbol{x}_{c,n}^{t,i}, \boldsymbol{x}_{s,n}^{t,i}\}\right) - \sum_{n=1}^{N}\sum_{i=0}^{\tau-1} \nabla_{\boldsymbol{x}_s} F_n\left(\boldsymbol{x}^t\right)\right\|^2\right]
$$
$$
- \eta^t \tau \left\|\nabla_{\boldsymbol{x}_s} f\left(\boldsymbol{x}^t\right)\right\|^2
$$
$$
\leq -\frac{\eta^t \tau}{2} \left\|\nabla_{\boldsymbol{x}_s} f\left(\boldsymbol{x}^t\right)\right\|^2 + \frac{\eta^t}{2\tau} \mathbb{E}\left[\left\|\sum_{n=1}^{N}\sum_{i=0}^{\tau-1} \left(\nabla_{\boldsymbol{x}_s} F_n\left(\{\boldsymbol{x}_{c,n}^{t,i}, \boldsymbol{x}_{s,n}^{t,i}\}\right) - \nabla_{\boldsymbol{x}_s} F_n\left(\boldsymbol{x}^t\right)\right)\right\|^2\right]
$$
$$
\leq -\frac{\eta^t \tau}{2} \left\|\nabla_{\boldsymbol{x}_s} f\left(\boldsymbol{x}^t\right)\right\|^2 + \frac{N\eta^t}{2\tau} \sum_{n=1}^{N} \mathbb{E}\left[\left\|\sum_{i=0}^{\tau-1} \left(\nabla_{\boldsymbol{x}_s} F_n\left(\{\boldsymbol{x}_{c,n}^{t,i}, \boldsymbol{x}_{s,n}^{t,i}\}\right) - \nabla_{\boldsymbol{x}_s} F_n\left(\boldsymbol{x}^t\right)\right)\right\|^2\right]
$$
$$
\leq -\frac{\eta^t \tau}{2} \left\|\nabla_{\boldsymbol{x}_s} f\left(\boldsymbol{x}^t\right)\right\|^2 + \frac{N\eta^t S^2}{2} \sum_{n=1}^{N}\sum_{i=0}^{\tau-1} \mathbb{E}\left[\left\|\boldsymbol{x}_{s,n}^{t,i} - \boldsymbol{x}_s^t\right\|^2\right], \tag{115}
$$

where we apply Assumption C.1, $\nabla_{\boldsymbol{x}_s} f\left(\boldsymbol{x}^t\right) = \sum_{n=1}^{N} \nabla_{\boldsymbol{x}_s} F_n\left(\boldsymbol{x}^t\right)$, and $\langle a, b \rangle \leq \frac{a^2 + b^2}{2}$.

By Lemma C.6 with $\eta^t \leq \frac{1}{\sqrt{8}S\tau}$, we have

$$
\sum_{i=0}^{\tau-1} \mathbb{E}\left[\left\|\boldsymbol{x}_{s,n}^{t,i} - \boldsymbol{x}_s^t\right\|^2\right] \leq 2\tau^2 \left(8\tau \left(\eta^t\right)^2 \sigma_n^2 + 8\tau \left(\eta^t\right)^2 \epsilon^2 + 8\tau \left(\eta^t\right)^2 \left\|\nabla_{\boldsymbol{x}_s} f\left(\boldsymbol{x}_s^t\right)\right\|^2\right). \tag{116}
$$

Thus, (115) becomes

$$
\mathbb{E}\left[\langle \nabla_{\boldsymbol{x}_s} f\left(\boldsymbol{x}^t\right), \boldsymbol{x}_s^{t+1} - \boldsymbol{x}_s^t \rangle\right]
$$
$$
\leq -\frac{\eta^t \tau}{2} \left\|\nabla_{\boldsymbol{x}_s} f\left(\boldsymbol{x}^t\right)\right\|^2 + \frac{N\eta^t S^2}{2} \sum_{n=1}^{N} 2\tau^2 \left(8\tau \left(\eta^t\right)^2 \sigma_n^2 + 8\tau \left(\eta^t\right)^2 \epsilon^2 + 8\tau \left(\eta^t\right)^2 \left\|\nabla_{\boldsymbol{x}_s} f\left(\boldsymbol{x}_s^t\right)\right\|^2\right)
$$
$$
\leq \left(-\frac{\eta^t \tau}{2} + 8N^2 \left(\eta^t\right)^3 \tau^3 S^2\right) \left\|\nabla_{\boldsymbol{x}_s} f\left(\boldsymbol{x}^t\right)\right\|^2 + 8N\eta^t S^2 \tau^3 \sum_{n=1}^{N} \left(\eta^t\right)^2 \left(\sigma_n^2 + \epsilon^2\right). \tag{117}
$$

Furthermore, we have

$$\frac{S}{2}\mathbb{E}\left[\left\|\boldsymbol{x}_s^{t+1} - \boldsymbol{x}_s^t\right\|^2\right]$$

$$= \frac{SN\left(\eta^t\right)^2}{2}\sum_{n=1}^{N}\mathbb{E}\left[\left\|\sum_{i=0}^{\tau-1}\boldsymbol{g}_{s,n}^{t,i}\right\|^2\right]$$

$$\leq \frac{SN\left(\eta^t\right)^2}{2}\sum_{n=1}^{N}\mathbb{E}\left[\left\|\sum_{i=0}^{\tau-1}\boldsymbol{g}_{s,n}^{t,i}\right\|^2\right]$$

$$\leq \frac{SN\left(\eta^t\right)^2\tau}{2}\sum_{n=1}^{N}\sum_{i=0}^{\tau-1}\mathbb{E}\left[\left\|\boldsymbol{g}_{s,n}^{t,i}\right\|^2\right]$$

$$\leq \frac{SN\left(\eta^t\right)^2\tau}{2}\sum_{n=1}^{N}\sum_{i=0}^{\tau-1}\mathbb{E}\left[\left\|\boldsymbol{g}_{s,n}^{t,i} - \boldsymbol{g}_{s,n}^t + \boldsymbol{g}_{s,n}^t\right\|^2\right]$$

$$\leq \frac{SN\left(\eta^t\right)^2\tau}{2}\sum_{n=1}^{N}\sum_{i=0}^{\tau-1}\left(\mathbb{E}\left[\left\|\boldsymbol{g}_{s,n}^{t,i} - \boldsymbol{g}_{s,n}^t\right\|^2\right] + \mathbb{E}\left[\left\|\boldsymbol{g}_{s,n}^t\right\|^2\right]\right)$$

$$\leq \frac{SN\left(\eta^t\right)^2\tau}{2}\sum_{n=1}^{N}\sum_{i=0}^{\tau-1}\left(\mathbb{E}\left[\left\|\boldsymbol{g}_{s,n}^{t,i} - \boldsymbol{g}_{s,n}^t\right\|^2\right] + \mathbb{E}\left[\left\|\nabla_{\boldsymbol{x}_s}F_n\left(\boldsymbol{x}^t\right)\right\|^2 + \sigma_n^2\right]\right), \quad (118)$$

where the last line uses Assumption C.1 and $\mathbb{E}\left[\|\mathbf{z}\|^2\right] = \|\mathbb{E}[\mathbf{z}]\|^2 + \mathbb{E}[\|\mathbf{z} - \mathbb{E}[\mathbf{z}]\|^2]$ for any random variable $\mathbf{z}$.

By Lemma C.7 with $\eta^t \leq \frac{1}{2S\tau}$, we have

$$\sum_{i=0}^{\tau-1}\mathbb{E}\left[\left\|\boldsymbol{g}_{s,n}^{t,i} - \boldsymbol{g}_{s,n}^t\right\|^2\right] \leq 8\tau^3\left(\eta^t\right)^2 S^2\left(\left\|\nabla_{\boldsymbol{x}_s}F_n\left(\boldsymbol{x}^t\right)\right\|^2 + \sigma_n^2\right). \quad (119)$$

Thus, (118) becomes

$$\frac{S}{2}\mathbb{E}\left[\left\|\boldsymbol{x}_s^{t+1} - \boldsymbol{x}_s^t\right\|^2\right]$$

$$\leq \frac{SN\left(\eta^t\right)^2\tau}{2}\sum_{n=1}^{N}\left(8\tau^3\left(\eta^t\right)^2 S^2\left(\left\|\nabla_{\boldsymbol{x}_s}F_n\left(\boldsymbol{x}^t\right)\right\|^2 + \sigma_n^2\right) + \tau\mathbb{E}\left[\left\|\nabla_{\boldsymbol{x}_s}F_n\left(\boldsymbol{x}^t\right)\right\|^2 + \sigma_n^2\right]\right)$$

$$\leq \frac{SN\left(\eta^t\right)^2\tau}{2}\sum_{n=1}^{N}\left(\tau + 8\tau^3\left(\eta^t\right)^2 S^2\right)\left(\left\|\nabla_{\boldsymbol{x}_s}F_n\left(\boldsymbol{x}^t\right)\right\|^2 + \sigma_n^2\right)$$

$$\leq \frac{SN\left(\eta^t\right)^2\tau}{2}\sum_{n=1}^{N}\left(\tau + 8\tau^3\left(\eta^t\right)^2 S^2\right)\left(\left\|\nabla_{\boldsymbol{x}_s}F_n\left(\boldsymbol{x}^t\right) - \nabla_{\boldsymbol{x}_s}f\left(\boldsymbol{x}^t\right) + \nabla_{\boldsymbol{x}_s}f\left(\boldsymbol{x}^t\right)\right\|^2 + \sigma_n^2\right)$$

$$\leq \frac{SN\left(\eta^t\right)^2\tau}{2}\sum_{n=1}^{N}\left(\tau + 8\tau^3\left(\eta^t\right)^2 S^2\right)\left(2\left\|\nabla_{\boldsymbol{x}_s}f\left(\boldsymbol{x}^t\right)\right\|^2 + 2\epsilon^2 + \sigma_n^2\right). \quad (120)$$

### E.3.2 One-round Parallel Update for Client-Side Models

The analysis of the client-side model update is the same as the client's model update in version 1. Thus, we have

$$\mathbb{E}\left[\left\langle \nabla_{\boldsymbol{x}_c}f\left(\boldsymbol{x}^t\right), \boldsymbol{x}_c^{t+1} - \boldsymbol{x}_c^t\right\rangle\right]$$

$$\leq \left(-\frac{\eta^t\tau}{2} + 8N\left(\eta^t\right)^3\tau^3 S^2\sum_{n=1}^{N}a_n^2\right)\left\|\nabla_{\boldsymbol{x}_c}f\left(\boldsymbol{x}^t\right)\right\|^2 + 8N\eta^t S^2\tau^3\sum_{n=1}^{N}a_n^2\left(\eta^t\right)^2\left(\sigma_n^2 + \epsilon^2\right).$$

$$(121)$$

For $\eta^t \le \frac{1}{2S\tau}$,

$$\frac{S}{2}\mathbb{E}\left[\left\|\boldsymbol{x}_c^{t+1} - \boldsymbol{x}_c^t\right\|^2\right]$$

$$\le \frac{SN\left(\eta^t\right)^2\tau}{2}\sum_{n=1}^{N}a_n^2\left(\tau + 8\tau^3\left(\eta^t\right)^2 S^2\right)\left(2\left\|\nabla_{\boldsymbol{x}_c}f\left(\boldsymbol{x}^t\right)\right\|^2 + 2\epsilon^2 + \sigma_n^2\right). \tag{122}$$

### E.3.3 Superposition of M-Server and Clients

Applying (117), (120), (122) and (121) into (36) in Proposition C.4, we have

$$\mathbb{E}\left[f\left(\boldsymbol{x}^{t+1}\right)\right] - f\left(\boldsymbol{x}^t\right)$$

$$\le \mathbb{E}\left[\langle\nabla_{\boldsymbol{x}_c}f\left(\boldsymbol{x}^t\right),\boldsymbol{x}_c^{t+1} - \boldsymbol{x}_c^t\rangle\right] + \frac{S}{2}\mathbb{E}\left[\left\|\boldsymbol{x}_c^{t+1} - \boldsymbol{x}_c^t\right\|^2\right] + \mathbb{E}\left[\langle\nabla_{\boldsymbol{x}_s}f\left(\boldsymbol{x}^t\right),\boldsymbol{x}_s^{t+1} - \boldsymbol{x}_s^t\rangle\right] + \frac{S}{2}\mathbb{E}\left[\left\|\boldsymbol{x}_s^{t+1} - \boldsymbol{x}_s^t\right\|^2\right]$$

$$\le \left(-\frac{\eta^t\tau}{2} + 8N^2\left(\eta^t\right)^3\tau^3 S^2\right)\left\|\nabla_{\boldsymbol{x}}f\left(\boldsymbol{x}^t\right)\right\|^2$$

$$+ 8N\eta^t S^2\tau^3\sum_{n=1}^{N}\left(\eta^t\right)^2\left(a_n^2 + 1\right)\left(\sigma_n^2 + \epsilon^2\right)$$

$$+ \frac{SN\left(\eta^t\right)^2\tau}{2}\sum_{n=1}^{N}\left(\tau + 8\tau^3\left(\eta^t\right)^2 S^2\right)2\left\|\nabla_{\boldsymbol{x}}f\left(\boldsymbol{x}^t\right)\right\|^2$$

$$+ \frac{SN\left(\eta^t\right)^2\tau}{2}\sum_{n=1}^{N}\left(a_n^2 + 1\right)\left(\tau + 8\tau^3\left(\eta^t\right)^2 S^2\right)\left(2\epsilon^2 + \sigma_n^2\right)$$

$$\le \left(-\frac{\eta^t\tau}{2} + 8N^2\left(\eta^t\right)^3 S^2\tau^3 + SN\left(\eta^t\right)^2\tau\sum_{n=1}^{N}\left(\tau + 8\tau^3\left(\eta^t\right)^2 S^2\right)\right)\left\|\nabla_{\boldsymbol{x}}f\left(\boldsymbol{x}^t\right)\right\|^2$$

$$+ 8N\eta^t S^2\tau^3\sum_{n=1}^{N}\left(\eta^t\right)^2\left(a_n^2 + 1\right)\left(\sigma_n^2 + \epsilon^2\right)$$

$$+ \frac{1}{2}SN\left(\eta^t\right)^2\tau\left(\tau + 8\tau^3\left(\eta^t\right)^2 S^2\right)\sum_{n=1}^{N}\left(a_n^2 + 1\right)\left(2\epsilon^2 + \sigma_n^2\right)$$

$$\le \left(-\frac{\eta^t\tau}{2} + SN^2\left(\eta^t\right)^2\tau^2 + 8N^2\left(\eta^t\right)^3 S^2\tau^3 + 8S^3 N^2\left(\eta^t\right)^4\tau^4\right)\left\|\nabla_{\boldsymbol{x}}f\left(\boldsymbol{x}^t\right)\right\|^2$$

$$+ 8N\left(\eta^t\right)^3 S^2\tau^3\sum_{n=1}^{N}\left(a_n^2 + 1\right)\sigma_n^2 + 8N\left(\eta^t\right)^3 S^2\tau^3\epsilon^2\sum_{n=1}^{N}\left(a_n^2 + 1\right)$$

$$+ SN\left(\eta^t\right)^2\tau^2\epsilon^2\sum_{n=1}^{N}\left(a_n^2 + 1\right) + \frac{1}{2}SN\left(\eta^t\right)^2\tau^2\sum_{n=1}^{N}\left(a_n^2 + 1\right)\sigma_n^2$$

$$+ 8NS^3\left(\eta^t\right)^4\tau^4\epsilon^2\sum_{n=1}^{N}\left(a_n^2 + 1\right) + 4NS^3\left(\eta^t\right)^4\tau^4\sum_{n=1}^{N}\left(a_n^2 + 1\right)\sigma_n^2$$

$$\le -\frac{\eta^t\tau}{2}\left(1 - 2SN^2\eta^t\frac{\tau^2}{\tau}\left(1 + 8S\eta^t\tau + 8S^2\left(\eta^t\right)^2\tau^2\right)\right)\left\|\nabla_{\boldsymbol{x}}f\left(\boldsymbol{x}^t\right)\right\|^2$$

$$+ \left(\frac{1}{2}NS\left(\eta^t\right)^2\tau^2 + 8N\left(\eta^t\right)^3 S^2\tau^3 + 4NS^3\left(\eta^t\right)^4\tau^4\right)\sum_{n=1}^{N}\left(a_n^2 + 1\right)\sigma_n^2$$

$$+ \left(NS\left(\eta^t\right)^2\tau^2 + 8N\left(\eta^t\right)^3 S^2\tau^3 + 8NSL^3\left(\eta^t\right)^4\tau^4\right)\sum_{n=1}^{N}\left(a_n^2 + 1\right)\epsilon^2$$

$$\le -\frac{\eta^t\tau}{2}\left(1 - 2N^2 S\eta^t\frac{\tau^2}{\tau}\left(1 + \frac{1}{2} + \frac{1}{32}\right)\right)\left\|\nabla_{\boldsymbol{x}}f\left(\boldsymbol{x}^t\right)\right\|^2$$

$$+ NS \left(\eta^t\right)^2 \tau^2 \left(\frac{1}{2} + \frac{1}{2} + \frac{1}{64}\right) \sum_{n=1}^{N} (a_n^2 + 1)\sigma_n^2 + 2SN \left(\eta^t\right)^2 \tau^2 \left(\frac{1}{2} + \frac{1}{4} + \frac{1}{64}\right) \sum_{n=1}^{N} (a_n^2 + 1)\epsilon^2$$

$$\leq -\frac{\eta^t \tau}{2} \left(1 - 4N^2 S \eta^t \frac{\tau^2}{\tau}\right) \left\|\nabla_{\boldsymbol{x}} f\left(\boldsymbol{x}^t\right)\right\|^2 + 2NS \left(\eta^t\right)^2 \sum_{n=1}^{N} \left(\tau^2 a_n^2 + \tau^2\right) \left(\sigma_n^2 + \epsilon^2\right)$$

$$\leq -\frac{\eta^t \tau}{4} \left\|\nabla_{\boldsymbol{x}} f\left(\boldsymbol{x}^t\right)\right\|^2 + 2NS \left(\eta^t\right)^2 \tau^2 \sum_{n=1}^{N} \left(a_n^2 + 1\right) \left(\sigma_n^2 + \epsilon^2\right), \tag{123}$$

where we first let $\eta^t \leq \frac{1}{16S\tau}$ and then let $\eta^t \leq \frac{1}{8SN^2\tau}$. We have applied $\sum_{n=1}^{N} a_n^2 \leq N$.
Rearranging the above we have

$$\eta^t \left\|\nabla_{\boldsymbol{x}} f\left(\boldsymbol{x}^t\right)\right\|^2 \leq \frac{4}{\tau} \left(f\left(\boldsymbol{x}^t\right) - \mathbb{E}\left[f\left(\boldsymbol{x}_s^{t+1}\right)\right]\right) + 8NS \left(\eta^t\right)^2 \tau \sum_{n=1}^{N} \frac{a_n^2 + 1}{\tau} \left(\sigma_n^2 + \epsilon^2\right). \tag{124}$$

Taking expectation and averaging over all $t$, we have

$$\frac{1}{T} \sum_{t=0}^{T-1} \eta^t \mathbb{E}\left[\left\|\nabla_{\boldsymbol{x}} f\left(\boldsymbol{x}^t\right)\right\|^2\right] \leq \frac{4}{T\tau} \left(f\left(\boldsymbol{x}_0\right) - f^*\right) + \frac{8NS\tau}{T} \sum_{n=1}^{N} (a_n^2 + 1) \left(\sigma_n^2 + \epsilon^2\right) \sum_{t=0}^{T-1} \left(\eta^t\right)^2. \tag{125}$$

# F   Proof of Theorem 3.8

- In Sec. F.1, we prove the strongly convex case.
- In Sec. F.2, we prove the general convex case.
- In Sec. F.3, we prove the non-convex case.

## F.1   Strongly convex case for SFL-V1

### F.1.1   One-round Parallel Update for M-Server-Side Model

We first bound the M-server-side model update in one round for full participation ($q_n = 1$ for all $n$), and then compute the difference between full participation and partial participation ($q_n < 1$ for some $n$). We denote $\mathbf{I}_n^t$ as a binary variable, taking 1 if client $n$ participates in model training in round $t$, and 0 otherwise. Practically, $\mathbf{I}_n^t$ follows a Bernoulli distribution with an expectation of $q_n$.

For full participation, Lemma D.1 gives

$$
\mathbb{E}\left[\left\|\overline{\boldsymbol{x}}_s^{t+1} - \boldsymbol{x}_s^*\right\|^2\right]
$$
$$
\leq \left(1 - \frac{\eta^t \tilde{\tau} \mu}{2}\right) \mathbb{E}\left[\left\|\boldsymbol{x}_s^t - \boldsymbol{x}_s^*\right\|^2\right] - 2\eta^t \tilde{\tau} \mathbb{E}\left[f\left(\boldsymbol{x}^t\right) - f\left(\boldsymbol{x}^*\right)\right]
$$
$$
+ \left(\eta^t\right)^2 (\tilde{\tau})^2 N \sum_{n=1}^{N} a_n^2 \left(2\sigma_n^2 + G^2\right) + 24S\left(\tilde{\tau}\right)^3 \left(\eta^t\right)^3 \sum_{n=1}^{N} a_n \left(2\sigma_n^2 + G^2\right). \tag{126}
$$

Considering that each client $n$ participates in model training with a probability $q_n$, we have

$$
\mathbb{E}\left[\left\|\boldsymbol{x}_s^{t+1} - \overline{\boldsymbol{x}}_s^{t+1}\right\|^2\right]
$$
$$
= \mathbb{E}\left[\left\|\boldsymbol{x}_s^{t+1} - \boldsymbol{x}_s^t + \boldsymbol{x}_s^t - \overline{\boldsymbol{x}}_s^{t+1}\right\|^2\right]
$$
$$
\leq \mathbb{E}\left[\left\|\boldsymbol{x}_s^{t+1} - \boldsymbol{x}_s^t\right\|^2\right]
$$
$$
\leq \mathbb{E}\left[\left\|\sum_{n=1}^{N} \eta^t \frac{a_n \mathbf{I}_t^n}{q_n} \sum_{i=0}^{\tilde{\tau}-1} \boldsymbol{g}_{s,n}^{t,i}\left(\left\{\boldsymbol{x}_{c,n}^{t,i}, \boldsymbol{x}_{s,n}^{t,i}\right\}\right)\right\|^2\right]
$$
$$
\leq N\tilde{\tau} \sum_{n=1}^{N} \left(\eta^t\right)^2 \frac{a_n^2}{q_n} \sum_{i=0}^{\tilde{\tau}-1} \mathbb{E}\left[\left\|\boldsymbol{g}_{s,n}^{t,i}\left(\left\{\boldsymbol{x}_{c,n}^{t,i}, \boldsymbol{x}_{s,n}^{t,i}\right\}\right)\right\|^2\right]
$$
$$
\leq N \left(\tilde{\tau}\right)^2 \left(\eta^t\right)^2 G^2 \sum_{n=1}^{N} \frac{a_n^2}{q_n}, \tag{127}
$$

where we use $\mathbb{E}\|X - \mathbb{E}X\|^2 \leq \mathbb{E}\|X\|^2$, $\mathbb{E}[\mathbf{I}_n^t] = q_n$, and $\boldsymbol{x}_s^{t+1} = \boldsymbol{x}_s^t - \eta^t \sum_{n \in \mathcal{P}^t(\boldsymbol{q})} \sum_{i=0}^{\tilde{\tau}-1} \frac{a_n^2}{q_n} \boldsymbol{g}_{s,n}^{t,i}\left(\left\{\boldsymbol{x}_{c,n}^{t,i}, \boldsymbol{x}_{s,n}^{t,i}\right\}\right)$.

Combining the above gives

$$
\mathbb{E}\left[\left\|\boldsymbol{x}_s^{t+1} - \boldsymbol{x}_s^*\right\|^2\right] = \mathbb{E}\left[\left\|\boldsymbol{x}_s^{t+1} - \overline{\boldsymbol{x}}_s^{t+1} + \overline{\boldsymbol{x}}_s^{t+1} - \boldsymbol{x}_s^*\right\|^2\right]
$$
$$
\leq \left(1 - \frac{\eta^t \tilde{\tau} \mu}{2}\right) \mathbb{E}\left[\left\|\boldsymbol{x}_s^t - \boldsymbol{x}_s^*\right\|^2\right]
$$
$$
+ \left(\eta^t\right)^2 (\tilde{\tau})^2 N \sum_{n=1}^{N} a_n^2 \left(2\sigma_n^2 + G^2\right) + 24S\left(\tilde{\tau}\right)^3 \left(\eta^t\right)^3 \sum_{n=1}^{N} a_n \left(2\sigma_n^2 + G^2\right)
$$
$$
+ N \left(\tilde{\tau}\right)^2 \left(\eta^t\right)^2 G^2 \sum_{n=1}^{N} \frac{a_n^2}{q_n}. \tag{128}
$$

Let $\Delta^{t+1} \triangleq \mathbb{E}\left[\left\|\boldsymbol{x}_s^{t+1} - \boldsymbol{x}_s^*\right\|^2\right]$. We can rewrite (128) as:

$$\Delta^{t+1} \leq \left(1 - \frac{\eta^t \tilde{\tau} \mu}{2}\right)\Delta^t + \frac{\left(\eta^t\right)^2 (\tilde{\tau})^2}{4} B_1 + \frac{\left(\eta^t\right)^3 (\tilde{\tau})^3}{8} B_2. \tag{129}$$

where $B_1 := 4N \sum_{n=1}^N a_n^2 \left(2\sigma_n^2 + G^2\right) + 4NG^2 \sum_{n=1}^N \frac{a_n^2}{q_n}$ and $B := 192S \sum_{n=1}^N a_n \left(2\sigma_n^2 + G^2\right)$.

Consider a diminishing stepsize $\eta^t = \frac{2\beta}{\tilde{\tau}(\gamma_s + t)}$, i.e, $\frac{\eta^t \tilde{\tau}}{2} = \frac{\beta}{\gamma_s + t}$, where $\beta = \frac{2}{\mu}, \gamma_s = \frac{8S}{\mu} - 1$. It is easy to show that $\eta^t \leq \frac{1}{2S\tilde{\tau}}$ for all $t$. We can prove that $\Delta^t \leq \frac{v}{\gamma_s + t}, \forall t$. Therefore, we have

$$\begin{aligned}
\mathbb{E}\left[\left\|\boldsymbol{x}_s^t - \boldsymbol{x}_s^*\right\|^2\right] = \Delta^t &\leq \frac{v}{\gamma_s + t} = \frac{\max\left\{\frac{4B_1}{\mu^2} + \frac{8B_2}{\mu^3(\gamma_s + 1)}, (\gamma_s + 1)\mathbb{E}\left[\left\|\boldsymbol{x}_s^0 - \boldsymbol{x}_s^*\right\|^2\right]\right\}}{\gamma_s + t} \\
&\leq \frac{16N \sum_{n=1}^N a_n^2 \left(2\sigma_n^2 + G^2\right) + 16NG^2 \sum_{n=1}^N \frac{a_n^2}{q_n}}{\mu^2 (\gamma_s + t)} + \frac{1536S \sum_{n=1}^N a_n \left(2\sigma_n^2 + G^2\right)}{\mu^3 (\gamma_s + t)(\gamma_s + 1)} \\
&\quad + \frac{(\gamma_s + 1)\mathbb{E}\left[\left\|\boldsymbol{x}_s^0 - \boldsymbol{x}_s^*\right\|^2\right]}{\gamma_s + t}.
\end{aligned} \tag{130}$$

### F.1.2 One-round Parallel Update for Client-Side Models

Define $\overline{\boldsymbol{x}}_t^c = \sum_{n=1}^N a_n \boldsymbol{x}_{c,n}^t$, which represents the aggregating weights in round $t$ for full participation. Using a similar derivation as the M-server side, we first bound the client-side model update in one round for full participation $\mathbb{E}\left[\left\|\overline{\boldsymbol{x}}_c^{t+1} - \boldsymbol{x}_c^*\right\|^2\right]$ and then bound the difference of client-side model parameters between full participation and partial participation $\mathbb{E}\left[\left\|\boldsymbol{x}_c^{t+1} - \boldsymbol{x}_c^*\right\|^2\right]$. The overall gradient update rule of clients in each training round is $\boldsymbol{x}_c^{t+1} = \boldsymbol{x}_c^t - \eta^t \sum_{n \in \mathcal{P}^t(\boldsymbol{q})} \sum_{i=0}^{\tau-1} \frac{a_n}{q_n} \boldsymbol{g}_{c,n}^{t,i}\left(\{\boldsymbol{x}_{c,n}^{t,i}, \boldsymbol{x}_{s,n}^{t,i}\}\right)$.

Under Assumptions C.1 and C.2, if $\eta^t \leq \frac{1}{2S\tau}$, in round $t$, Lemma D.1 gives

$$\begin{aligned}
&\mathbb{E}\left[\left\|\overline{\boldsymbol{x}}_c^{t+1} - \boldsymbol{x}_c^*\right\|^2\right] \\
&\leq \left(1 - \frac{\eta^t \tau \mu}{2}\right)\mathbb{E}\left[\left\|\boldsymbol{x}_c^t - \boldsymbol{x}_c^*\right\|^2\right] - 2\eta^t \tau \mathbb{E}\left[f\left(\boldsymbol{x}^t\right) - f\left(\boldsymbol{x}^*\right)\right] \\
&\quad + \left(\eta^t\right)^2 (\tau)^2 N \sum_{n=1}^N a_n^2 \left(2\sigma_n^2 + G^2\right) + 24S(\tau)^3 \left(\eta^t\right)^3 \sum_{n=1}^N a_n \left(2\sigma_n^2 + G^2\right).
\end{aligned} \tag{131}$$

Considering that each client $n$ participates in model training with a probability $q_n$, we have

$$\begin{aligned}
&\mathbb{E}\left[\left\|\boldsymbol{x}_c^{t+1} - \overline{\boldsymbol{x}}_c^{t+1}\right\|^2\right] \\
&= \mathbb{E}\left[\left\|\boldsymbol{x}_c^{t+1} - \boldsymbol{x}_c^t + \boldsymbol{x}_c^t - \overline{\boldsymbol{x}}_c^{t+1}\right\|^2\right] \\
&\leq \mathbb{E}\left[\left\|\boldsymbol{x}_c^{t+1} - \boldsymbol{x}_c^t\right\|^2\right] \\
&\leq \mathbb{E}\left[\left\|\sum_{n=1}^N \eta^t \frac{a_n \mathbf{I}_t^n}{q_n} \sum_{i=0}^{\tau-1} \boldsymbol{g}_{c,n}^{t,i}\left(\{\boldsymbol{x}_{c,n}^{t,i}, \boldsymbol{x}_{s,n}^{t,i}\}\right)\right\|^2\right] \\
&\leq N\tau \sum_{n=1}^N \left(\eta^t\right)^2 \frac{a_n^2}{q_n} \sum_{i=0}^{\tau-1} \mathbb{E}\left[\left\|\boldsymbol{g}_{c,n}^{t,i}\left(\{\boldsymbol{x}_{c,n}^{t,i}, \boldsymbol{x}_{s,n}^{t,i}\}\right)\right\|^2\right] \\
&\leq N(\tau)^2 \left(\eta^t\right)^2 G^2 \sum_{n=1}^N \frac{a_n^2}{q_n},
\end{aligned} \tag{132}$$

where we use $\mathbb{E}\|X - \mathbb{E}X\|^2 \leq \mathbb{E}\|X\|^2$, $\mathbb{E}[\mathbf{I}_n^t] = q_n$, and $\boldsymbol{x}_c^{t+1} = \boldsymbol{x}_c^t - \eta^t \sum_{n \in \mathcal{P}^t(\boldsymbol{q})} \sum_{i=0}^{\tau-1} \frac{a_n}{q_n} \boldsymbol{g}_{c,n}^{t,i} (\{\boldsymbol{x}_{c,n}^{t,i}, \boldsymbol{x}_{s,n}^{t,i}\})$.

We obtain the client-side model parameter update in one round for partial participation by combining the two terms and we have

$$
\begin{aligned}
\mathbb{E}\left[\left\|\boldsymbol{x}_c^{t+1} - \boldsymbol{x}_c^*\right\|^2\right] &= \mathbb{E}\left[\left\|\boldsymbol{x}_c^{t+1} - \overline{\boldsymbol{x}}_c^{t+1} + \overline{\boldsymbol{x}}_c^{t+1} - \boldsymbol{x}_c^*\right\|^2\right] \\
&\leq \left(1 - \frac{\eta^t \tau \mu}{2}\right) \mathbb{E}\left[\left\|\boldsymbol{x}_c^t - \boldsymbol{x}_c^*\right\|^2\right] \\
&\quad + \left(\eta^t\right)^2 (\tau)^2 N \sum_{n=1}^{N} a_n^2 \left(2\sigma_n^2 + G^2\right) + 24S(\tau)^3 \left(\eta^t\right)^3 \sum_{n=1}^{N} a_n \left(2\sigma_n^2 + G^2\right) \\
&\quad + N(\tau)^2 \left(\eta^t\right)^2 G^2 \sum_{n=1}^{N} \frac{a_n^2}{q_n},
\end{aligned}
\tag{133}
$$

where we consider $\mathbb{E}[f(\boldsymbol{x}^t) - f(\boldsymbol{x}^*)] \geq 0$.

Let $\Delta^{t+1} \triangleq \mathbb{E}\left[\left\|\boldsymbol{x}_c^{t+1} - \boldsymbol{x}_c^*\right\|^2\right]$. We can rewrite (163) as:

$$
\Delta^{t+1} \leq \left(1 - \frac{\eta^t \tau \mu}{2}\right) \Delta^t + \frac{\left(\eta^t\right)^2 (\tau)^2}{4} B_1 + \frac{\left(\eta^t\right)^3 (\tau)^3}{8} B_2.
\tag{134}
$$

where $B_1 := 4N \sum_{n=1}^{N} a_n^2 \left(2\sigma_n^2 + G^2\right) + 4NG^2 \sum_{n=1}^{N} \frac{a_n^2}{q_n}$ and $B_2 := 192S \sum_{n=1}^{N} a_n \left(2\sigma_n^2 + G^2\right)$.

Consider a diminishing stepsize $\eta^t = \frac{2\beta}{\tau(\gamma_c + t)}$, i.e, $\frac{\eta^t \tau}{2} = \frac{\beta}{\gamma_c + t}$, where $\beta = \frac{2}{\mu}, \gamma_c = \frac{8S}{\mu} - 1$. It is easy to show that $\eta^t \leq \frac{1}{2S\tau}$ for all $t$. For $v = \max\left\{\frac{4B_1}{\mu^2} + \frac{8B_2}{\mu^3(\gamma_c+1)}, (\gamma_c + 1)\Delta^0\right\}$, we can prove that $\Delta^t \leq \frac{v}{\gamma_c + t}, \forall t$. Therefore, we have

$$
\begin{aligned}
\mathbb{E}\left[\left\|\boldsymbol{x}_c^t - \boldsymbol{x}_c^*\right\|^2\right] = \Delta^t &\leq \frac{v}{\gamma_c + t} = \frac{\max\left\{\frac{4B_1}{\mu^2} + \frac{8B_2}{\mu^3(\gamma_c+1)}, (\gamma_c + 1)\mathbb{E}\left[\left\|\boldsymbol{x}_c^0 - \boldsymbol{x}_c^*\right\|^2\right]\right\}}{\gamma_c + t} \\
&\leq \frac{16N \sum_{n=1}^{N} a_n^2 \left(2\sigma_n^2 + G^2\right) + 16NG^2 \sum_{n=1}^{N} \frac{a_n^2}{q_n}}{\mu^2 (\gamma_c + t)} + \frac{1536S \sum_{n=1}^{N} a_n \left(2\sigma_n^2 + G^2\right)}{\mu^3 (\gamma_c + t)(\gamma_c + 1)} \\
&\quad + \frac{(\gamma_c + 1)\mathbb{E}\left[\left\|\boldsymbol{x}_c^0 - \boldsymbol{x}_c^*\right\|^2\right]}{\gamma_c + t}.
\end{aligned}
\tag{135}
$$

### F.1.3 Superposition of M-Server and Clients

We merge the M-server-side and client-side models in (130) and (135) using Proposition 3.5. For $\eta^t \leq \frac{1}{2S\max\{\tau, \tilde{\tau}\}}$ and $\gamma = \frac{8S}{\mu} - 1$, we have

$$
\begin{aligned}
&\mathbb{E}\left[f(\boldsymbol{x}^T)\right] - f(\boldsymbol{x}^*) \\
&\leq \frac{S}{2}\left(\mathbb{E}\|\boldsymbol{x}_s^T - \boldsymbol{x}_s^*\|^2 + \mathbb{E}\|\boldsymbol{x}_c^T - \boldsymbol{x}_c^*\|^2\right) \\
&\leq \frac{8SN \sum_{n=1}^{N} a_n^2 \left(2\sigma_n^2 + G^2 + \frac{G^2}{q_n}\right)}{\mu^2 (\gamma + T)} + \frac{768S^2 \sum_{n=1}^{N} a_n \left(2\sigma_n^2 + G^2\right)}{\mu^3 (\gamma + T)(\gamma + 1)} + \frac{S(\gamma + 1)\mathbb{E}\left[\left\|\boldsymbol{x}^0 - \boldsymbol{x}^*\right\|^2\right]}{2(\gamma + T)}.
\end{aligned}
\tag{136}
$$

## F.2 General convex case for SFL-V1

### F.2.1 One-round Parallel Update for M-Server-Side Model

By Lemma D.1 with $\mu = 0$ and $\eta^t \leq \frac{1}{2S\tilde{\tau}}$, we have

$$
\mathbb{E}\left[\left\|\overline{\boldsymbol{x}}_s^{t+1} - \boldsymbol{x}_s^*\right\|^2\right]
$$
$$
\leq \mathbb{E}\left[\left\|\boldsymbol{x}_s^t - \boldsymbol{x}_s^*\right\|^2\right] - 2\eta^t\tilde{\tau}\mathbb{E}\left[f\left(\boldsymbol{x}^t\right) - f\left(\boldsymbol{x}^*\right)\right]
$$
$$
+ \left(\eta^t\right)^2\tilde{\tau}^2 N \sum_{n=1}^N a_n^2 \left(2\sigma_n^2 + G^2\right) + 24S\tilde{\tau}^3\left(\eta^t\right)^3 \sum_{n=1}^N a_n\left(2\sigma_n^2 + G^2\right). \tag{137}
$$

Considering that each client $n$ participates in model training with a probability $q_n$, we have

$$
\mathbb{E}\left[\left\|\boldsymbol{x}_s^{t+1} - \overline{\boldsymbol{x}}_s^{t+1}\right\|^2\right] \leq N\tilde{\tau}^2\left(\eta^t\right)^2 G^2 \sum_{n=1}^N \frac{a_n^2}{q_n}. \tag{138}
$$

Thus, we have

$$
\mathbb{E}\left[\left\|\boldsymbol{x}_s^{t+1} - \boldsymbol{x}_s^*\right\|^2\right]
$$
$$
\leq \mathbb{E}\left[\left\|\boldsymbol{x}_s^t - \boldsymbol{x}_s^*\right\|^2\right] - 2\eta^t\tilde{\tau}\mathbb{E}\left[f\left(\boldsymbol{x}^t\right) - f\left(\boldsymbol{x}^*\right)\right]
$$
$$
+ \left(\eta^t\right)^2\tilde{\tau}^2 N \sum_{n=1}^N a_n^2 \left(2\sigma_n^2 + G^2\right) + 24S\tilde{\tau}^3\left(\eta^t\right)^3 \sum_{n=1}^N a_n\left(2\sigma_n^2 + G^2\right) + N\tilde{\tau}^2\left(\eta^t\right)^2 G^2 \sum_{n=1}^N \frac{a_n^2}{q_n}. \tag{139}
$$

### F.2.2 One-round Parallel Update for Client-Side Models

By Lemma D.1 with $\mu = 0$ and $\eta^t \leq \frac{1}{2S\tau}$, we have

$$
\mathbb{E}\left[\left\|\overline{\boldsymbol{x}}_c^{t+1} - \boldsymbol{x}_c^*\right\|^2\right]
$$
$$
\leq \mathbb{E}\left[\left\|\boldsymbol{x}_c^t - \boldsymbol{x}_c^*\right\|^2\right] - 2\eta^t\tau\mathbb{E}\left[f\left(\boldsymbol{x}^t\right) - f\left(\boldsymbol{x}^*\right)\right]
$$
$$
+ \left(\eta^t\right)^2\tau^2 N \sum_{n=1}^N a_n^2 \left(2\sigma_n^2 + G^2\right) + 24S\tau^3\left(\eta^t\right)^3 \sum_{n=1}^N a_n\left(2\sigma_n^2 + G^2\right). \tag{140}
$$

Considering that each client $n$ participates in model training with a probability $q_n$, we have

$$
\mathbb{E}\left[\left\|\boldsymbol{x}_c^{t+1} - \overline{\boldsymbol{x}}_c^{t+1}\right\|^2\right] \leq N\tau^2\left(\eta^t\right)^2 G^2 \sum_{n=1}^N \frac{a_n^2}{q_n}. \tag{141}
$$

Thus, we have

$$
\mathbb{E}\left[\left\|\boldsymbol{x}_c^{t+1} - \boldsymbol{x}_c^*\right\|^2\right]
$$
$$
\leq \mathbb{E}\left[\left\|\boldsymbol{x}_c^t - \boldsymbol{x}_c^*\right\|^2\right] - 2\eta^t\tau\mathbb{E}\left[f\left(\boldsymbol{x}^t\right) - f\left(\boldsymbol{x}^*\right)\right]
$$
$$
+ \left(\eta^t\right)^2\tau^2 N \sum_{n=1}^N a_n^2 \left(2\sigma_n^2 + G^2\right) + 24S\tau^3\left(\eta^t\right)^3 \sum_{n=1}^N a_n\left(2\sigma_n^2 + G^2\right) + N\tau^2\left(\eta^t\right)^2 G^2 \sum_{n=1}^N \frac{a_n^2}{q_n}. \tag{142}
$$

### F.2.3 Superposition of M-Server and Clients

We merge the M-server-side and client-side models in (139) and (142) as follows

$$\mathbb{E}\left[\left\|\boldsymbol{x}^{t+1}-\boldsymbol{x}^*\right\|^2\right] \leq \mathbb{E}\left[\left\|\boldsymbol{x}_s^{t+1}-\boldsymbol{x}_s^*\right\|^2\right] + \mathbb{E}\left[\left\|\boldsymbol{x}_c^{t+1}-\boldsymbol{x}_c^*\right\|^2\right],$$

$$\leq \mathbb{E}\left[\left\|\boldsymbol{x}_s^t - \boldsymbol{x}_s^*\right\|^2\right] - 2\eta^t\tilde{\tau}\mathbb{E}\left[f\left(\boldsymbol{x}^t\right) - f\left(\boldsymbol{x}^*\right)\right]$$

$$+ \left(\eta^t\right)^2 \tilde{\tau}^2 N \sum_{n=1}^N a_n^2 \left(2\sigma_n^2 + G^2\right) + 24S\tilde{\tau}^3 \left(\eta^t\right)^3 \sum_{n=1}^N a_n \left(2\sigma_n^2 + G^2\right) + N\tilde{\tau}^2 \left(\eta^t\right)^2 G^2 \sum_{n=1}^N \frac{a_n^2}{q_n}$$

$$+ \mathbb{E}\left[\left\|\boldsymbol{x}_c^t - \boldsymbol{x}_c^*\right\|^2\right] - 2\eta^t\tau\mathbb{E}\left[f\left(\boldsymbol{x}^t\right) - f\left(\boldsymbol{x}^*\right)\right]$$

$$+ \left(\eta^t\right)^2 \tau^2 N \sum_{n=1}^N a_n^2 \left(2\sigma_n^2 + G^2\right) + 24S\tau^3 \left(\eta^t\right)^3 \sum_{n=1}^N a_n \left(2\sigma_n^2 + G^2\right) + N\tau^2 \left(\eta^t\right)^2 G^2 \sum_{n=1}^N \frac{a_n^2}{q_n}$$

$$= \mathbb{E}\left[\left\|\boldsymbol{x}^t - \boldsymbol{x}^*\right\|^2\right] - 4\eta^t \min\{\tau, \tilde{\tau}\} \mathbb{E}\left[f\left(\boldsymbol{x}^t\right) - f\left(\boldsymbol{x}^*\right)\right]$$

$$+ \left(\eta^t\right)^2 N \sum_{n=1}^N a_n^2(\tilde{\tau}^2 + \tau^2)\left(2\sigma_n^2 + G^2\right) + 24S\left(\eta^t\right)^3 \sum_{n=1}^N a_n(\tilde{\tau}^3 + \tau^3)\left(2\sigma_n^2 + G^2\right) + N\left(\tau^2 + \tilde{\tau}^2\right)\left(\eta^t\right)^2 G^2 \sum_{n=1}^N \frac{a_n^2}{q_n}.$$

$$(143)$$

Then, we can obtain the relation between $\mathbb{E}\left[\left\|\boldsymbol{x}^{t+1}-\boldsymbol{x}^*\right\|^2\right]$ and $\mathbb{E}\left[\left\|\boldsymbol{x}^t-\boldsymbol{x}^*\right\|^2\right]$, which is related to $\mathbb{E}\left[f\left(\boldsymbol{x}^t\right) - f\left(\boldsymbol{x}^*\right)\right]$. Applying Lemma 8 in [17] and let $\tau_{\min} := \min\{\tilde{\tau}, \tau\}$, we obtain the performance bound as

$$\mathbb{E}\left[f\left(\boldsymbol{x}^T\right)\right] - f\left(\boldsymbol{x}^*\right)$$

$$\leq \frac{1}{2}\left(\frac{\tilde{\tau}^2 + \tau^2}{\tau_{\min}^2} N \sum_{n=1}^N a_n^2 \left(2\sigma_n^2 + G^2 + \frac{G_n^2}{q_n}\right)\right)^{\frac{1}{2}} \left(\frac{\left\|\boldsymbol{x}^0 - \boldsymbol{x}^*\right\|^2}{T+1}\right)^{\frac{1}{2}}$$

$$+ \frac{1}{2}\left(\frac{\tilde{\tau}^2 + \tau^2}{\tau_{\min}^2} 24S \sum_{n=1}^N a_n \left(2\sigma_n^2 + G^2\right)\right)^{\frac{1}{3}} \left(\frac{\left\|\boldsymbol{x}^0 - \boldsymbol{x}^*\right\|^2}{T+1}\right)^{\frac{1}{3}} + \frac{S\left\|\boldsymbol{x}^0 - \boldsymbol{x}^*\right\|^2}{2(T+1)}. \qquad (144)$$

## F.3 Non-convex case for SFL-V1

### F.3.1 One-round Parallel Update for M-Server-Side Model

For the server, we have

$$\mathbb{E}\left[\left\langle \nabla_{\boldsymbol{x}_s} f\left(\boldsymbol{x}^t\right), \boldsymbol{x}_s^{t+1} - \boldsymbol{x}_s^t\right\rangle\right]$$

$$\leq \mathbb{E}\left[\left\langle \nabla_{\boldsymbol{x}_s} f\left(\boldsymbol{x}^t\right), \boldsymbol{x}_s^{t+1} - \boldsymbol{x}_s^t + \eta^t\tilde{\tau}\nabla_{\boldsymbol{x}_s} f\left(\boldsymbol{x}^t\right) - \eta^t\tilde{\tau}\nabla_{\boldsymbol{x}_s} f\left(\boldsymbol{x}^t\right)\right\rangle\right]$$

$$\leq \mathbb{E}\left[\left\langle \nabla_{\boldsymbol{x}_s} f\left(\boldsymbol{x}^t\right), \boldsymbol{x}_s^{t+1} - \boldsymbol{x}_s^t + \eta^t\tilde{\tau}\nabla_{\boldsymbol{x}_s} f\left(\boldsymbol{x}^t\right)\right\rangle - \left\langle \nabla_{\boldsymbol{x}_s} f\left(\boldsymbol{x}_s^t\right), \eta^t\tilde{\tau}\nabla_{\boldsymbol{x}_s} f\left(\boldsymbol{x}^t\right)\right\rangle\right]$$

$$\leq \left\langle \nabla_{\boldsymbol{x}_s} f\left(\boldsymbol{x}^t\right), \mathbb{E}\left[-\eta^t \sum_{n=1}^{N}\sum_{i=0}^{\tilde{\tau}-1}\frac{a_n\mathbf{I}_n^t}{q_n}\boldsymbol{g}_{s,n}^{t,i}\right] + \eta^t\tilde{\tau}\nabla_{\boldsymbol{x}_s} f\left(\boldsymbol{x}^t\right)\right\rangle - \eta^t\tilde{\tau}\left\|\nabla_{\boldsymbol{x}_s} f\left(\boldsymbol{x}^t\right)\right\|^2$$

$$\leq \left\langle \nabla_{\boldsymbol{x}_s} f\left(\boldsymbol{x}^t\right), \mathbb{E}\left[-\eta^t \sum_{n=1}^{N}\sum_{i=0}^{\tilde{\tau}-1}\frac{a_n\mathbf{I}_n^t}{q_n}\nabla_{\boldsymbol{x}_s} F_n\left(\upsilon\left\{\boldsymbol{x}_{c,n}^{t,i}, \boldsymbol{x}_{s,n}^{t,i}\right\}\right)\right] + \eta^t\tilde{\tau}\nabla_{\boldsymbol{x}_s} f\left(\boldsymbol{x}^t\right)\right\rangle - \eta^t\tilde{\tau}\left\|\nabla_{\boldsymbol{x}_s} f\left(\boldsymbol{x}^t\right)\right\|^2$$

$$\leq \left\langle \nabla_{\boldsymbol{x}_s} f\left(\boldsymbol{x}^t\right), \mathbb{E}\left[-\eta^t \sum_{n=1}^{N}\sum_{i=0}^{\tilde{\tau}-1}\frac{a_n\mathbf{I}_n^t}{q_n}\nabla_{\boldsymbol{x}_s} F_n\left(\left\{\boldsymbol{x}_{c,n}^{t,i}, \boldsymbol{x}_{s,n}^{t,i}\right\}\right) + \eta^t\sum_{n=1}^{N}\sum_{i=0}^{\tilde{\tau}-1}\frac{a_n\mathbf{I}_n^t}{q_n}\nabla_{\boldsymbol{x}_s} F_n\left(\boldsymbol{x}^t\right)\right]\right\rangle$$
$$- \eta^t\tilde{\tau}\left\|\nabla_{\boldsymbol{x}_s} f\left(\boldsymbol{x}^t\right)\right\|^2$$

$$\leq \eta^t\tilde{\tau}\left\langle \nabla_{\boldsymbol{x}_s} f\left(\boldsymbol{x}^t\right), \mathbb{E}\left[-\frac{1}{\tilde{\tau}} \sum_{n=1}^{N}\sum_{i=0}^{\tilde{\tau}-1}\frac{a_n\mathbf{I}_n^t}{q_n}\nabla_{\boldsymbol{x}_s} F_n\left(\left\{\boldsymbol{x}_{c,n}^{t,i}, \boldsymbol{x}_{s,n}^{t,i}\right\}\right) + \frac{1}{\tilde{\tau}}\sum_{n=1}^{N}\sum_{i=0}^{\tilde{\tau}-1}\frac{a_n\mathbf{I}_n^t}{q_n}\nabla_{\boldsymbol{x}_s} F_n\left(\boldsymbol{x}^t\right)\right]\right\rangle$$
$$- \eta^t\tilde{\tau}\left\|\nabla_{\boldsymbol{x}_s} f\left(\boldsymbol{x}^t\right)\right\|^2$$

$$\leq \frac{\eta^t\tilde{\tau}}{2}\left\|\nabla_{\boldsymbol{x}_s} f\left(\boldsymbol{x}^t\right)\right\|^2 + \frac{\eta^t}{2\tilde{\tau}}\mathbb{E}\left[\left\|\sum_{n=1}^{N}\sum_{i=0}^{\tilde{\tau}-1}\frac{a_n\mathbf{I}_n^t}{q_n}\nabla_{\boldsymbol{x}_s} F_n\left(\left\{\boldsymbol{x}_{c,n}^{t,i}, \boldsymbol{x}_{s,n}^{t,i}\right\}\right) - \sum_{n=1}^{N}\sum_{i=0}^{\tilde{\tau}-1}\frac{a_n\mathbf{I}_n^t}{q_n}\nabla_{\boldsymbol{x}_s} F_n\left(\boldsymbol{x}^t\right)\right\|^2\right]$$
$$- \eta^t\tilde{\tau}\left\|\nabla_{\boldsymbol{x}_s} f\left(\boldsymbol{x}^t\right)\right\|^2$$

$$\leq -\frac{\eta^t\tilde{\tau}}{2}\left\|\nabla_{\boldsymbol{x}_s} f\left(\boldsymbol{x}^t\right)\right\|^2 + \frac{\eta^t}{2\tilde{\tau}}\mathbb{E}\left[\left\|\sum_{n=1}^{N}\frac{a_n\mathbf{I}_n^t}{q_n}\sum_{i=0}^{\tilde{\tau}-1}\left(\nabla_{\boldsymbol{x}_s} F_n\left(\left\{\boldsymbol{x}_{c,n}^{t,i}, \boldsymbol{x}_{s,n}^{t,i}\right\}\right) - \nabla_{\boldsymbol{x}_s} F_n\left(\boldsymbol{x}^t\right)\right)\right\|^2\right]$$

$$\leq -\frac{\eta^t\tilde{\tau}}{2}\left\|\nabla_{\boldsymbol{x}_s} f\left(\boldsymbol{x}^t\right)\right\|^2 + \frac{N\eta^t}{2\tilde{\tau}}\sum_{n=1}^{N}\frac{a_n^2}{q_n}\mathbb{E}\left[\left\|\sum_{i=0}^{\tilde{\tau}-1}\left(\nabla_{\boldsymbol{x}_s} F_n\left(\left\{\boldsymbol{x}_{c,n}^{t,i}, \boldsymbol{x}_{s,n}^{t,i}\right\}\right) - \nabla_{\boldsymbol{x}_s} F_n\left(\boldsymbol{x}^t\right)\right)\right\|^2\right]$$

$$\leq -\frac{\eta^t\tilde{\tau}}{2}\left\|\nabla_{\boldsymbol{x}_s} f\left(\boldsymbol{x}^t\right)\right\|^2 + \frac{N\eta^t S^2}{2}\sum_{n=1}^{N}\frac{a_n^2}{q_n}\sum_{i=0}^{\tilde{\tau}-1}\mathbb{E}\left[\left\|\boldsymbol{x}_{s,n}^{t,i} - \boldsymbol{x}_s^t\right\|^2\right], \tag{145}$$

where we apply Assumption C.1, $\nabla_{\boldsymbol{x}_s} f\left(\boldsymbol{x}^t\right) = \sum_{n=1}^{N} a_n\nabla_{\boldsymbol{x}_s} F_n\left(\boldsymbol{x}^t\right)$, $\langle a, b\rangle \leq \frac{a^2+b^2}{2}$, and $\mathbb{E}\left[\mathbf{I}_n^t\right] = q_n$.

By Lemma C.6 with $\eta^t \leq \frac{1}{\sqrt{8}S\tilde{\tau}}$, we have

$$\sum_{i=0}^{\tilde{\tau}-1}\mathbb{E}\left[\left\|\boldsymbol{x}_{s,n}^{t,i} - \boldsymbol{x}_s^t\right\|^2\right] \leq 2\tilde{\tau}^2\left(8\tilde{\tau}\left(\eta^t\right)^2\sigma_n^2 + 8\tilde{\tau}\left(\eta^t\right)^2\epsilon^2 + 8\tilde{\tau}\left(\eta^t\right)^2\left\|\nabla_{\boldsymbol{x}_s} f\left(\boldsymbol{x}_s^t\right)\right\|^2\right). \tag{146}$$

Thus, (145) becomes

$$\mathbb{E}\left[\left\langle \nabla_{\boldsymbol{x}_s} f\left(\boldsymbol{x}^t\right), \boldsymbol{x}_s^{t+1} - \boldsymbol{x}_s^t\right\rangle\right]$$

$$\leq -\frac{\eta^t\tilde{\tau}}{2}\left\|\nabla_{\boldsymbol{x}_s} f\left(\boldsymbol{x}^t\right)\right\|^2 + \frac{N\eta^t S^2}{2}\sum_{n=1}^{N}\frac{a_n^2}{q_n}2\tilde{\tau}^2\left(8\tilde{\tau}\left(\eta^t\right)^2\sigma_n^2 + 8\tilde{\tau}\left(\eta^t\right)^2\epsilon^2 + 8\tilde{\tau}\left(\eta^t\right)^2\left\|\nabla_{\boldsymbol{x}_s} f\left(\boldsymbol{x}_s^t\right)\right\|^2\right)$$

$$\leq \left(-\frac{\eta^t\tilde{\tau}}{2} + 8N\left(\eta^t\right)^3\tilde{\tau}^3 S^2\sum_{n=1}^{N}\frac{a_n^2}{q_n}\right)\left\|\nabla_{\boldsymbol{x}_s} f\left(\boldsymbol{x}^t\right)\right\|^2 + 8N\eta^t S^2\tilde{\tau}^3\sum_{n=1}^{N}\frac{a_n^2}{q_n}\left(\eta^t\right)^2\left(\sigma_n^2 + \epsilon^2\right).$$
$$\tag{147}$$

Furthermore, we have

$$\frac{S}{2}\mathbb{E}\left[\left\|\boldsymbol{x}_s^{t+1}-\boldsymbol{x}_s^t\right\|^2\right]$$

$$=\frac{SN\left(\eta^t\right)^2}{2}\sum_{n=1}^N\mathbb{E}\left[\left\|\sum_{i=0}^{\tilde{\tau}-1}\frac{a_n\mathbf{I}_n^t}{q_n}\boldsymbol{g}_{s,n}^{t,i}\right\|^2\right]$$

$$\leq\frac{SN\left(\eta^t\right)^2}{2}\sum_{n=1}^N\frac{a_n^2}{q_n}\mathbb{E}\left[\left\|\sum_{i=0}^{\tilde{\tau}-1}\boldsymbol{g}_{s,n}^{t,i}\right\|^2\right]$$

$$\leq\frac{SN\left(\eta^t\right)^2\tilde{\tau}}{2}\sum_{n=1}^N\frac{a_n^2}{q_n}\sum_{i=0}^{\tilde{\tau}-1}\mathbb{E}\left[\left\|\boldsymbol{g}_{s,n}^{t,i}\right\|^2\right]$$

$$\leq\frac{SN\left(\eta^t\right)^2\tilde{\tau}}{2}\sum_{n=1}^N\frac{a_n^2}{q_n}\sum_{i=0}^{\tilde{\tau}-1}\mathbb{E}\left[\left\|\boldsymbol{g}_{s,n}^{t,i}-\boldsymbol{g}_{s,n}^t+\boldsymbol{g}_{s,n}^t\right\|^2\right]$$

$$\leq\frac{SN\left(\eta^t\right)^2\tilde{\tau}}{2}\sum_{n=1}^N\frac{a_n^2}{q_n}\sum_{i=0}^{\tilde{\tau}-1}\left(\mathbb{E}\left[\left\|\boldsymbol{g}_{s,n}^{t,i}-\boldsymbol{g}_{s,n}^t\right\|^2\right]+\mathbb{E}\left[\left\|\boldsymbol{g}_{s,n}^t\right\|^2\right]\right)$$

$$\leq\frac{SN\left(\eta^t\right)^2\tilde{\tau}}{2}\sum_{n=1}^N\frac{a_n^2}{q_n}\sum_{i=0}^{\tilde{\tau}-1}\left(\mathbb{E}\left[\left\|\boldsymbol{g}_{s,n}^{t,i}-\boldsymbol{g}_{s,n}^t\right\|^2\right]+\mathbb{E}\left[\left\|\nabla_{\boldsymbol{x}_s}F_n\left(\boldsymbol{x}^t\right)\right\|^2+\sigma_n^2\right]\right),\qquad(148)$$

where the last line uses Assumption C.1 and $\mathbb{E}\left[\|\mathbf{z}\|^2\right]=\|\mathbb{E}[\mathbf{z}]\|^2+\mathbb{E}[\|\mathbf{z}-\mathbb{E}[\mathbf{z}]\|^2]$ for any random variable $\mathbf{z}$.

By Lemma C.7 with $\eta^t\leq\frac{1}{2S\tilde{\tau}}$, we have

$$\sum_{i=0}^{\tilde{\tau}-1}\mathbb{E}\left[\left\|\boldsymbol{g}_{s,n}^{t,i}-\boldsymbol{g}_{s,n}^t\right\|^2\right]\leq8\tilde{\tau}^3\left(\eta^t\right)^2S^2\left(\left\|\nabla_{\boldsymbol{x}_s}F_n\left(\boldsymbol{x}^t\right)\right\|^2+\sigma_n^2\right).\qquad(149)$$

Thus, (148) becomes

$$\frac{S}{2}\mathbb{E}\left[\left\|\boldsymbol{x}_s^{t+1}-\boldsymbol{x}_s^t\right\|^2\right]$$

$$\leq\frac{SN\left(\eta^t\right)^2\tilde{\tau}}{2}\sum_{n=1}^N\frac{a_n^2}{q_n}\left(8\tilde{\tau}^3\left(\eta^t\right)^2S^2\left(\left\|\nabla_{\boldsymbol{x}_s}F_n\left(\boldsymbol{x}^t\right)\right\|^2+\sigma_n^2\right)+\tilde{\tau}\mathbb{E}\left[\left\|\nabla_{\boldsymbol{x}_s}F_n\left(\boldsymbol{x}^t\right)\right\|^2+\sigma_n^2\right]\right)$$

$$\leq\frac{SN\left(\eta^t\right)^2\tilde{\tau}}{2}\sum_{n=1}^N\frac{a_n^2}{q_n}\left(\tilde{\tau}+8\tilde{\tau}^3\left(\eta^t\right)^2S^2\right)\left(\left\|\nabla_{\boldsymbol{x}_s}F_n\left(\boldsymbol{x}^t\right)\right\|^2+\sigma_n^2\right)$$

$$\leq\frac{SN\left(\eta^t\right)^2\tilde{\tau}}{2}\sum_{n=1}^N\frac{a_n^2}{q_n}\left(\tilde{\tau}+8\tilde{\tau}^3\left(\eta^t\right)^2S^2\right)\left(\left\|\nabla_{\boldsymbol{x}_s}F_n\left(\boldsymbol{x}^t\right)-\nabla_{\boldsymbol{x}_s}f\left(\boldsymbol{x}^t\right)+\nabla_{\boldsymbol{x}_s}f\left(\boldsymbol{x}^t\right)\right\|^2+\sigma_n^2\right)$$

$$\leq\frac{SN\left(\eta^t\right)^2\tilde{\tau}}{2}\sum_{n=1}^N\frac{a_n^2}{q_n}\left(\tilde{\tau}+8\tilde{\tau}^3\left(\eta^t\right)^2S^2\right)\left(2\left\|\nabla_{\boldsymbol{x}_s}f\left(\boldsymbol{x}^t\right)\right\|^2+2\epsilon^2+\sigma_n^2\right),\qquad(150)$$

### F.3.2 One-round Parallel Update for Client-Side Models

The analysis of the client-side model update is similar to the server. Thus, we have

$$\mathbb{E}\left[\left\langle\nabla_{\boldsymbol{x}_c}f\left(\boldsymbol{x}^t\right),\boldsymbol{x}_c^{t+1}-\boldsymbol{x}_c^t\right\rangle\right]$$

$$\leq\left(-\frac{\eta^t\tau}{2}+8N\left(\eta^t\right)^3\tau^3S^2\sum_{n=1}^N\frac{a_n^2}{q_n}\right)\left\|\nabla_{\boldsymbol{x}_c}f\left(\boldsymbol{x}^t\right)\right\|^2+8N\eta^tS^2\tau^3\sum_{n=1}^N\frac{a_n^2}{q_n}\left(\eta^t\right)^2\left(\sigma_n^2+\epsilon^2\right).$$

$$(151)$$

For $\eta^t \leq \frac{1}{2S\tau}$,

$$\frac{S}{2}\mathbb{E}\left[\left\|\boldsymbol{x}_c^{t+1} - \boldsymbol{x}_c^t\right\|^2\right]$$

$$\leq \frac{SN\left(\eta^t\right)^2 \tau}{2} \sum_{n=1}^{N} \frac{a_n^2}{q_n}\left(\tau + 8\tau^3\left(\eta^t\right)^2 S^2\right)\left(2\left\|\nabla_{\boldsymbol{x}_c} f\left(\boldsymbol{x}^t\right)\right\|^2 + 2\epsilon^2 + \sigma_n^2\right). \qquad (152)$$

### F.3.3  Superposition of M-Server and Clients

Applying (147), (150), (152) and (151) into (36) in Proposition C.4 and define $\tau_{\min} \triangleq \min\{\tau, \tilde{\tau}\}$, $\tau_{\max} \triangleq \max\{\tau, \tilde{\tau}\}$ we have

$$\mathbb{E}\left[f\left(\boldsymbol{x}^{t+1}\right)\right] - f\left(\boldsymbol{x}^t\right)$$

$$\leq \mathbb{E}\left[\langle\nabla_{\boldsymbol{x}_c} f\left(\boldsymbol{x}^t\right), \boldsymbol{x}_c^{t+1} - \boldsymbol{x}_{c}^t\rangle\right] + \frac{S}{2}\mathbb{E}\left[\left\|\boldsymbol{x}_c^{t+1} - \boldsymbol{x}_c^t\right\|^2\right] + \mathbb{E}\left[\langle\nabla_{\boldsymbol{x}_s} f\left(\boldsymbol{x}^t\right), \boldsymbol{x}_s^{t+1} - \boldsymbol{x}_{s}^t\rangle\right] + \frac{S}{2}\mathbb{E}\left[\left\|\boldsymbol{x}_s^{t+1} - \boldsymbol{x}_s^t\right\|^2\right]$$

$$\leq \left(-\frac{\eta^t \min\{\tau, \tilde{\tau}\}}{2} + 8N\left(\eta^t\right)^3\left(\max\{\tau, \tilde{\tau}\}\right)^3 S^2 \sum_{n=1}^{N} \frac{a_n^2}{q_n}\right)\left\|\nabla_{\boldsymbol{x}} f\left(\boldsymbol{x}^t\right)\right\|^2$$

$$+ 8N\eta^t S^2\left(\tau^3 + \tilde{\tau}^3\right)\sum_{n=1}^{N}\left(\eta^t\right)^2 \frac{a_n^2}{q_n}\left(\sigma_n^2 + \epsilon^2\right)$$

$$+ \frac{SN\left(\eta^t\right)^2 \max\{\tau, \tilde{\tau}\}}{2}\sum_{n=1}^{N}\frac{a_n^2}{q_n}\left(\max\{\tau, \tilde{\tau}\} + 8\left(\max\{\tau, \tilde{\tau}\}\right)^3\left(\eta^t\right)^2 S^2\right)2\left\|\nabla_{\boldsymbol{x}} f\left(\boldsymbol{x}^t\right)\right\|^2$$

$$+ \frac{SN\left(\eta^t\right)^2 \tau}{2}\sum_{n=1}^{N}\frac{a_n^2}{q_n}\left(\tau + 8\tau^3\left(\eta^t\right)^2 S^2\right)\left(2\epsilon^2 + \sigma_n^2\right) + \frac{SN\left(\eta^t\right)^2 \tilde{\tau}}{2}\sum_{n=1}^{N}\frac{a_n^2}{q_n}\left(\tilde{\tau} + 8\tilde{\tau}^3\left(\eta^t\right)^2 S^2\right)\left(2\epsilon^2 + \sigma_n^2\right)$$

$$\leq \left(-\frac{\eta^t \tau_{\min}}{2} + 8N\left(\eta^t\right)^3 S^2 \tau_{\max}^3 \sum_{n=1}^{N}\frac{a_n^2}{q_n} + SN\left(\eta^t\right)^2 \tau_{\max}\sum_{n=1}^{N}\frac{a_n^2}{q_n}\left(\tau_{\max} + 8\tau_{\max}^3\left(\eta^t\right)^2 S^2\right)\right)\left\|\nabla_{\boldsymbol{x}} f\left(\boldsymbol{x}^t\right)\right\|^2$$

$$+ 8N\eta^t S^2\left(\tau^3 + \tilde{\tau}^3\right)\sum_{n=1}^{N}\frac{a_n^2}{q_n}\left(\eta^t\right)^2\left(\sigma_n^2 + \epsilon^2\right)$$

$$+ \frac{1}{2}SN\left(\eta^t\right)^2 \tau\left(\tau + 8\tau^3\left(\eta^t\right)^2 S^2\right)\sum_{n=1}^{N}\frac{a_n^2}{q_n}\left(2\epsilon^2 + \sigma_n^2\right)$$

$$+ \frac{1}{2}SN\left(\eta^t\right)^2 \tilde{\tau}\left(\tilde{\tau} + 8\tilde{\tau}^3\left(\eta^t\right)^2 S^2\right)\sum_{n=1}^{N}\frac{a_n^2}{q_n}\left(2\epsilon^2 + \sigma_n^2\right)$$

$$\leq \left(-\frac{\eta^t \tau_{\min}}{2} + SN\left(\eta^t\right)^2 \tau_{\max}^2 \sum_{n=1}^{N}\frac{a_n^2}{q_n} + 8N\left(\eta^t\right)^3 S^2 \tau_{\max}^3 \sum_{n=1}^{N}\frac{a_n^2}{q_n} + 8S^3 N\left(\eta^t\right)^4 \tau_{\max}^4 \sum_{n=1}^{N}\frac{a_n^2}{q_n}\right)\left\|\nabla_{\boldsymbol{x}} f\left(\boldsymbol{x}^t\right)\right\|^2$$

$$+ 8N\left(\eta^t\right)^3 S^2 \tau^3 \sum_{n=1}^{N}\frac{a_n^2}{q_n}\sigma_n^2 + 8N\left(\eta^t\right)^3 S^2 \tau^3 \epsilon^2 \sum_{n=1}^{N}\frac{a_n^2}{q_n}$$

$$+ SN\left(\eta^t\right)^2 \tau^2 \epsilon^2 \sum_{n=1}^{N}\frac{a_n^2}{q_n} + \frac{1}{2}SN\left(\eta^t\right)^2 \tilde{\tau}^2 \sum_{n=1}^{N}\frac{a_n^2}{q_n}\sigma_n^2$$

$$+ 8NS^3\left(\eta^t\right)^4 \tau^4 \epsilon^2 \sum_{n=1}^{N}\frac{a_n^2}{q_n} + 4NS^3\left(\eta^t\right)^4 \tau^4 \sum_{n=1}^{N}\frac{a_n^2}{q_n}\sigma_n^2$$

$$+ 8N\left(\eta^t\right)^3 S^2 \tilde{\tau}^3 \sum_{n=1}^{N}\frac{a_n^2}{q_n}\sigma_n^2 + 8N\left(\eta^t\right)^3 S^2 \tilde{\tau}^3 \epsilon^2 \sum_{n=1}^{N}\frac{a_n^2}{q_n}$$

$$+ SN\left(\eta^t\right)^2 \tilde{\tau}^2 \epsilon^2 \sum_{n=1}^{N}\frac{a_n^2}{q_n} + \frac{1}{2}SN\left(\eta^t\right)^2 \tilde{\tau}^2 \sum_{n=1}^{N}\frac{a_n^2}{q_n}\sigma_n^2$$

$$+8NS^3\left(\eta^t\right)^4\tilde{\tau}^4\epsilon^2\sum_{n=1}^{N}\frac{a_n^2}{q_n}+4NS^3\left(\eta^t\right)^4\tilde{\tau}^4\sum_{n=1}^{N}\frac{a_n^2}{q_n}\sigma_n^2$$

$$\leq-\frac{\eta^t\tau_{\min}}{2}\left(1-2SN\eta^t\frac{\tau_{\max}^2}{\tau_{\min}}\sum_{n=1}^{N}\frac{a_n^2}{q_n}\left(1+8S\eta^t\tau+8S^2\left(\eta^t\right)^2\tau_{\max}^2\right)\right)\left\|\nabla_{\boldsymbol{x}}f\left(\boldsymbol{x}^t\right)\right\|^2$$

$$+\left(\frac{1}{2}NS\left(\eta^t\right)^2\tau^2+8N\left(\eta^t\right)^3S^2\tau^3+4NS^3\left(\eta^t\right)^4\tau^4\right)\sum_{n=1}^{N}\frac{a_n^2}{q_n}\sigma_n^2$$

$$+\left(NS\left(\eta^t\right)^2\tau^2+8N\left(\eta^t\right)^3S^2\tau^3+8NSL^3\left(\eta^t\right)^4\tau^4\right)\sum_{n=1}^{N}\frac{a_n^2}{q_n}\epsilon^2$$

$$+\left(\frac{1}{2}NS\left(\eta^t\right)^2\tilde{\tau}^2+8N\left(\eta^t\right)^3S^2\tilde{\tau}^3+4NS^3\left(\eta^t\right)^4\tilde{\tau}^4\right)\sum_{n=1}^{N}\frac{a_n^2}{q_n}\sigma_n^2$$

$$+\left(NS\left(\eta^t\right)^2\tau^2+8N\left(\eta^t\right)^3S^2\tilde{\tau}^3+8NSL^3\left(\eta^t\right)^4\tilde{\tau}^4\right)\sum_{n=1}^{N}\frac{a_n^2}{q_n}\epsilon^2$$

$$\leq-\frac{\eta^t\tau_{\min}}{2}\left(1-2NS\eta^t\frac{\tau_{\max}^2}{\tau_{\min}}\sum_{n=1}^{N}\frac{a_n^2}{q_n}\left(1+\frac{1}{2}+\frac{1}{32}\right)\right)\left\|\nabla_{\boldsymbol{x}}f\left(\boldsymbol{x}^t\right)\right\|^2$$

$$+NS\left(\eta^t\right)^2\tau^2\left(\frac{1}{2}+\frac{1}{2}+\frac{1}{64}\right)\sum_{n=1}^{N}\frac{a_n^2}{q_n}\sigma_n^2+2SN\left(\eta^t\right)^2\tau^2\left(\frac{1}{2}+\frac{1}{4}+\frac{1}{64}\right)\sum_{n=1}^{N}\frac{a_n^2}{q_n}\epsilon^2$$

$$+NS\left(\eta^t\right)^2\tilde{\tau}^2\left(\frac{1}{2}+\frac{1}{2}+\frac{1}{64}\right)\sum_{n=1}^{N}\frac{a_n^2}{q_n}\sigma_n^2+2SN\left(\eta^t\right)^2\tilde{\tau}^2\left(\frac{1}{2}+\frac{1}{4}+\frac{1}{64}\right)\sum_{n=1}^{N}\frac{a_n^2}{q_n}\epsilon^2$$

$$\leq-\frac{\eta^t\tau_{\min}}{2}\left(1-4NS\eta^t\frac{\tau_{\max}^2}{\tau_{\min}}\sum_{n=1}^{N}\frac{a_n^2}{q_n}\right)\left\|\nabla_{\boldsymbol{x}}f\left(\boldsymbol{x}^t\right)\right\|^2+2NS\left(\eta^t\right)^2\left(\tau^2+\tilde{\tau}^2\right)\sum_{n=1}^{N}\frac{a_n^2}{q_n}\left(\sigma_n^2+\epsilon^2\right)$$

$$\leq-\frac{\eta^t\tau_{\min}}{4}\left\|\nabla_{\boldsymbol{x}}f\left(\boldsymbol{x}^t\right)\right\|^2+2NS\left(\eta^t\right)^2\left(\tau^2+\tilde{\tau}^2\right)\sum_{n=1}^{N}\frac{a_n^2}{q_n}\left(\sigma_n^2+\epsilon^2\right),\tag{153}$$

where we first let $\eta^t\leq\frac{1}{16S\tau_{\max}}$ and then let $\eta^t\leq\frac{1}{8SN\frac{\tau_{\max}^2}{\tau_{\min}}\sum_{n=1}^{N}\frac{a_n^2}{q_n}}$. We also use $\left\|\nabla_{\boldsymbol{x}}f\left(\boldsymbol{x}^t\right)\right\|^2=\left\|\nabla_{\boldsymbol{x}_c}f\left(\boldsymbol{x}^t\right)\right\|^2+\left\|\nabla_{\boldsymbol{x}_s}f\left(\boldsymbol{x}^t\right)\right\|^2$.

Rearranging the above we have

$$\eta^t\left\|\nabla_{\boldsymbol{x}}f\left(\boldsymbol{x}^t\right)\right\|^2\leq\frac{4}{\tau_{\min}}\left(f\left(\boldsymbol{x}^t\right)-\mathbb{E}\left[f\left(\boldsymbol{x}_s^{t+1}\right)\right]\right)+8NS\left(\eta^t\right)^2\frac{\tau^2+\tilde{\tau}^2}{\tau_{\min}}\sum_{n=1}^{N}\frac{a_n^2}{q_n}\left(\sigma_n^2+\epsilon^2\right)\tag{154}$$

Taking expectation and averaging over all $t$, we have

$$\frac{1}{T}\sum_{t=0}^{T-1}\eta^t\mathbb{E}\left[\left\|\nabla_{\boldsymbol{x}}f\left(\boldsymbol{x}^t\right)\right\|^2\right]\leq\frac{4}{T\tau_{\min}}\left(f\left(\boldsymbol{x}_0\right)-f^*\right)+\frac{8NS\frac{\tau^2+\tilde{\tau}^2}{\tau_{\min}}}{T}\sum_{n=1}^{N}\frac{a_n^2}{q_n}\left(\sigma_n^2+\epsilon^2\right)\sum_{t=0}^{T-1}\left(\eta^t\right)^2.\tag{155}$$

# G  Proof of Theorem 3.9

- In Sec. G.1, we prove the strongly convex case.
- In Sec. G.2, we prove the general convex case.
- In Sec. G.3, we prove the non-convex case.

## G.1  Strongly convex case for SFL-V2

### G.1.1  One-round Sequential Update for M-Server-Side Model

We first bound the M-server-side model update in one round for full participation ($q_n = 1$ for all $n$), and then compute the difference between full participation and partial participation ($q_n < 1$ for some $n$). We denote $\mathbf{I}_n^t$ as a binary variable, taking 1 if client $n$ participates in model training in round $t$, and 0 otherwise.

For full participation, Lemma E.1 gives

$$
\mathbb{E}\left[\left\|\overline{\boldsymbol{x}}_s^{t+1} - \boldsymbol{x}_s^*\right\|^2\right]
$$

$$
\leq \left(1 - \frac{N\eta^t\tau\mu}{2}\right)\mathbb{E}\left[\left\|\boldsymbol{x}_s^t - \boldsymbol{x}_s^*\right\|^2\right] - 2\eta^t\tau\mathbb{E}\left[f\left(\boldsymbol{x}^t\right) - f\left(\boldsymbol{x}^*\right)\right]
$$

$$
+ \left(\eta^t\right)^2\tau^2 N\sum_{n=1}^N\left(2\sigma_n^2 + G^2\right) + 24S\tau^3\left(\eta^t\right)^3\sum_{n=1}^N\left(2\sigma_n^2 + G^2\right). \tag{156}
$$

Considering that each client $n$ participates in model training with a probability $q_n$, we have

$$
\mathbb{E}\left[\left\|\boldsymbol{x}_s^{t+1} - \overline{\boldsymbol{x}}_s^{t+1}\right\|^2\right]
$$

$$
= \mathbb{E}\left[\left\|\boldsymbol{x}_s^{t+1} - \boldsymbol{x}_s^t + \boldsymbol{x}_s^t - \overline{\boldsymbol{x}}_s^{t+1}\right\|^2\right]
$$

$$
\leq \mathbb{E}\left[\left\|\boldsymbol{x}_s^{t+1} - \boldsymbol{x}_s^t\right\|^2\right]
$$

$$
\leq \mathbb{E}\left[\left\|\sum_{n=1}^N\eta^t\frac{\mathbf{I}_t^n}{q_n}\sum_{i=0}^{\tau-1}\boldsymbol{g}_{s,n}^{t,i}\left(\left\{\boldsymbol{x}_{c,n}^{t,i}, \boldsymbol{x}_{s,n}^{t,i}\right\}\right)\right\|^2\right]
$$

$$
\leq N\tau\sum_{n=1}^N\left(\eta^t\right)^2\frac{1}{q_n}\sum_{i=0}^{\tau-1}\mathbb{E}\left[\left\|\boldsymbol{g}_{s,n}^{t,i}\left(\left\{\boldsymbol{x}_{c,n}^{t,i}, \boldsymbol{x}_{s,n}^{t,i}\right\}\right)\right\|^2\right]
$$

$$
\leq N\tau^2\left(\eta^t\right)^2 G^2\sum_{n=1}^N\frac{1}{q_n}, \tag{157}
$$

where we use $\mathbb{E}\|X - \mathbb{E}X\|^2 \leq \mathbb{E}\|X\|^2$, $\mathbb{E}\left[\mathbf{I}_n^t\right] = q_n$, and $\boldsymbol{x}_s^{t+1} = \boldsymbol{x}_s^t - \eta^t\sum_{n\in\mathcal{P}^t(\boldsymbol{q})}\sum_{i=0}^{\tau-1}\frac{1}{q_n}\boldsymbol{g}_{s,n}^{t,i}\left(\left\{\boldsymbol{x}_{c,n}^{t,i}, \boldsymbol{x}_{s,n}^{t,i}\right\}\right)$.

Combining the above gives

$$
\mathbb{E}\left[\left\|\boldsymbol{x}_s^{t+1} - \boldsymbol{x}_s^*\right\|^2\right] = \mathbb{E}\left[\left\|\boldsymbol{x}_s^{t+1} - \overline{\boldsymbol{x}}_s^{t+1} + \overline{\boldsymbol{x}}_s^{t+1} - \boldsymbol{x}_s^*\right\|^2\right]
$$

$$
\leq \left(1 - \frac{N\eta^t\tau\mu}{2}\right)\mathbb{E}\left[\left\|\boldsymbol{x}_s^t - \boldsymbol{x}_s^*\right\|^2\right]
$$

$$
+ \left(\eta^t\right)^2\tau^2 N\sum_{n=1}^N\left(2\sigma_n^2 + G^2\right) + 24S\tau^3\left(\eta^t\right)^3\sum_{n=1}^N\left(2\sigma_n^2 + G^2\right)
$$

$$
+ N\tau^2\left(\eta^t\right)^2 G^2\sum_{n=1}^N\frac{1}{q_n}. \tag{158}
$$

Let $\Delta^{t+1} \triangleq \mathbb{E}\left[\left\|\boldsymbol{x}_s^{t+1} - \boldsymbol{x}_s^*\right\|^2\right]$. We can rewrite (158) as:

$$\Delta^{t+1} \le \left(1 - \frac{\eta^t N \tau \mu}{2}\right) \Delta^t + \frac{\left(\eta^t\right)^2 \tau^2}{4} B_1 + \frac{\left(\eta^t\right)^3 \tau^3}{8} B_2. \tag{159}$$

where $B_1 := 4N \sum_{n=1}^N \left(2\sigma_n^2 + G^2\right) + 4NG^2 \sum_{n=1}^N \frac{1}{q_n}$ and $B := 192S \sum_{n=1}^N \left(2\sigma_n^2 + G^2\right)$.

Consider a diminishing stepsize $\eta^t = \frac{2\beta}{N\tau(\gamma_s + t)}$, i.e, $\frac{N\eta^t \tau}{2} = \frac{\beta}{\gamma_s + t}$, where $\beta = \frac{2}{\mu}, \gamma_s = \frac{8S}{N\mu} - 1$. It is easy to show that $\eta^t \le \frac{1}{2S\tau}$ for all $t$. We can prove that $\Delta^t \le \frac{v}{\gamma_s + t}, \forall t$. Therefore, we have

$$\begin{aligned}
\mathbb{E}\left[\left\|\boldsymbol{x}_s^t - \boldsymbol{x}_s^*\right\|^2\right] = \Delta^t &\le \frac{v}{\gamma_s + t} = \frac{\max\left\{\frac{4B_1}{\mu^2} + \frac{8B_2}{\mu^3(\gamma_s + 1)}, (\gamma_s + 1)\mathbb{E}\left[\left\|\boldsymbol{x}_s^0 - \boldsymbol{x}_s^*\right\|^2\right]\right\}}{\gamma_s + t} \\
&\le \frac{16N \sum_{n=1}^N \left(2\sigma_n^2 + G^2\right) + 16NG^2 \sum_{n=1}^N \frac{1}{q_n}}{\mu^2 (\gamma_s + t)} + \frac{1536S \sum_{n=1}^N \left(2\sigma_n^2 + G^2\right)}{\mu^3 (\gamma_s + t) (\gamma_s + 1)} \\
&\quad + \frac{(\gamma_s + 1)\mathbb{E}\left[\left\|\boldsymbol{x}_s^0 - \boldsymbol{x}_s^*\right\|^2\right]}{\gamma_s + t}.
\end{aligned} \tag{160}$$

### G.1.2 One-round Parallel Update for Client-Side Models

Define $\overline{\boldsymbol{x}}_t^c = \sum_{n=1}^N a_n \boldsymbol{x}_{c,n}^t$, which represents the aggregating weights in round $t$ for full participation. Using a similar derivation as the M-server side, we first bound the client-side model update in one round for full participation $\mathbb{E}\left[\left\|\overline{\boldsymbol{x}}_c^{t+1} - \boldsymbol{x}_c^*\right\|^2\right]$ and then bound the difference of client-side model parameters between full participation and partial participation $\mathbb{E}\left[\left\|\boldsymbol{x}_c^{t+1} - \boldsymbol{x}_c^*\right\|^2\right]$. The overall gradient update rule of clients in each training round is $\boldsymbol{x}_c^{t+1} = \boldsymbol{x}_c^t - \eta^t \sum_{n \in \mathcal{P}^t(\boldsymbol{q})} \sum_{i=0}^{\tau-1} \frac{a_n}{q_n} \boldsymbol{g}_{c,n}^{t,i}\left(\left\{\boldsymbol{x}_{c,n}^{t,i}, \boldsymbol{x}_{s,n}^{t,i}\right\}\right)$.

Under Assumptions C.1 and C.2, if $\eta^t \le \frac{1}{2S\tau}$, in round $t$, Lemma D.1 gives

$$\begin{aligned}
&\mathbb{E}\left[\left\|\overline{\boldsymbol{x}}_c^{t+1} - \boldsymbol{x}_c^*\right\|^2\right] \\
&\le \left(1 - \frac{\eta^t \tau \mu}{2}\right) \mathbb{E}\left[\left\|\boldsymbol{x}_c^t - \boldsymbol{x}_c^*\right\|^2\right] - 2\eta^t \tau \mathbb{E}\left[f\left(\boldsymbol{x}^t\right) - f\left(\boldsymbol{x}^*\right)\right] \\
&\quad + \left(\eta^t\right)^2 \tau^2 N \sum_{n=1}^N a_n^2 \left(2\sigma_n^2 + G^2\right) + 24S\tau^3 \left(\eta^t\right)^3 \sum_{n=1}^N a_n \left(2\sigma_n^2 + G^2\right).
\end{aligned} \tag{161}$$

Considering that each client $n$ participates in model training with a probability $q_n$, we have

$$\begin{aligned}
&\mathbb{E}\left[\left\|\boldsymbol{x}_c^{t+1} - \overline{\boldsymbol{x}}_c^{t+1}\right\|^2\right] \\
&= \mathbb{E}\left[\left\|\boldsymbol{x}_c^{t+1} - \boldsymbol{x}_c^t + \boldsymbol{x}_c^t - \overline{\boldsymbol{x}}_c^{t+1}\right\|^2\right] \\
&\le \mathbb{E}\left[\left\|\boldsymbol{x}_c^{t+1} - \boldsymbol{x}_c^t\right\|^2\right] \\
&\le \mathbb{E}\left[\left\|\sum_{n=1}^N \eta^t \frac{a_n \mathbf{I}_t^n}{q_n} \sum_{i=0}^{\tau-1} \boldsymbol{g}_{c,n}^{t,i}\left(\left\{\boldsymbol{x}_{c,n}^{t,i}, \boldsymbol{x}_{s,n}^{t,i}\right\}\right)\right\|^2\right] \\
&\le N\tau \sum_{n=1}^N \left(\eta^t\right)^2 \frac{a_n^2}{q_n} \sum_{i=0}^{\tau-1} \mathbb{E}\left[\left\|\boldsymbol{g}_{c,n}^{t,i}\left(\left\{\boldsymbol{x}_{c,n}^{t,i}, \boldsymbol{x}_{s,n}^{t,i}\right\}\right)\right\|^2\right] \\
&\le N\tau^2 \left(\eta^t\right)^2 G^2 \sum_{n=1}^N \frac{a_n^2}{q_n},
\end{aligned} \tag{162}$$

where we use $\mathbb{E}\|X - \mathbb{E}X\|^2 \leq \mathbb{E}\|X\|^2$, $\mathbb{E}[\mathbf{I}_n^t] = q_n$, and $\boldsymbol{x}_c^{t+1} = \boldsymbol{x}_c^t - \eta^t \sum_{n \in \mathcal{P}^t(\boldsymbol{q})} \sum_{i=0}^{\tau-1} \frac{a_n}{q_n} \boldsymbol{g}_{c,n}^{t,i}\left(\{\boldsymbol{x}_{c,n}^{t,i}, \boldsymbol{x}_{s,n}^{t,i}\}\right)$.

We obtain the client-side model parameter update in one round for partial participation by combining the two terms and we have

$$
\begin{aligned}
\mathbb{E}\left[\left\|\boldsymbol{x}_c^{t+1} - \boldsymbol{x}_c^*\right\|^2\right] &= \mathbb{E}\left[\left\|\boldsymbol{x}_c^{t+1} - \overline{\boldsymbol{x}}_c^{t+1} + \overline{\boldsymbol{x}}_c^{t+1} - \boldsymbol{x}_c^*\right\|^2\right] \\
&\leq \left(1 - \frac{\eta^t \tau \mu}{2}\right) \mathbb{E}\left[\left\|\boldsymbol{x}_c^t - \boldsymbol{x}_c^*\right\|^2\right] \\
&\quad + \left(\eta^t\right)^2 \tau^2 N \sum_{n=1}^N a_n^2 \left(2\sigma_n^2 + G^2\right) + 24S\tau^3 \left(\eta^t\right)^3 \sum_{n=1}^N a_n \left(2\sigma_n^2 + G^2\right) \\
&\quad + N\tau^2 \left(\eta^t\right)^2 G^2 \sum_{n=1}^N \frac{a_n^2}{q_n}.
\end{aligned}
\tag{163}
$$

Let $\Delta^{t+1} \triangleq \mathbb{E}\left[\left\|\boldsymbol{x}_c^{t+1} - \boldsymbol{x}_c^*\right\|^2\right]$. We can rewrite (163) as:

$$
\Delta^{t+1} \leq \left(1 - \frac{\eta^t \tau \mu}{2}\right) \Delta^t + \frac{\left(\eta^t\right)^2 \tau^2}{4} B_1 + \frac{\left(\eta^t\right)^3 \tau^3}{8} B_2.
\tag{164}
$$

where $B_1 := 4N \sum_{n=1}^N a_n^2 \left(2\sigma_n^2 + G^2\right) + 4NG^2 \sum_{n=1}^N \frac{a_n^2}{q_n}$ and $B_2 := 192S \sum_{n=1}^N a_n \left(2\sigma_n^2 + G^2\right)$.

Consider a diminishing stepsize $\eta^t = \frac{2\beta}{\tau(\gamma_c + t)}$, i.e, $\frac{\eta^t \tau}{2} = \frac{\beta}{\gamma_c + t}$, where $\beta = \frac{2}{\mu}, \gamma_c = \frac{8S}{\mu} - 1$. It is easy to show that $\eta^t \leq \frac{1}{2S\tau}$ for all $t$. For $v = \max\left\{\frac{4B_1}{\mu^2} + \frac{8B_2}{\mu^3(\gamma_c+1)}, (\gamma_c + 1)\Delta^0\right\}$, we can prove that $\Delta^t \leq \frac{v}{\gamma_c + t}, \forall t$. Therefore, we have

$$
\begin{aligned}
\mathbb{E}\left[\left\|\boldsymbol{x}_c^t - \boldsymbol{x}_c^*\right\|^2\right] = \Delta^t &\leq \frac{v}{\gamma_c + t} = \frac{\max\left\{\frac{4B_1}{\mu^2} + \frac{8B_2}{\mu^3(\gamma_c+1)}, (\gamma_c + 1)\mathbb{E}\left[\left\|\boldsymbol{x}_c^0 - \boldsymbol{x}_c^*\right\|^2\right]\right\}}{\gamma_c + t} \\
&\leq \frac{16N \sum_{n=1}^N a_n^2 \left(2\sigma_n^2 + G^2\right) + 16NG^2 \sum_{n=1}^N \frac{a_n^2}{q_n}}{\mu^2(\gamma_c + t)} + \frac{1536S \sum_{n=1}^N a_n \left(2\sigma_n^2 + G^2\right)}{\mu^3(\gamma_c + t)(\gamma_c + 1)} \\
&\quad + \frac{(\gamma_c + 1)\mathbb{E}\left[\left\|\boldsymbol{x}_c^0 - \boldsymbol{x}_c^*\right\|^2\right]}{\gamma_c + t}.
\end{aligned}
\tag{165}
$$

### G.1.3 Superposition of M-Server and Clients

We merge the M-server-side and client-side models in (160) and (165) using Proposition 3.5. For $\eta^t \leq \frac{1}{2S\tau}$ and $\gamma = \frac{8S}{\mu} - 1$, we have

$$
\begin{aligned}
\mathbb{E}\left[f(\boldsymbol{x}^T)\right] &- f(\boldsymbol{x}^*) \\
&\leq \frac{S}{2}\left(\mathbb{E}\|\boldsymbol{x}_s^T - \boldsymbol{x}_s^*\|^2 + \mathbb{E}\|\boldsymbol{x}_c^T - \boldsymbol{x}_c^*\|^2\right) \\
&\leq \frac{8SN \sum_{n=1}^N (a_n^2 + 1)\left(2\sigma_n^2 + G^2 + \frac{G^2}{q_n}\right)}{\mu^2(\gamma + T)} + \frac{768S^2 \sum_{n=1}^N (a_n + 1)\left(2\sigma_n^2 + G^2\right)}{\mu^3(\gamma + T)(\gamma + 1)} \\
&\quad + \frac{S(\gamma + 1)\mathbb{E}\left[\left\|\boldsymbol{x}_c^0 - \boldsymbol{x}_c^*\right\|^2\right]}{2(\gamma + T)}
\end{aligned}
\tag{166}
$$

### G.2 General convex case for SFL-V2

#### G.2.1 One-round Sequential Update for M-Server-Side Model

By Lemma E.1 with $\mu = 0$ and $\eta^t \leq \frac{1}{2S\tau}$, we have

$$
\mathbb{E}\left[\left\|\boldsymbol{x}_s^{t+1} - \boldsymbol{x}_s^*\right\|^2\right]
$$
$$
\leq \mathbb{E}\left[\left\|\boldsymbol{x}_s^t - \boldsymbol{x}_s^*\right\|^2\right] - 2\eta^t\tau\mathbb{E}\left[f\left(\boldsymbol{x}^t\right) - f\left(\boldsymbol{x}^*\right)\right]
$$
$$
+ \left(\eta^t\right)^2\tau^2 N\sum_{n=1}^{N}\left(2\sigma_n^2 + G^2\right) + 24S\tau^3\left(\eta^t\right)^3\sum_{n=1}^{N}\left(2\sigma_n^2 + G^2\right). \tag{167}
$$

Considering that each client $n$ participates in model training with a probability $q_n$, we have

$$
\mathbb{E}\left[\left\|\boldsymbol{x}_s^{t+1} - \overline{\boldsymbol{x}}_s^{t+1}\right\|^2\right] \leq N\tau^2\left(\eta^t\right)^2 G^2\sum_{n=1}^{N}\frac{1}{q_n}. \tag{168}
$$

Thus, we have

$$
\mathbb{E}\left[\left\|\boldsymbol{x}_s^{t+1} - \boldsymbol{x}_s^*\right\|^2\right]
$$
$$
\leq \mathbb{E}\left[\left\|\boldsymbol{x}_s^t - \boldsymbol{x}_s^*\right\|^2\right] - 2\eta^t\tau\mathbb{E}\left[f\left(\boldsymbol{x}^t\right) - f\left(\boldsymbol{x}^*\right)\right]
$$
$$
+ \left(\eta^t\right)^2\tau^2 N\sum_{n=1}^{N}\left(2\sigma_n^2 + G^2\right) + 24S\tau^3\left(\eta^t\right)^3\sum_{n=1}^{N}\left(2\sigma_n^2 + G^2\right) + N\tau^2\left(\eta^t\right)^2 G^2\sum_{n=1}^{N}\frac{1}{q_n}. \tag{169}
$$

#### G.2.2 One-round Parallel Update for Client-Side Models

By Lemma D.1 with $\mu = 0$ and $\eta^t \leq \frac{1}{2S\tau}$, we have

$$
\mathbb{E}\left[\left\|\boldsymbol{x}_c^{t+1} - \boldsymbol{x}_c^*\right\|^2\right]
$$
$$
\leq \mathbb{E}\left[\left\|\boldsymbol{x}_c^t - \boldsymbol{x}_c^*\right\|^2\right] - 2\eta^t\tau\mathbb{E}\left[f\left(\boldsymbol{x}^t\right) - f\left(\boldsymbol{x}^*\right)\right]
$$
$$
+ \left(\eta^t\right)^2\tau^2 N\sum_{n=1}^{N}a_n^2\left(2\sigma_n^2 + G^2\right) + 24S\tau^3\left(\eta^t\right)^3\sum_{n=1}^{N}a_n\left(2\sigma_n^2 + G^2\right). \tag{170}
$$

Considering that each client $n$ participates in model training with a probability $q_n$, we have

$$
\mathbb{E}\left[\left\|\boldsymbol{x}_c^{t+1} - \overline{\boldsymbol{x}}_c^{t+1}\right\|^2\right] \leq N\tau^2\left(\eta^t\right)^2 G^2\sum_{n=1}^{N}\frac{a_n^2}{q_n}. \tag{171}
$$

Thus, we have

$$
\mathbb{E}\left[\left\|\boldsymbol{x}_c^{t+1} - \boldsymbol{x}_c^*\right\|^2\right]
$$
$$
\leq \mathbb{E}\left[\left\|\boldsymbol{x}_c^t - \boldsymbol{x}_c^*\right\|^2\right] - 2\eta^t\tau\mathbb{E}\left[f\left(\boldsymbol{x}^t\right) - f\left(\boldsymbol{x}^*\right)\right]
$$
$$
+ \left(\eta^t\right)^2\tau^2 N\sum_{n=1}^{N}a_n^2\left(2\sigma_n^2 + G^2\right) + 24S\tau^3\left(\eta^t\right)^3\sum_{n=1}^{N}a_n\left(2\sigma_n^2 + G^2\right) + N\tau^2\left(\eta^t\right)^2 G^2\sum_{n=1}^{N}\frac{a_n^2}{q_n}. \tag{172}
$$

#### G.2.3 Superposition of M-Server and Clients

We merge the M-server-side and client-side models in (169) and (172) as follows

$$
\mathbb{E}\left[\left\|\boldsymbol{x}^{t+1} - \boldsymbol{x}^*\right\|^2\right] \leq \mathbb{E}\left[\left\|\boldsymbol{x}_s^{t+1} - \boldsymbol{x}_s^*\right\|^2\right] + \mathbb{E}\left[\left\|\boldsymbol{x}_c^{t+1} - \boldsymbol{x}_c^*\right\|^2\right],
$$

$$\leq \mathbb{E}\left[\left\|\boldsymbol{x}_s^t - \boldsymbol{x}_s^*\right\|^2\right] - 2\eta^t\tau\mathbb{E}\left[f\left(\boldsymbol{x}^t\right) - f\left(\boldsymbol{x}^*\right)\right]$$

$$+ \mathbb{E}\left[\left\|\boldsymbol{x}_c^t - \boldsymbol{x}_c^*\right\|^2\right] - 2\eta^t\tau\mathbb{E}\left[f\left(\boldsymbol{x}^t\right) - f\left(\boldsymbol{x}^*\right)\right]$$

$$+ \left(\eta^t\right)^2\tau^2 N\sum_{n=1}^{N}(a_n^2 + 1)\left(2\sigma_n^2 + G^2\right) + 24S\tau^3\left(\eta^t\right)^3\sum_{n=1}^{N}(a_n + 1)\left(2\sigma_n^2 + G^2\right)$$

$$+ N\tau^2\left(\eta^t\right)^2 G^2\sum_{n=1}^{N}\frac{a_n^2 + 1}{q_n}$$

$$= \mathbb{E}\left[\left\|\boldsymbol{x}^t - \boldsymbol{x}^*\right\|^2\right] - 4\eta^t\tau\mathbb{E}\left[f\left(\boldsymbol{x}^t\right) - f\left(\boldsymbol{x}^*\right)\right]$$

$$+ \left(\eta^t\right)^2\tau^2 N\sum_{n=1}^{N}(a_n^2 + 1)\left(2\sigma_n^2 + G^2\right) + 24S\tau^3\left(\eta^t\right)^3\sum_{n=1}^{N}(a_n + 1)\left(2\sigma_n^2 + G^2\right)$$

$$+ N\tau^2\left(\eta^t\right)^2 G^2\sum_{n=1}^{N}\frac{a_n^2 + 1}{q_n}. \tag{173}$$

Then, we can obtain the relation between $\mathbb{E}\left[\left\|\boldsymbol{x}^{t+1} - \boldsymbol{x}^*\right\|^2\right]$ and $\mathbb{E}\left[\left\|\boldsymbol{x}^t - \boldsymbol{x}^*\right\|^2\right]$, which is related to $\mathbb{E}\left[f\left(\boldsymbol{x}^t\right) - f\left(\boldsymbol{x}^*\right)\right]$. Applying Lemma 8 in [17], we obtain the performance bound as

$$\mathbb{E}\left[f\left(\boldsymbol{x}^T\right)\right] - f\left(\boldsymbol{x}^*\right)$$

$$\leq \frac{1}{2}\left(N\sum_{n=1}^{N}(a_n^2 + 1)\left(2\sigma_n^2 + G^2 + \frac{G_n^2}{q_n}\right)\right)^{\frac{1}{2}}\left(\frac{\left\|\boldsymbol{x}^0 - \boldsymbol{x}^*\right\|^2}{T + 1}\right)^{\frac{1}{2}}$$

$$+ \frac{1}{2}\left(24S\sum_{n=1}^{N}(a_n + 1)\left(2\sigma_n^2 + G^2\right)\right)^{\frac{1}{3}}\left(\frac{\left\|\boldsymbol{x}^0 - \boldsymbol{x}^*\right\|^2}{T + 1}\right)^{\frac{1}{3}} + \frac{S\left\|\boldsymbol{x}^0 - \boldsymbol{x}^*\right\|^2}{2(T + 1)}. \tag{174}$$

### G.3 Non-convex case for SFL-V2

#### G.3.1 One-round Sequential Update for M-Server-Side Model

For the server, we have

$$\mathbb{E}\left[\left\langle \nabla_{\boldsymbol{x}_s} f\left(\boldsymbol{x}^t\right), \boldsymbol{x}_s^{t+1} - \boldsymbol{x}_s^t\right\rangle\right]$$

$$\leq \mathbb{E}\left[\left\langle \nabla_{\boldsymbol{x}_s} f\left(\boldsymbol{x}^t\right), \boldsymbol{x}_s^{t+1} - \boldsymbol{x}_s^t + \eta^t \tau \nabla_{\boldsymbol{x}_s} f\left(\boldsymbol{x}^t\right) - \eta^t \tau \nabla_{\boldsymbol{x}_s} f\left(\boldsymbol{x}^t\right)\right\rangle\right]$$

$$\leq \mathbb{E}\left[\left\langle \nabla_{\boldsymbol{x}_s} f\left(\boldsymbol{x}^t\right), \boldsymbol{x}_s^{t+1} - \boldsymbol{x}_s^t + \eta^t \tau \nabla_{\boldsymbol{x}_s} f\left(\boldsymbol{x}^t\right)\right\rangle - \left\langle \nabla_{\boldsymbol{x}_s} f\left(\boldsymbol{x}_s^t\right), \eta^t \tau \nabla_{\boldsymbol{x}_s} f\left(\boldsymbol{x}^t\right)\right\rangle\right]$$

$$\leq \left\langle \nabla_{\boldsymbol{x}_s} f\left(\boldsymbol{x}^t\right), \mathbb{E}\left[-\eta^t \sum_{n=1}^{N} \sum_{i=0}^{\tau-1} \frac{\mathbf{I}_n^t}{q_n} \boldsymbol{g}_{s,n}^{t,i}\right] + \eta^t \tau \nabla_{\boldsymbol{x}_s} f\left(\boldsymbol{x}^t\right)\right\rangle - \eta^t \tau \left\|\nabla_{\boldsymbol{x}_s} f\left(\boldsymbol{x}^t\right)\right\|^2$$

$$\leq \left\langle \nabla_{\boldsymbol{x}_s} f\left(\boldsymbol{x}^t\right), \mathbb{E}\left[-\eta^t \sum_{n=1}^{N} \sum_{i=0}^{\tau-1} \frac{\mathbf{I}_n^t}{q_n} \nabla_{\boldsymbol{x}_s} F_n\left(\left\{\boldsymbol{x}_{c,n}^{t,i}, \boldsymbol{x}_{s,n}^{t,i}\right\}\right)\right] + \eta^t \tau \nabla_{\boldsymbol{x}_s} f\left(\boldsymbol{x}^t\right)\right\rangle - \eta^t \tau \left\|\nabla_{\boldsymbol{x}_s} f\left(\boldsymbol{x}^t\right)\right\|^2$$

$$\leq \left\langle \nabla_{\boldsymbol{x}_s} f\left(\boldsymbol{x}^t\right), \mathbb{E}\left[-\eta^t \sum_{n=1}^{N} \sum_{i=0}^{\tau-1} \frac{\mathbf{I}_n^t}{q_n} \nabla_{\boldsymbol{x}_s} F_n\left(\left\{\boldsymbol{x}_{c,n}^{t,i}, \boldsymbol{x}_{s,n}^{t,i}\right\}\right) + \eta^t \sum_{n=1}^{N} \sum_{i=0}^{\tau-1} \frac{\mathbf{I}_n^t}{q_n} \nabla_{\boldsymbol{x}_s} F_n\left(\boldsymbol{x}^t\right)\right]\right\rangle$$
$$- \eta^t \tau \left\|\nabla_{\boldsymbol{x}_s} f\left(\boldsymbol{x}^t\right)\right\|^2$$

$$\leq \eta^t \tau \left\langle \nabla_{\boldsymbol{x}_s} f\left(\boldsymbol{x}^t\right), \mathbb{E}\left[-\frac{1}{\tau} \sum_{n=1}^{N} \sum_{i=0}^{\tau-1} \frac{\mathbf{I}_n^t}{q_n} \nabla_{\boldsymbol{x}_s} F_n\left(\left\{\boldsymbol{x}_{c,n}^{t,i}, \boldsymbol{x}_{s,n}^{t,i}\right\}\right) + \frac{1}{\tau} \sum_{n=1}^{N} \sum_{i=0}^{\tau-1} \frac{\mathbf{I}_n^t}{q_n} \nabla_{\boldsymbol{x}_s} F_n\left(\boldsymbol{x}^t\right)\right]\right\rangle$$
$$- \eta^t \tau \left\|\nabla_{\boldsymbol{x}_s} f\left(\boldsymbol{x}^t\right)\right\|^2$$

$$\leq \frac{\eta^t \tau}{2} \left\|\nabla_{\boldsymbol{x}_s} f\left(\boldsymbol{x}^t\right)\right\|^2 + \frac{\eta^t}{2\tau} \mathbb{E}\left[\left\|\sum_{n=1}^{N} \sum_{i=0}^{\tau-1} \frac{\mathbf{I}_n^t}{q_n} \nabla_{\boldsymbol{x}_s} F_n\left(\left\{\boldsymbol{x}_{c,n}^{t,i}, \boldsymbol{x}_{s,n}^{t,i}\right\}\right) - \sum_{n=1}^{N} \sum_{i=0}^{\tau-1} \frac{\mathbf{I}_n^t}{q_n} \nabla_{\boldsymbol{x}_s} F_n\left(\boldsymbol{x}^t\right)\right\|^2\right]$$
$$- \eta^t \tau \left\|\nabla_{\boldsymbol{x}_s} f\left(\boldsymbol{x}^t\right)\right\|^2$$

$$\leq -\frac{\eta^t \tau}{2} \left\|\nabla_{\boldsymbol{x}_s} f\left(\boldsymbol{x}^t\right)\right\|^2 + \frac{\eta^t}{2\tau} \mathbb{E}\left[\left\|\sum_{n=1}^{N} \sum_{i=0}^{\tau-1} \frac{\mathbf{I}_n^t}{q_n} \left(\nabla_{\boldsymbol{x}_s} F_n\left(\left\{\boldsymbol{x}_{c,n}^{t,i}, \boldsymbol{x}_{s,n}^{t,i}\right\}\right) - \nabla_{\boldsymbol{x}_s} F_n\left(\boldsymbol{x}^t\right)\right)\right\|^2\right]$$

$$\leq -\frac{\eta^t \tau}{2} \left\|\nabla_{\boldsymbol{x}_s} f\left(\boldsymbol{x}^t\right)\right\|^2 + \frac{N\eta^t}{2\tau} \sum_{n=1}^{N} \frac{1}{q_n} \mathbb{E}\left[\left\|\sum_{i=0}^{\tau-1} \left(\nabla_{\boldsymbol{x}_s} F_n\left(\left\{\boldsymbol{x}_{c,n}^{t,i}, \boldsymbol{x}_{s,n}^{t,i}\right\}\right) - \nabla_{\boldsymbol{x}_s} F_n\left(\boldsymbol{x}^t\right)\right)\right\|^2\right]$$

$$\leq -\frac{\eta^t \tau}{2} \left\|\nabla_{\boldsymbol{x}_s} f\left(\boldsymbol{x}^t\right)\right\|^2 + \frac{N\eta^t S^2}{2} \sum_{n=1}^{N} \frac{1}{q_n} \sum_{i=0}^{\tau-1} \mathbb{E}\left[\left\|\boldsymbol{x}_{s,n}^{t,i} - \boldsymbol{x}_s^t\right\|^2\right], \tag{175}$$

where we apply Assumption C.1, $\nabla_{\boldsymbol{x}_s} f\left(\boldsymbol{x}^t\right) = \sum_{n=1}^{N} \nabla_{\boldsymbol{x}_s} F_n\left(\boldsymbol{x}^t\right)$, $\langle a, b\rangle \leq \frac{a^2+b^2}{2}$, and $\mathbb{E}\left[\mathbf{I}_n^t\right] = q_n$.

By Lemma C.6 with $\eta^t \leq \frac{1}{\sqrt{8}S\tau}$, we have

$$\sum_{i=0}^{\tau-1} \mathbb{E}\left[\left\|\boldsymbol{x}_{s,n}^{t,i} - \boldsymbol{x}_s^t\right\|^2\right] \leq 2\tau^2 \left(8\tau \left(\eta^t\right)^2 \sigma_n^2 + 8\tau \left(\eta^t\right)^2 \epsilon^2 + 8\tau \left(\eta^t\right)^2 \left\|\nabla_{\boldsymbol{x}_s} f\left(\boldsymbol{x}_s^t\right)\right\|^2\right) \tag{176}$$

Thus, (175) becomes

$$\mathbb{E}\left[\left\langle \nabla_{\boldsymbol{x}_s} f\left(\boldsymbol{x}^t\right), \boldsymbol{x}_s^{t+1} - \boldsymbol{x}_s^t\right\rangle\right]$$

$$\leq -\frac{\eta^t \tau}{2} \left\|\nabla_{\boldsymbol{x}_s} f\left(\boldsymbol{x}^t\right)\right\|^2 + \frac{N\eta^t S^2}{2} \sum_{n=1}^{N} \frac{1}{q_n} 2\tau^2 \left(8\tau \left(\eta^t\right)^2 \sigma_n^2 + 8\tau \left(\eta^t\right)^2 \epsilon^2 + 8\tau \left(\eta^t\right)^2 \left\|\nabla_{\boldsymbol{x}_s} f\left(\boldsymbol{x}_s^t\right)\right\|^2\right)$$

$$\leq \left(-\frac{\eta^t \tau}{2} + 8N \left(\eta^t\right)^3 \tau^3 S^2 \sum_{n=1}^{N} \frac{1}{q_n}\right) \left\|\nabla_{\boldsymbol{x}_s} f\left(\boldsymbol{x}^t\right)\right\|^2 + 8N\eta^t S^2 \tau^3 \sum_{n=1}^{N} \frac{1}{q_n} \left(\eta^t\right)^2 \left(\sigma_n^2 + \epsilon^2\right).$$
$$\tag{177}$$

Furthermore, we have

$$\frac{S}{2}\mathbb{E}\left[\left\|\boldsymbol{x}_s^{t+1} - \boldsymbol{x}_s^t\right\|^2\right]$$

$$= \frac{SN\left(\eta^t\right)^2}{2}\sum_{n=1}^N \mathbb{E}\left[\left\|\sum_{i=0}^{\tau-1}\frac{\mathbf{I}_n^t}{q_n}\boldsymbol{g}_{s,n}^{t,i}\right\|^2\right]$$

$$\le \frac{SN\left(\eta^t\right)^2}{2}\sum_{n=1}^N\frac{1}{q_n}\mathbb{E}\left[\left\|\sum_{i=0}^{\tau-1}\boldsymbol{g}_{s,n}^{t,i}\right\|^2\right]$$

$$\le \frac{SN\left(\eta^t\right)^2\tau}{2}\sum_{n=1}^N\frac{1}{q_n}\sum_{i=0}^{\tau-1}\mathbb{E}\left[\left\|\boldsymbol{g}_{s,n}^{t,i}\right\|^2\right]$$

$$\le \frac{SN\left(\eta^t\right)^2\tau}{2}\sum_{n=1}^N\frac{1}{q_n}\sum_{i=0}^{\tau-1}\mathbb{E}\left[\left\|\boldsymbol{g}_{s,n}^{t,i} - \boldsymbol{g}_{s,n}^t + \boldsymbol{g}_{s,n}^t\right\|^2\right]$$

$$\le \frac{SN\left(\eta^t\right)^2\tau}{2}\sum_{n=1}^N\frac{1}{q_n}\sum_{i=0}^{\tau-1}\left(\mathbb{E}\left[\left\|\boldsymbol{g}_{s,n}^{t,i} - \boldsymbol{g}_{s,n}^t\right\|^2\right] + \mathbb{E}\left[\left\|\boldsymbol{g}_{s,n}^t\right\|^2\right]\right)$$

$$\le \frac{SN\left(\eta^t\right)^2\tau}{2}\sum_{n=1}^N\frac{1}{q_n}\sum_{i=0}^{\tau-1}\left(\mathbb{E}\left[\left\|\boldsymbol{g}_{s,n}^{t,i} - \boldsymbol{g}_{s,n}^t\right\|^2\right] + \mathbb{E}\left[\left\|\nabla_{\boldsymbol{x}_s}F_n\left(\boldsymbol{x}^t\right)\right\|^2 + \sigma_n^2\right]\right), \qquad (178)$$

where the last line uses Assumption C.1 and $\mathbb{E}\left[\|\mathbf{z}\|^2\right] = \|\mathbb{E}[\mathbf{z}]\|^2 + \mathbb{E}[\|\mathbf{z} - \mathbb{E}[\mathbf{z}]\|^2]$ for any random variable $\mathbf{z}$.

By Lemma C.7 with $\eta^t \le \frac{1}{2S\tau}$, we have

$$\sum_{i=0}^{\tau-1}\mathbb{E}\left[\left\|\boldsymbol{g}_{s,n}^{t,i} - \boldsymbol{g}_{s,n}^t\right\|^2\right] \le 8\tau^3\left(\eta^t\right)^2 S^2\left(\left\|\nabla_{\boldsymbol{x}_s}F_n\left(\boldsymbol{x}^t\right)\right\|^2 + \sigma_n^2\right). \qquad (179)$$

Thus, (178) becomes

$$\frac{S}{2}\mathbb{E}\left[\left\|\boldsymbol{x}_s^{t+1} - \boldsymbol{x}_s^t\right\|^2\right]$$

$$\le \frac{SN\left(\eta^t\right)^2\tau}{2}\sum_{n=1}^N\frac{1}{q_n}\left(8\tau^3\left(\eta^t\right)^2 S^2\left(\left\|\nabla_{\boldsymbol{x}_s}F_n\left(\boldsymbol{x}^t\right)\right\|^2 + \sigma_n^2\right) + \tau\mathbb{E}\left[\left\|\nabla_{\boldsymbol{x}_s}F_n\left(\boldsymbol{x}^t\right)\right\|^2 + \sigma_n^2\right]\right)$$

$$\le \frac{SN\left(\eta^t\right)^2\tau}{2}\sum_{n=1}^N\frac{1}{q_n}\left(\tau + 8\tau^3\left(\eta^t\right)^2 S^2\right)\left(\left\|\nabla_{\boldsymbol{x}_s}F_n\left(\boldsymbol{x}^t\right)\right\|^2 + \sigma_n^2\right)$$

$$\le \frac{SN\left(\eta^t\right)^2\tau}{2}\sum_{n=1}^N\frac{1}{q_n}\left(\tau + 8\tau^3\left(\eta^t\right)^2 S^2\right)\left(\left\|\nabla_{\boldsymbol{x}_s}F_n\left(\boldsymbol{x}^t\right) - \nabla_{\boldsymbol{x}_s}f\left(\boldsymbol{x}^t\right) + \nabla_{\boldsymbol{x}_s}f\left(\boldsymbol{x}^t\right)\right\|^2 + \sigma_n^2\right)$$

$$\le \frac{SN\left(\eta^t\right)^2\tau}{2}\sum_{n=1}^N\frac{1}{q_n}\left(\tau + 8\tau^3\left(\eta^t\right)^2 S^2\right)\left(2\left\|\nabla_{\boldsymbol{x}_s}f\left(\boldsymbol{x}^t\right)\right\|^2 + 2\epsilon^2 + \sigma_n^2\right). \qquad (180)$$

### G.3.2 One-round Parallel Update for Client-Side Models

The analysis of the client-side model update is the same as the client's model update in version 1. Thus, we have

$$\mathbb{E}\left[\left\langle\nabla_{\boldsymbol{x}_c}f\left(\boldsymbol{x}^t\right), \boldsymbol{x}_c^{t+1} - \boldsymbol{x}_c^t\right\rangle\right]$$

$$\le \left(-\frac{\eta^t\tau}{2} + 8N\left(\eta^t\right)^3\tau^3 S^2\sum_{n=1}^N\frac{a_n^2}{q_n}\right)\left\|\nabla_{\boldsymbol{x}_c}f\left(\boldsymbol{x}^t\right)\right\|^2 + 8N\eta^t S^2\tau^3\sum_{n=1}^N\frac{a_n^2}{q_n}\left(\eta^t\right)^2\left(\sigma_n^2 + \epsilon^2\right). \qquad (181)$$

For $\eta^t \leq \frac{1}{2S\tau}$,

$$\frac{S}{2}\mathbb{E}\left[\left\|\boldsymbol{x}_c^{t+1} - \boldsymbol{x}_c^t\right\|^2\right]$$

$$\leq \frac{SN\left(\eta^t\right)^2 \tau}{2} \sum_{n=1}^{N} \frac{a_n^2}{q_n} \left(\tau + 8\tau^3 \left(\eta^t\right)^2 S^2\right) \left(2\left\|\nabla_{\boldsymbol{x}_c} f\left(\boldsymbol{x}^t\right)\right\|^2 + 2\epsilon^2 + \sigma_n^2\right). \qquad (182)$$

### G.3.3  Superposition of M-Server and Clients

Applying (177), (180), (182) and (181) into (36) in Proposition C.4, we have

$$\mathbb{E}\left[f\left(\boldsymbol{x}^{t+1}\right)\right] - f\left(\boldsymbol{x}^t\right)$$

$$\leq \mathbb{E}\left[\langle\nabla_{\boldsymbol{x}_c} f\left(\boldsymbol{x}^t\right), \boldsymbol{x}_c^{t+1} - \boldsymbol{x}_c^t\rangle\right] + \frac{S}{2}\mathbb{E}\left[\left\|\boldsymbol{x}_c^{t+1} - \boldsymbol{x}_c^t\right\|^2\right] + \mathbb{E}\left[\langle\nabla_{\boldsymbol{x}_s} f\left(\boldsymbol{x}^t\right), \boldsymbol{x}_s^{t+1} - \boldsymbol{x}_s^t\rangle\right] + \frac{S}{2}\mathbb{E}\left[\left\|\boldsymbol{x}_s^{t+1} - \boldsymbol{x}_s^t\right\|^2\right]$$

$$\leq \left(-\frac{\eta^t \tau}{2} + 8N\left(\eta^t\right)^3 \tau^3 S^2 \sum_{n=1}^{N} \frac{1}{q_n}\right) \left\|\nabla_{\boldsymbol{x}} f\left(\boldsymbol{x}^t\right)\right\|^2$$

$$+ 8N\eta^t S^2 \tau^3 \sum_{n=1}^{N} \frac{a_n^2 + 1}{q_n} \left(\eta^t\right)^2 \left(\sigma_n^2 + \epsilon^2\right)$$

$$+ \frac{SN\left(\eta^t\right)^2 \tau}{2} \sum_{n=1}^{N} \frac{1}{q_n} \left(\tau + 8\tau^3 \left(\eta^t\right)^2 S^2\right) 2\left\|\nabla_{\boldsymbol{x}} f\left(\boldsymbol{x}^t\right)\right\|^2$$

$$+ \frac{SN\left(\eta^t\right)^2 \tau}{2} \sum_{n=1}^{N} \frac{a_n^2 + 1}{q_n} \left(\tau + 8\tau^3 \left(\eta^t\right)^2 S^2\right) \left(2\epsilon^2 + \sigma_n^2\right)$$

$$\leq \left(-\frac{\eta^t \tau}{2} + 8N\left(\eta^t\right)^3 S^2 \tau^3 \sum_{n=1}^{N} \frac{1}{q_n} + SN\left(\eta^t\right)^2 \tau \sum_{n=1}^{N} \frac{1}{q_n} \left(\tau + 8\tau^3 \left(\eta^t\right)^2 S^2\right)\right) \left\|\nabla_{\boldsymbol{x}} f\left(\boldsymbol{x}^t\right)\right\|^2$$

$$+ 8N\eta^t S^2 \tau^3 \sum_{n=1}^{N} \frac{a_n^2 + 1}{q_n} \left(\eta^t\right)^2 \left(\sigma_n^2 + \epsilon^2\right)$$

$$+ \frac{1}{2}SN\left(\eta^t\right)^2 \tau \left(\tau + 8\tau^3 \left(\eta^t\right)^2 S^2\right) \sum_{n=1}^{N} \frac{a_n^2 + 1}{q_n} \left(2\epsilon^2 + \sigma_n^2\right)$$

$$\leq \left(-\frac{\eta^t \tau}{2} + SN\left(\eta^t\right)^2 \tau^2 \sum_{n=1}^{N} \frac{1}{q_n} + 8N\left(\eta^t\right)^3 S^2 \tau^3 \sum_{n=1}^{N} \frac{1}{q_n} + 8S^3 N\left(\eta^t\right)^4 \tau^4 \sum_{n=1}^{N} \frac{1}{q_n}\right) \left\|\nabla_{\boldsymbol{x}} f\left(\boldsymbol{x}^t\right)\right\|^2$$

$$+ 8N\left(\eta^t\right)^3 S^2 \tau^3 \sum_{n=1}^{N} \frac{a_n^2 + 1}{q_n}\sigma_n^2 + 8N\left(\eta^t\right)^3 S^2 \tau^3 \epsilon^2 \sum_{n=1}^{N} \frac{a_n^2 + 1}{q_n}$$

$$+ SN\left(\eta^t\right)^2 \tau^2 \epsilon^2 \sum_{n=1}^{N} \frac{a_n^2 + 1}{q_n} + \frac{1}{2}SN\left(\eta^t\right)^2 \tau^2 \sum_{n=1}^{N} \frac{a_n^2 + 1}{q_n}\sigma_n^2$$

$$+ 8NS^3 \left(\eta^t\right)^4 \tau^4 \epsilon^2 \sum_{n=1}^{N} \frac{a_n^2 + 1}{q_n} + 4NS^3 \left(\eta^t\right)^4 \tau^4 \sum_{n=1}^{N} \frac{a_n^2 + 1}{q_n}\sigma_n^2$$

$$\leq -\frac{\eta^t \tau}{2} \left(1 - 2SN\eta^t \frac{\tau^2}{\tau} \sum_{n=1}^{N} \frac{1}{q_n} \left(1 + 8S\eta^t \tau + 8S^2 \left(\eta^t\right)^2 \tau^2\right)\right) \left\|\nabla_{\boldsymbol{x}} f\left(\boldsymbol{x}^t\right)\right\|^2$$

$$+ \left(\frac{1}{2}NS\left(\eta^t\right)^2 \tau^2 + 8N\left(\eta^t\right)^3 S^2 \tau^3 + 4NS^3 \left(\eta^t\right)^4 \tau^4\right) \sum_{n=1}^{N} \frac{a_n^2 + 1}{q_n}\sigma_n^2$$

$$+ \left(NS\left(\eta^t\right)^2 \tau^2 + 8N\left(\eta^t\right)^3 S^2 \tau^3 + 8NSL^3 \left(\eta^t\right)^4 \tau^4\right) \sum_{n=1}^{N} \frac{a_n^2 + 1}{q_n}\epsilon^2$$

$$\leq -\frac{\eta^t \tau}{2} \left(1 - 2NS\eta^t \frac{\tau^2}{\tau} \sum_{n=1}^{N} \frac{1}{q_n} \left(1 + \frac{1}{2} + \frac{1}{32}\right)\right) \left\|\nabla_{\boldsymbol{x}} f\left(\boldsymbol{x}^t\right)\right\|^2$$

$$+ NS\left(\eta^t\right)^2 \tau^2 \left(\frac{1}{2} + \frac{1}{2} + \frac{1}{64}\right) \sum_{n=1}^{N} \frac{a_n^2 + 1}{q_n} \sigma_n^2 + 2SN\left(\eta^t\right)^2 \tau^2 \left(\frac{1}{2} + \frac{1}{4} + \frac{1}{64}\right) \sum_{n=1}^{N} \frac{a_n^2 + 1}{q_n} \epsilon^2$$

$$\leq -\frac{\eta^t \tau}{2} \left(1 - 4N^2 S\eta^t \frac{\tau^2}{\tau} \sum_{n=1}^{N} \frac{1}{q_n}\right) \left\|\nabla_{\boldsymbol{x}} f\left(\boldsymbol{x}^t\right)\right\|^2 + 2NS\left(\eta^t\right)^2 \tau^2 \sum_{n=1}^{N} \frac{a_n^2 + 1}{q_n}\left(\sigma_n^2 + \epsilon^2\right)$$

$$\leq -\frac{\eta^t \tau}{4} \left\|\nabla_{\boldsymbol{x}} f\left(\boldsymbol{x}^t\right)\right\|^2 + 2NS\left(\eta^t\right)^2 \tau^2 \sum_{n=1}^{N} \frac{a_n^2 + 1}{q_n}\left(\sigma_n^2 + \epsilon^2\right), \tag{183}$$

where we first let $\eta^t \leq \frac{1}{16S\tau}$ and then let $\eta^t \leq \frac{1}{8SN^2\tau \sum_{n=1}^{N} \frac{1}{q_n}}$. We have applied $\sum_{n=1}^{N} a_n^2 \leq N$.

Rearranging the above we have

$$\eta^t \left\|\nabla_{\boldsymbol{x}} f\left(\boldsymbol{x}^t\right)\right\|^2 \leq \frac{4}{\tau}\left(f\left(\boldsymbol{x}^t\right) - \mathbb{E}\left[f\left(\boldsymbol{x}_s^{t+1}\right)\right]\right) + 8NS\left(\eta^t\right)^2 \tau \sum_{n=1}^{N} \frac{a_n^2 + 1}{q_n}\left(\sigma_n^2 + \epsilon^2\right). \tag{184}$$

Taking expectation and averaging over all $t$, we have

$$\frac{1}{T} \sum_{t=0}^{T-1} \eta^t \mathbb{E}\left[\left\|\nabla_{\boldsymbol{x}} f\left(\boldsymbol{x}^t\right)\right\|^2\right] \leq \frac{4}{T\tau}\left(f\left(\boldsymbol{x}_0\right) - f^*\right) + \frac{8NS\tau}{T} \sum_{n=1}^{N} \frac{a_n^2 + 1}{q_n}\left(\sigma_n^2 + \epsilon^2\right) \sum_{t=0}^{T-1}\left(\eta^t\right)^2. \tag{185}$$

# H  Comparative Analysis

## H.1  Main technical results

We conclude the convergence results in our paper in Table 1.

Table 1: Performance upper bounds for different objectives (let $Q := \sum_{n=1}^{N} \frac{1}{q_n}$).

| Scenario | Case | Method | Convergence result |
|---|---|---|---|
| Full participation | Strongly convex | SFL-V1 | $\frac{S}{\mu(S+\mu T)}(N(\sigma^2 + G^2) + \mu S I^{\mathrm{err}})$ |
| | | SFL-V2 | $\frac{S}{\mu(S+\mu T)}(N^2(\sigma^2 + G^2) + \mu S I^{\mathrm{err}})$ |
| | General convex | SFL-V1 | $(\frac{N(\sigma^2+G^2)}{T})^{\frac{1}{2}} + (\frac{N(\sigma^2+G^2)}{T})^{\frac{1}{3}} + \frac{S}{T}I^{\mathrm{err}}$ |
| | | SFL-V2 | $(\frac{N^2(\sigma^2+G^2)}{T})^{\frac{1}{2}} + (\frac{N^2(\sigma^2+G^2)}{T})^{\frac{1}{3}} + \frac{S}{T}I^{\mathrm{err}}$ |
| | Non-convex | SFL-V1 | $\frac{NS(\sigma^2+\epsilon^2)}{T} + \frac{\boldsymbol{F}^{err}}{T}$ |
| | | SFL-V2 | $\frac{N^2 S(\sigma^2+\epsilon^2)}{T} + \frac{\boldsymbol{F}^{err}}{T}$ |
| Partial participation | Strongly convex | SFL-V1 | $\frac{S}{\mu(S+\mu T)}(N(\sigma^2 + G^2(1+Q)) + \mu S I^{\mathrm{err}})$ |
| | | SFL-V2 | $\frac{S}{\mu(S+\mu T)}(N^2(\sigma^2 + G^2(1+Q)) + \mu S I^{\mathrm{err}})$ |
| | General convex | SFL-V1 | $(\frac{N(\sigma^2+G^2(1+Q))}{T})^{\frac{1}{2}} + (\frac{N(\sigma^2+G^2)}{T})^{\frac{1}{3}} + \frac{S}{T}I^{\mathrm{err}}$ |
| | | SFL-V2 | $(\frac{N^2(\sigma^2+G^2(1+Q))}{T})^{\frac{1}{2}} + (\frac{N^2(\sigma^2+G^2)}{T})^{\frac{1}{3}} + \frac{S}{T}I^{\mathrm{err}}$ |
| | Non-convex | SFL-V1 | $\frac{NS(\sigma^2+\epsilon^2)Q}{T} + \frac{\boldsymbol{F}^{err}}{T}$ |
| | | SFL-V2 | $\frac{N^2 S(\sigma^2+\epsilon^2)Q}{T} + \frac{\boldsymbol{F}^{err}}{T}$ |

## H.2  Comparison of Bounds

We compare our derived bounds for SFL to other distributed approaches. For simplicity, we let $a_n = 1/N$ in (1) and $\sigma_n = \sigma$ for all $n$ in (6). The result are summarized in Table 2. Since different convergence theories make slightly different assumptions, we clarify them below.

In [33], $\sigma_*^2$ is the variance of the stochastic gradient at the optimum: $\mathbb{E}_{\zeta_n \sim \mathcal{D}_n}\left[\|\boldsymbol{g}_n(\boldsymbol{x}^*, \zeta_n) - \nabla F_n(\boldsymbol{x}^*)\|^2\right] \leq \sigma_*^2$. In [16], $\Gamma = f^* - \sum_{n=1}^{N} F_n^*/N$ characterizes the client heterogeneity. In [17], $\epsilon_*^2$ characterizes the client heterogeneity at the optimum, similar to [12], i.e., $\frac{1}{N}\sum_{n=1}^{N}\|\nabla F_n(\boldsymbol{x}^*)\|^2 = \epsilon_*^2$.

Table 2: Performance upper bounds for strongly convex objectives with full client participation. Here, absolute constants and polylogarithmic factors are omitted. We further relax the upper bounds of SFL-V2 for an easier comparison.

| Method | Performance upper bound |
|---|---|
| Mini-Batch SGD [33] | $\frac{\sigma_*^2}{\mu N \tau T} + \frac{S(f(\boldsymbol{x}^0)-f^*)}{\mu}\exp\left(\frac{-\mu T}{S}\right)$ |
| FL | |
| [16] | $\frac{S}{\mu\tau+T}\left(\frac{\sigma^2+S\Gamma N + N\tau^2 G^2}{\mu N} + S I^{\mathrm{err}}\right)$ |
| [10] | $\frac{\sigma^2}{\mu N \tau T} + \frac{S\sigma^2}{\mu^2 \tau T^2} + \frac{S\epsilon^2}{\mu^2 T^2} + \mu I^{\mathrm{err}}\exp\left(\frac{-\mu T}{S}\right)$ |
| SL [17] | $\frac{\sigma^2}{\mu N \tau T} + \frac{S\sigma^2}{\mu^2 N\tau T^2} + \frac{S\epsilon_*^2}{\mu^2 N T^2} + \mu I^{\mathrm{err}}\exp\left(\frac{-\mu T}{S}\right)$ |
| SFL | |
| SFL-V1 (Theorem 3.6) | $\frac{S}{\mu(S+\mu T)}\left(N(\sigma^2 + G^2) + \mu S I^{\mathrm{err}}\right)$ |
| SFL-V2 (Theorem 3.7) | $\frac{S}{\mu(S+\mu T)}\left(N^2(\sigma^2 + G^2) + \mu S I^{\mathrm{err}}\right)$ |

The key observation is that our derived bounds match the other distributed approaches in the order of $T$ and they all achieve $O(1/T)$.

Furthermore, we will compare the convergence upper bounds of SFL to those of distributed SGD in [12] (with parameters $p = 1$ and $\bar{\zeta}^2 = 0$) as follows:

Table 3: Performance upper bounds for different objectives with full client participation.

| Case | Method | Convergence results |
|---|---|---|
| Strongly convex | Distributed SGD | $\frac{\sigma^2}{\mu NT} + S\boldsymbol{I}^{err} \exp\left(-\frac{\mu T}{S}\right)$ |
| | SFL-V1 | $\frac{S}{\mu(S+\mu T)}(N(\sigma^2 + G^2) + \mu SI^{\text{err}})$ |
| | SFL-V2 | $\frac{S}{\mu(S+\mu T)}(N^2(\sigma^2 + G^2) + \mu SI^{\text{err}})$ |
| General convex | Distributed SGD | $(\frac{\sigma^2}{NT^2} + \frac{S}{T})\boldsymbol{I}^{err}$ |
| | SFL-V1 | $(\frac{N(\sigma^2+G^2)}{T})^{\frac{1}{2}} + (\frac{N(\sigma^2+G^2)}{T})^{\frac{1}{3}} + \frac{S}{T}I^{\text{err}}$ |
| | SFL-V2 | $(\frac{N^2(\sigma^2+G^2)}{T})^{\frac{1}{2}} + (\frac{N^2(\sigma^2+G^2)}{T})^{\frac{1}{3}} + \frac{S}{T}I^{\text{err}}$ |
| Non-convex | Distributed SGD | $(\frac{\sigma^2}{NT^2} + \frac{1}{T})S\boldsymbol{F}^{err}$ |
| | SFL-V1 | $\frac{NS(\sigma^2+\epsilon^2)}{T} + \frac{\boldsymbol{F}^{err}}{T}$ |
| | SFL-V2 | $\frac{N^2 S(\sigma^2+\epsilon^2)}{T} + \frac{\boldsymbol{F}^{err}}{T}$ |

In the aforementioned table, we denote $\sigma_n = \sigma$, define $I^{\text{err}} \triangleq \|\boldsymbol{x}^0 - \boldsymbol{x}^*\|^2$, and represent $\boldsymbol{F}^{err} \triangleq f(\boldsymbol{x}^0) - f^*$. The absolute constants and polylogarithmic factors are omitted for brevity. Our SFL algorithms show the same convergence rate as the distributed SGD.

## H.3 Comparison of Communication and Computation Overheads

There have been are some papers discussing the overhead of SFL, e.g., [27, 4]. We mainly use the analysis from [27].

We start with the definitions. Let $K$ represent the total number of clients involved, $D$ denote the aggregate size of the data, and $v$ indicate the size of the smashed layer. The rate of communication is given by $R$, while $T_{\text{fb}}$ signifies the duration required for a complete forward and backward propagation cycle on the entire model for a dataset of size $D$, applicable across various architectures. The time needed to aggregate the full model is expressed as $T_{\text{fedavg}}$. The full model's size is denoted by $|W|$, and $r$ reflects the proportion of the full model's size that is accessible to a client in SFL, specifically, $|W_C| = r|W|$. The factor $2r|W|$ in the communication per client arises from the necessity for clients to download and upload their model updates before and after the training process. These findings are encapsulated in Table 4. It is observed that as $K$ escalates, the cumulative cost of training time tends to rise following the sequence: SFL-V2 being less than SFL-V1.

Table 4: Communication and computation comparison between FL, SL, and SFL.

| Method | Communication per client | Total communication | Total model training time |
|---|---|---|---|
| FL | $2|W|$ | $2K|W|$ | $T_{\text{fb}} + \frac{2|W|}{R} + T_{\text{fedavg}}$ |
| SL | $(\frac{2D}{K})v + 2r|W|$ | $2Dv + 2rK|W|$ | $T_{\text{fb}} + \frac{2Dv}{R} + \frac{2r|W|K}{R}$ |
| SFL-V1 | $(\frac{2D}{K})v + 2r|W|$ | $2Dv + 2rK|W|$ | $T_{\text{fb}} + \frac{2Dv}{RK} + \frac{2r|W|}{R} + T_{\text{fedavg}}$ |
| SFL-V2 | $(\frac{2D}{K})v + 2r|W|$ | $2Dv + 2rK|W|$ | $T_{\text{fb}} + \frac{2Dv}{RK} + \frac{2r|W|}{R} + \frac{T_{\text{fedavg}}}{2}$ |

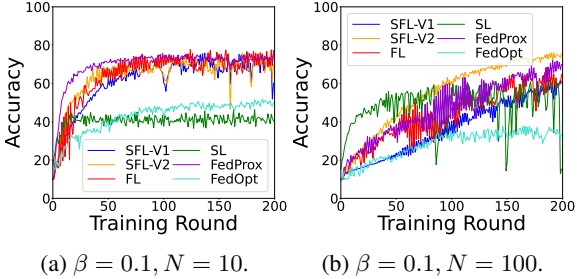

(a) $\beta = 0.1, N = 10.$     (b) $\beta = 0.1, N = 100.$

Figure 6: Performance comparison on CIFAR-10.

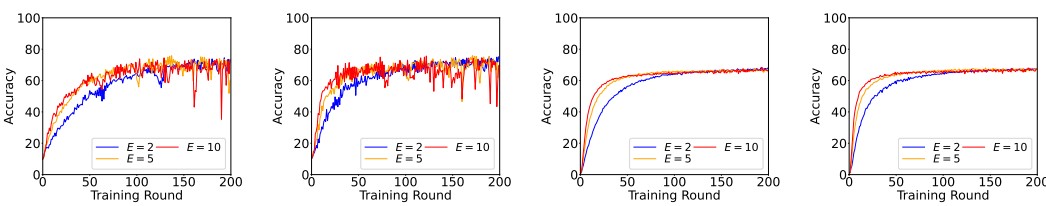

(a) SFL-V1 on CIFAR-10. (b) SFL-V2 on CIFAR-10. (c) SFL-V1 on CIFAR-100. (d) SFL-V2 on CIFAR-100.

Figure 7: Impact of local iteration on SFL performance.

# I  Additional Experiments

## I.1  More comparing methods

We compare the SFL methods with the benchmarks, i.e., FedProx [15] and FedOpt [19]. We have used the same hyperparameters, and trained the models on CIFAR-10. The results are provided in Figure 6. From the figures, we observe that SFL-V2 continues to be the best-performing algorithm. This is consistent with our observations in the main paper.

## I.2  Impact of local iteration

We further study the impact of local epoch number $E$ on the SFL performance. The results are reported in Fig. 7. We observe that SFL generally converges faster with a larger $\tau$, demonstrating the benefit of SFL in practical distributed systems.

## I.3  Results using loss metric

We further report the results using the loss metric. More specifically:

- *Impact of cut layer*: The results are reported in Figs. 8 and 9.
- *Impact of data heterogeneity*: The results are reported in Fig. 10.
- *Impact of partial participation*: The results are reported in Fig. 11.

In general, we see similar (but opposite) trends with the observations in the main paper. That is, a higher accuracy is associated with a smaller loss. These results are again consistent with our theories.

## I.4  Results on the impact of the position of cut layer.

We can observe from the results that the performance of SFL-V1 and SFL-V2 increases in $L_c$. We look at the impact of the position of cut layer from the gradient perspective. We plot the gradient divergence in Fig. 12:

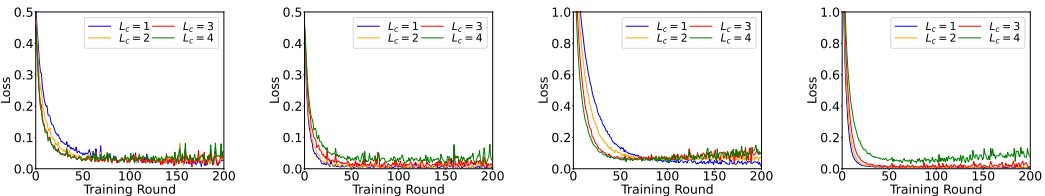

(a) SFL-V1 on CIFAR-10.  (b) SFL-V2 on CIFAR-10.  (c) SFL-V1 on CIFAR-100. (d) SFL-V2 on CIFAR-100.

Figure 8: Impact of the choice of cut layer on SFL training loss.

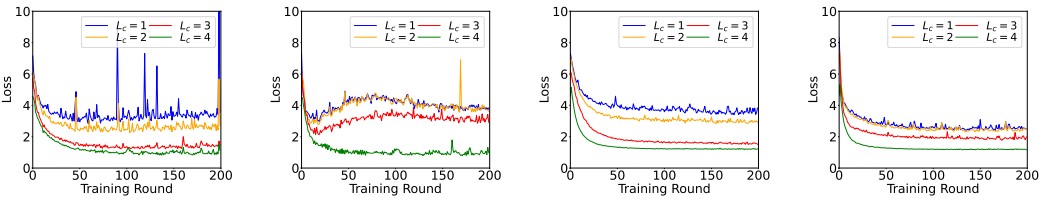

(a) SFL-V1 on CIFAR-10.  (b) SFL-V2 on CIFAR-10.  (c) SFL-V1 on CIFAR-100. (d) SFL-V2 on CIFAR-100.

Figure 9: Impact of the choice of cut layer on SFL test loss.

We see for SFL-V1 and SFL-V2, the gradient divergence decreases as we choose a latter cut layer (a larger $L_c$). This means that the client drift issue is less severe and hence the performance increases. In addition, from a theoretical perspective, if we write $\epsilon^2$ (i.e., upper bound of gradient divergence defined in Assumption 3.3) as a function of $L_c$, we can see that the upper bounds of performance loss decrease in $L_c$. This provides a theoretical angle that the performance of SFL-V1 and SFL-V2 increases in $L_c$.

## I.5   Results on FEMNIST dataset

We have conducted more simulations on a larger dataset FEMNIST. In particular, we consider $N = 100$ and train FL, SFL-V1, SFL-V2. Note that the data come from different sources and are heterogeneous across clients. The results are reported in Fig. 13.

We note that our key observation continues to hold. That is, SFL-V2 outperforms FL and SFL-V1 under-performs FL under heterogeneous data and a large number of clients.

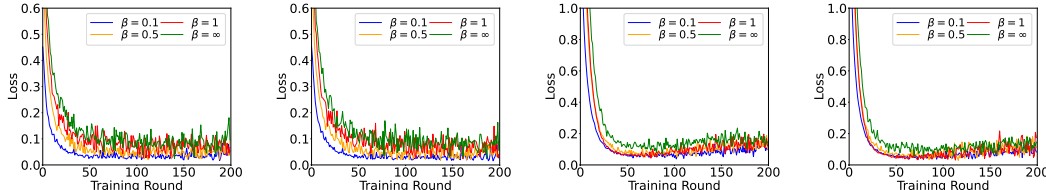

(a) SFL-V1 on CIFAR-10.  (b) SFL-V2 on CIFAR-10.  (c) SFL-V1 on CIFAR-100.  (d) SFL-V2 on CIFAR-100.

Figure 10: Impact of data heterogeneity on SFL training loss.

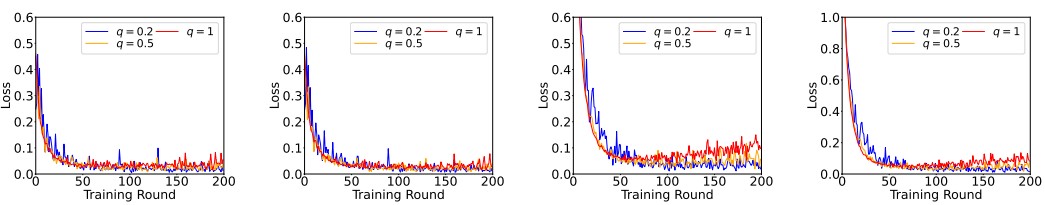

(a) SFL-V1 on CIFAR-10.  (b) SFL-V2 on CIFAR-10.  (c) SFL-V1 on CIFAR-100.  (d) SFL-V2 on CIFAR-100.

Figure 11: Impact of client participation on SFL training loss.

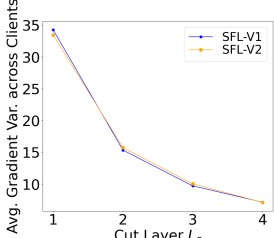

Figure 12: Results on the impact of the position of cut layer.

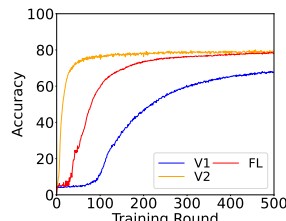

Figure 13: Results on FEMNIST.

