# OpenReview forum: "Convergence Analysis of Split Federated Learning on Heterogeneous Data"
_NeurIPS.cc/2024/Conference — NeurIPS 2024 poster_

### Official Review · Reviewer_8Y1a · 2024-06-23

**Soundness:** 3
**Presentation:** 2
**Contribution:** 2
**Rating:** 5
**Confidence:** 4

**Summary:**

Split federated learning combines the advantage of split learning and federated learning. However, there is
a lack of theoretical analysis. This paper provides the first comprehensive analysis on split federated learning. The settings include strongly convex, general convex and non-convex. This paper discusses both the full participation and partial participation case. Both two versions SFL-v1 and SFL-v2 are analyzed. Moreover, some experiments are conducted to compare split federated learning with federated learning and split learning.

**Strengths:**

This paper provides the first theoretical analysis of split federated learning. I think that this is an important problem.

The analysis is comprehensive. It includes strongly convex, general convex and non-convex case. It also discusses full client participation and partial client participation. Two variants of SFL are discussed, SFL v1 and SFL v2.

This paper has a long appendix, indicating that authors indeed have worked hard on this work.

**Weaknesses:**

I am impressed by the long appendix, which shows that the authors indeed work very hard on this problem. Nevertheless, I think that this paper is not ready for publication currently, due to many potential concerns.

1. In paragraph 2 of introduction, this paper states that split learning leads to high latency, but this paper does not provide theoretical/experimental evidences.

2. Inconsistency between theory and experiments. The theory only gives the convergence rate. It fails to explain the experiment phenomenon shown in this paper. In particular, the theoretical analysis does not reveal why SFL outperforms FL/SL under strong heterogeneity.

3. Line 128: statements need to be improved. It is weird to say "Based on x_s^{t,i}, based on which ... "

4. The assumptions need to be compared with existing works on federated learning.

For example, in Li et al. On the convergence of Fedavg on non-iid data, ICLR 2020, Assumption 3.3 is not required. Personally I also feel that the bound of heterogeneity by epsilon^2 may not be necessary. Hope that authors can explain more about why this assumption is necessary and how is the convergence rate affected if we remove this assumption.

5. In eq.(10), gamma=8S/\mu-1 need to be put later, after line 190. gamma is only used for the strongly convex case.

6. In Theorem 3.5, I do not see lower bound on the learning rate. To ensure that (11) holds, I think that a lower bound of \eta_t is needed. Merely requiring \eta_t\leq 1/(2Stau_max) may not be enough. Note that \eta_t=0 also satisfies \eta_t\leq 1/(2S\tau_{\max}), then E[f(x_T)]-f^* will not converge to zero.

7. I do not see any term including epsilon in strongly convex and general convex case. It is only included in the nonconvex case. Why the convergence is affected under nonconvex case but not convex case? Some intuitions are needed here.

8. In Figure 4 and Figure 5, From the curves, the accuracy oscillates violently. Therefore I suggest to reduce the learning rate.

9. List of inequalities after line 486. A^t is not defined. From the relationship above, one can only get
A^{t,i}\leq CA^{t,i-1}+B\sum_{j=0}^{i-1} C^j.

10. In line 520. First equality does not use (a+b)^2=a^2 +2ab+b^2. It just expands x_s^{t+1}. Perhaps the authors mean the second equality.

11. Table 1 needs to be put in the main paper. The comparison is important.

12. Line 828: Notation abuse of beta. In experiment section, it refers to the degree of heterogeneity.
Here it refers to the fraction of full model accessible to clients.

**Questions:**

1.In eq.(17): It seems that partial participation significantly increases the error. In particular, $G^2/q_n$ is significantly larger than $G^2$. It is not the case for federated learning. In the ICLR paper, compare eq.(6) with eq.(4), the only difference is that B is changed to B+C. Therefore, the results seem to suggest that the Splitfed has significantly worse performance for partial client participation. Please provide some intuitive explanations.

2. a_n is much smaller than 1, thus eq.(20) is significantly larger than eq.(17), indicating that SFL-V2 performs worse. This seems to be inconsistent with the experiment (which shows that SFL-V2 is better). My guess is that the current bound is not tight. Please explain.

3. Are there any supports from theoretical analysis, proving that SFL indeed performs better than FL/SL under high heterogeneity?

**Limitations:**

I think that the authors do not summarize the limitations well. The authors have mentioned how the choice of the cut layer affects the SFL performance. However, I think that the major limitation is that the good performance under strong heterogeneity is not well supported by theoretical analysis.

---

> ### Author Rebuttal · Authors · 2024-08-07
>
> Thank you for your valuable comments! Our response to your comments is as follows. We use "W*" to tag responses to comments in the Weakness and "Q*" for responses to comments in the Questions.
>
> **1. Latency of SL and SFL (for W1)**
>
> Split learning has higher latency than centralized learning due to the additional transmission of intermediate features and feature gradients between the two agents. In SFL, the communication between clients and the main server results in higher latency per training round compared to FL.
>
> We compare the communication overhead and total model training time for FL, SL, and SFL in Table 3 of the appendix.
> The total communication time of SFL is primarily determined by the size of model, the proportion of client-side model over the whole model, and the size of smashed layer. It is observed that SFL may induce higher latency when the data size and smashed layer size are large.
> We will revise our paper to enhance its precision.
>
> **2. Theoretical results for SFL and FL comparison (for W2, Q3)**
>
> We first explain the reason why SFL outperforms FL/SL under high data heterogeneity. Under high data heterogeneity, FL suffers from the notorious client drift issue, while SL suffers from the catastrophic forgetting problem. SFL (in particular SFL-V2) combines the training logic of FL (at clients) and SL (at server), which can help mitigate the issues of client drift and forgetting. Hence, SFL can lead to a better performance than both FL and SL. On the other hand, the theoretical comparison between SFL and FL/SL will serve as our future work. Through further characterizing the impact of cut layer on client drift and the degree of catastrophic forgetting in the theoretical analysis, we would be able to validate the aforementioned discussions and the theoretical improvement of SFL compared with FL/SL.
>
> Then, since our main focus is analyzing the convergence rate of SFL, we would like to summarize the observations that holds in both theoretical and empirical results. First, the performance of SFL degrades in data heterogeneity. In theoretical analysis, as $\sigma_n^2$ or $\epsilon_n^2$ increases, the convergence error increases. This is consistent with the empirical results that as $\beta$ decreases, the accuracy of the trained model decreases. Second, from both the theoretical and empirical results, as the number of clients participating in each round (i.e., $q$) increases, the convergence error decreases. Third, from the empirical results, the accuracy of the trained model increases with the cut layer. This can be indirectly supported by our theoretical analysis. In particular, we empirically show that as the cut layer $L_c$ increases from 1 to 4, the maximum bound of the gradients across clients (i.e., $G^2$ defined in Eq. (7)) reduces (in particular, its values are 21.53, 19.86, 15.48, 12.81 for Lc = 1, 2, 3, 4, respectively). Based on the theoretical analysis, such a reduction of $G^2$ indicates the reduction of the convergence error and hence the improvement of the model accuracy.
>
> **3. Assumptions in different cases (for W4, W7)**
>
> The Assumptions used under different cases for SFL-V1 and SFL-V2 are as follows:
>
> - $\mu$-strongly convex: Assumptions 3.1 (smoothness) and 3.2 (Unbiased and bounded stochastic gradients with bounded variance);
>
> - General convex: Assumptions 3.1 (smoothness) and 3.2 (Unbiased and bounded stochastic gradients with bounded variance);
>
> - Non-convex:  Assumptions 3.1 (smoothness), 3.2 (6) and (7) (Unbiased and bounded stochastic gradients), and 3.3 (heterogeneity).
>
> The above assumptions are widely applied in the literature. For instance, in the strongly convex case, the assumptions used in our paper are the same as those in Li et al. (2020). The heterogeneity assumption, captured by $\epsilon^2$, is used only for the non-convex case, as in Jhunjhunwala et al. (2023) and Wang et al. (2023).
>
> The difference in assumptions for strongly/general convex and non-convex cases is due to the form of convergence error we obtain. Specifically, we need to compute the model update in each local training with multiple iterations, denoted by $\tilde{\tau}$ and $\tau$.
> For strongly and general convex loss functions, we propose Lemma C.5, which directly provides the upper bound of the model parameter change over $\tau$ iterations. However, for non-convex loss functions, we compute the relationships between the model updates and the global gradient of the loss functions, as described in Lemma C.6.
> The upper bound of the squared norm of stochastic gradients, given by equation (7) in Assumption 3.2, and the upper bound of the divergence between local and global gradients, given by equation (8) in Assumption 3.3, are used to prove Lemmas C.5 and C.6, respectively.
>
> [Jhunjhunwala et al. 2023] D. Jhunjhunwala, S. Wang, and G. Joshi, “Fedexp: Speeding up federated averaging via extrapolation,”
> ICLR, 2023.
>
> [Wang et al. 2023] S. Wang, J. Perazzone, M. Ji, and K. S. Chan, “Federated learning with flexible control,” IEEE Conference on Computer Communications, 2023.
>
> **4. Theoretical results on partial participation (for Q1)**
>
> Our paper considers a different partial participation mechanism compared to the ICLR paper, resulting in different convergence error bounds. Specifically, the ICLR paper assumes a constant number of participating clients in each training round, with clients randomly selected with equal probability. In contrast, our paper considers independent participation probabilities for each client, allowing for arbitrary and heterogeneous participation probabilities.
> To prove the convergence error bound with clients' partial participation, we compute the difference between full and partial participation. In the future, we plan to improve these convergence bounds to make them tighter.
>
> Due to space limitations, we will add additional responses to your comments in Official Comments.

---

> ### Author Response · Authors · 2024-08-07
> **Additional response to comments**
>
> Due to space limitations, we add additional responses to your comments as follows.
>
> **5. Presentation improvement (for W3, W5, W9, W10, W11, W12)**
>
> We have carefully refined the presentation of the entire paper mainly including the following:
>
> - For line 128, it is revised as "SFL-V2: the main server computes the loss $F_n([\boldsymbol{x}_{c,n}^{t,i},\boldsymbol{x}_s^{t,i}])$ , based on which it then
>  updates the server-side model $\boldsymbol{x}_s^{t,i}$."
>
> - We will revise the presentation of $\gamma$ in equation (10) to include it only in the strongly convex case.
>
> - For line 487, we define $A^{t}:=A^{t,0}=\mathbb{E}\left[\left\Vert \boldsymbol{x}^{t,0}-\boldsymbol{x}^{t}\right\Vert^2\right]=0$. This is achieved by the fact that $\boldsymbol{x}^{t}=\boldsymbol{x}^{t,0}$ at the beginning of each training round $t$.
>
> - For line 520, the second equality is due to $\left(a+b\right)^2=a^2+2ab+b^2$.
>
> - We will move Table 1 to the main paper and explain the comparison between Mini-batch SGD, FL, SL, and SFL.
>
> - For line 828, we will redefine the notation to avoid confusion. Let $K$ represent the total number of clients involved and $D$ denotes the aggregate size of the data. Let $v$ indicate the size of the smashed layer.  The rate of communication is given by $R$.  $T_{\rm fb}$  signifies the duration required for a complete forward and backward propagation cycle on the entire model for a dataset of size $D$. The time needed to aggregate the full model is expressed as $T_{\text{fedavg}}$. The full model's size is denoted by $|W|$.  $r$ reflects the proportion of the full model’s size that is accessible to a client in SFL. The updated Table 3 is shown as follows.
>
> |Method|Communication per client|Total communication|Total model training time|
> |:-:|:-:|:-:|:-:|
> |FL|$2\|W\|$ | $2K\|W\|$|$T_{\rm fb}+\frac{2\|W\|}{R}+T_{\rm {fedavg}}$|
> |SL|$(\frac{2D}{K})v+2r\|W\|$ | $2Dv+2rK\|W\|$|$T_{\rm fb}+\frac{2Dv}{R} + \frac{2r\|W\|K}{R}$ |
> |SFL-V1|$(\frac{2D}{K})v+2r\|W\|$|$2Dv+2r K\|W\|$|$T_{\rm fb}+\frac{2Dv}{RK}+\frac{2r\|W\|}{R}+T_{\text{fedavg}}$ |
> |SFL-V2|$(\frac{2D}{K})v+2r\|W\|$|$2Dv+2r K\|W\|$|$T_{\rm fb}+\frac{2Dv}{RK}+\frac{2r\|W\|}{R}+\frac{T_{\text{fedavg}}}{2}$ |
>
> **6. Learning rate of SFL (for W6)**
>
> SFL and other deep learning algorithms, such as FL, require a positive learning rate to ensure the model is updated. Therefore, the lower bound of the learning rate is $\eta^t > 0$. We will update this condition in our main paper.
>
> **7. Explanation of experimental results (for W8)**
>
> The fluctuations in model accuracy observed in Figures 4 and 5 are due to clients' partial participation in SFL. In particular, clients are randomly sampled for training in each training round. Due to the data heterogeneity of the clients, the datasets of the sampled clients in each round may not represent those of the entire client population. Since the model is trained based on the local datasets of the sampled clients in each training round, the varying set of sampled clients makes the model accuracy fluctuate across rounds. This phenomenon has been discussed and verified in [Kairouz et al. 2021, Section 7.2] and [Tang et al. 2023]. Moreover, this explanation on the accuracy fluctuation can be supported by  Figure 4, where the fluctuations across training rounds become more pronounced at lower participation probabilities, $q$. On the other hand, regarding the learning rate,  we maintain the same learning rate across different participation probabilities to ensure a fair comparison. The learning rate was fine-tuned to achieve the best converged model accuracy under full participation.
>
> [Kairouz et al. 2021]  P. Kairouz and et al., ``Advances and open problems in federated learning", arXiv, Mar. 2021.
>
> [Tang et al. 2023] M. Tang and V. W.S. Wong, “Tackling system induced bias in federated learning: Stratification and convergence analysis,” IEEE International Conference on Computer Communications (INFOCOM), May 2023.
>
> **8. Theoretical results comparison between SFL-V1 and SFL-V2 (for Q2)**
>
> Our convergence bounds demonstrate the same convergence rate with respect to the training round $T$ for both SFL-V1 and SFL-V2. Despite $\alpha_n$, SFL-V1 and SFL-V2 have different bounds on the stochastic gradient variance
> $\sigma_n^2$. Specifically, SFL-V2 is more centralized than SFL-V1, resulting in a lower $\sigma_n^2$. In the future, we will improve the convergence error rate to make it tighter.
>
> **9. Limitation of our paper (for Limitation)**
>
> One limitation is that our bounds for SFL achieve the same order (in terms of training rounds) as in FL, yet the experiments showed that SFL outperforms FL under high heterogeneity. This is possibly due to that tighter bounds for SFL are to be derived, which is an important future work. Another limitation is that we did not include the theoretical analysis on how the choice of cut layer affects the convergence. We will add the more detailed discussions of limitations to the main paper.

---

> > ### Author Response · Authors · 2024-08-12
> >
> > Dear Reviewer 8Y1a,
> >
> > We hope that our rebuttal above has addressed your concerns. As the author-reviewer discussion period is
> > ending very soon, please kindly let us know if you have further questions or comments. We will add all the new results and discussions to the final paper. Thank you very much!
> >
> > Best regards,
> > Authors

---

> > > ### Comment · Reviewer_8Y1a · 2024-08-12
> > >
> > > Thanks for your detailed response. My main concern is the inconsistency between theory and experiments. I think that authors have provided a good qualitative discussion on the reason why SFL outperforms FL/SL under high data heterogeneity. However, for a solid theoretical work, I think that it is necessary to theoretically justify that SFL outperforms FL/SL. For example, it would be better if the bound contains $\beta$ in your revision, and the dependence of $\beta$ of SFL is better than FL/SL.
> > >
> > > I will not strongly oppose the acceptance if it is the consensus.

---

> > > > ### Author Response · Authors · 2024-08-13
> > > >
> > > > Thank you for your feedback! We plan to enhance our convergence results to enable a theoretical comparison with FL/SL in the future. One potential approach is to improve results for the client-side and server-side models respectively. Moreover, to our best knowledge, including the data heterogeneity parameter $\beta$ in the convergence analysis is not a common practice in the literature. Our analysis follows typical assumptions in the FL community and uses the upper bound of gradient divergence to characterize data heterogeneity.
> > > >
> > > > Again, we want to emphasize our primary contribution in this paper: presenting the first convergence analysis for SFL under different objectives and partial participation. The dual-paced model update mechanism in SFL adds complexity to the convergence analysis, making it a challenging yet significant achievement.
> > > >
> > > > Thank you once again for your valuable comments.

---

> > > > > ### Comment · Reviewer_8Y1a · 2024-08-13
> > > > >
> > > > > Thanks for your feedback. Since it is the first analysis on SFL, I think that it is reasonable that the analysis is not very complete, although I think that it would be better to let the theoretical analysis match your claims from experiments. Now I increase the soundness score to 3 and the overall rating to 5.
> > > > >
> > > > > As you said, I suggest to further emphasize in the paper that this is the first convergence analysis of SFL. Although this does not fully explain the advantage of SFL, the first analysis is still important. Another suggestion is to try your best to simplify the proof to reduce the length of the paper.

---

> ### Author Response · Authors · 2024-08-13
>
> Thank you very much for your quick response!
>
> We will enhance our main paper by emphasizing the contributions and simplifying the proof if the paper gets accepted.
> We will also improve our convergence results to make them more robust, as you suggested, in the future. Thanks again for your time and positive feedback on our paper.

---

### Official Review · Reviewer_NwMP · 2024-07-09

**Soundness:** 4
**Presentation:** 4
**Contribution:** 4
**Rating:** 8
**Confidence:** 5

**Summary:**

Split Federated Learning (SFL) is a recent distributed approach for collaborative model training among multiple clients. In SFL, a global model is typically split into two parts, where clients train one part in a parallel federated manner, and a main server trains the other. Despite recent research on SFL algorithm development, the convergence analysis of SFL is missing in the literature, and this work aims to fill this gap. The analysis of SFL can be more challenging than that of Federated Learning (FL), due to the potential dual-paced updates at the clients and the main server. This work provides a convergence analysis of SFL for strongly convex and general convex objectives on heterogeneous data.

**Strengths:**

1. Split federated learning is an interesting field in FL, however, existing work fail to provide some theoretical analysis for SFL, e.g., convergence rate. And this is the first work provide convergence analysis of SFL for strongly convex, general convex, and non-convex objectives on heterogeneous data.

2. This work offers comprehensive discussion about the convergence rate of SFL under several settings and the theoretical contributions are excellent. Specifically, this work provides the convergence analysis for SFL with strongly-convex, convex, and non-convex objectives under the full participation and partial participation settings. Moreover, the assumptions in the theoretical analysis are mild and widely-used in the literature of FL.

3. The presentation and organization of this work are good.

4. Extensive experiments are conducted to evaluate the properties of SFL, e.g., the impact of cut layer, impact of data heterogeneity, and so on.

**Weaknesses:**

I believe this is a solid and interesting work, I also have some minor concerns as follows.

1.	The section on Related Work is too brief, more discussion about the SFL needs to be included.

2.	I suggest that the author provide a table to summarize and list the convergence rate of SFL under different settings, such as convex, non-convex, full participation, and partial participation. In this way, the theoretical results can be clearer.

**Questions:**

Please refer to the Weaknesses.

---

> ### Author Rebuttal · Authors · 2024-08-07
>
> Thank you for your valuable comments! Our response to your comments is as follows.
>
> **1. Related work**
>
> We enriched the related work as follows. In particular, we have added more discussions on the references.
>
>  There are many convergence results on FL.  Most studies focus on data heterogeneity. For example, [Li et al. 2020] proved the convergence of FedAvg on heterogeneous data. [Wang et al. 2019] gave the convergence of FL, which is used to design a controlling algorithm to minimize loss under budget constraint. [koloskova et al. 2020] provided a general framework of convergence analysis for decentralized learning.  Some studies look at partial participation. For example, [Yang et al. 2021] speeds up FL convergence in the presence of partial participation and data heterogeneity. [Wang and Ji 2022] gave a unified analysis for partial participation using probabilistic models. [Tang and Wong 2023] analyzed how to tackle system induced bias due to partial participation.  There are also convergence results on Mini-Batch SGD, where [Woodworth et al. 2020] argued that the key difference between FL and Mini-Batch SGD is the communication frequency.
>
>
> To our best knowledge, there is only one recent study [Li and Lyu 2024] discussing the convergence of SL. The major difference to SL analysis lies in the sequential training manner across clients, while SFL clients perform parallel training.
>
>
> [li et al. 2020] Li X, Huang K, Yang W, et al. On the convergence of fedavg on non-iid data. ICLR, 2020.
>
> [Wang et al. 2019] Wang S, Tuor T, Salonidis T, et al. Adaptive federated learning in resource constrained edge computing systems[J]. IEEE journal on selected areas in communications, 2019, 37(6): 1205-1221.
>
> [Koloskova et al. 2020] Koloskova A, Loizou N, Boreiri S, et al. A unified theory of decentralized sgd with changing topology and local updates, ICML, PMLR, 2020: 5381-5393.
>
> [Yang et al. 2021]Yang H, Fang M, Liu J. Achieving linear speedup with partial worker participation in non-iid federated learning. ICLR 2021.
>
> [Wang and Ji 2022] Wang S, Ji M. A unified analysis of federated learning with arbitrary client participation. NeurIPs, 2022, 35: 19124-19137.
>
> [Tang and Wong 2023] Tang M, Wong V W S. Tackling system induced bias in federated learning: Stratification and convergence analysis. INFOCOM, 2023: 1-10.
>
> [Woodworth et al. 2020] Woodworth B E, Patel K K, Srebro N. Minibatch vs local sgd for heterogeneous distributed learning.  NeurIPs, 2020, 33: 6281-6292.
>
> [Li and Lyu 2024] Li Y, Lyu X. Convergence analysis of sequential federated learning on heterogeneous data. NeurIPs, 2024, 36.
>
>
> **2. Table of theoretical results**
>
> We conclude the convergence results in our paper as follows ($Q:=\sum_{n=1}^N\frac{1}{q_n}$):
>
> | Scenario | Case |    Method    |  Convergence result  |
> |---|---|---|---|
> |Full participation| Strongly convex | SFL-V1| $\frac{S}{\mu(S+\mu T)}( N(\sigma^2 + G^2)+ \mu S I^{\rm err})$|
> |Full participation| Strongly convex | SFL-V2|$\frac{S}{\mu(S+\mu T)}(N^2 (\sigma^2+G^2)+ \mu S I^{\rm err})$|
> |Full participation| General  convex| SFL-V1|$(\frac{N(\sigma^2 + G^2)}{T})^{\frac{1}{2}}+(\frac{N(\sigma^2 + G^2)}{T})^{\frac{1}{3}}+\frac{S}{T} I^{\rm err} $|
> |Full participation| General  convex| SFL-V2 | $(\frac{N^2(\sigma^2 + G^2)}{T})^{\frac{1}{2}}+(\frac{N^2(\sigma^2 + G^2)}{T})^{\frac{1}{3}}+\frac{S}{T} I^{\rm err} $|
> |Full participation| Non-convex | SFL-V1 | $\frac{NS(\sigma^2 + \epsilon^2)}{T}+\frac{\boldsymbol{F}^{err}}{T}$|
> |Full participation| Non-convex |  SFL-V2 | $\frac{N^2S(\sigma^2 + \epsilon^2)}{T}+\frac{\boldsymbol{F}^{err}}{T}$|
> |Partial participation | Strongly convex | SFL-V1 |$\frac{S}{\mu(S+\mu T)}( N(\sigma^2 + G^2(1+Q))+ \mu S I^{\rm err})$|
> |Partial participation | Strongly convex | SFL-V2 | $\frac{S}{\mu(S+\mu T)}(N^2 (\sigma^2+G^2(1+Q))+ \mu S I^{\rm err})$|
> |Partial participation | General  convex | FL-V1 | $(\frac{N(\sigma^2 + G^2(1+Q))}{T})^{\frac{1}{2}}+(\frac{N(\sigma^2 + G^2)}{T})^{\frac{1}{3}}+\frac{S}{T} I^{\rm err} $|
> |Partial participation | General  convex | SFL-V2 | $(\frac{N^2(\sigma^2 + G^2(1+Q))}{T})^{\frac{1}{2}}+(\frac{N^2(\sigma^2 + G^2)}{T})^{\frac{1}{3}}+\frac{S}{T} I^{\rm err} $|
> |Partial participation | Non-convex |SFL-V1 | $\frac{NS(\sigma^2 + \epsilon^2)Q}{T}+\frac{\boldsymbol{F}^{err}}{T}$|
> |Partial participation | Non-convex | SFL-V2 | $\frac{N^2S(\sigma^2 + \epsilon^2)Q}{T}+\frac{\boldsymbol{F}^{err}}{T}$|

---

> > ### Comment · Reviewer_NwMP · 2024-08-09
> >
> > Thanks for your reply. My questions have been well addressed with enriched related work and a conclusion of convergence results. I will keep the initial score.

---

> > > ### Author Response · Authors · 2024-08-13
> > >
> > > Thank you very much for the response! Thanks again for your work and positive feedback on our paper.

---

### Official Review · Reviewer_j2yY · 2024-07-12

**Soundness:** 2
**Presentation:** 3
**Contribution:** 2
**Rating:** 5
**Confidence:** 4

**Summary:**

This paper analyzes the convergence of split federated learning (SFL) on heterogeneous data and gives theoretical convergence rates under different assumptions on the convexity of the loss function and the participation schemes of the clients.

**Strengths:**

The paper considers the problem comprehensively, including the assumptions on the convexity of the loss function (strongly convex, convex, and non-convex), the participation schemes of the clients (full, partial), and split schemes (soft-sharing: SFL-V1, hard-sharing: SFL-V2), and provides a very detailed theoretical analysis of the convergence rates under these different settings.

**Weaknesses:**

Numerical experiments are comparably weak since the baselines for comparison are FedAvg and SL. The latter is pure local training under the perspective of federated learning. The former is a very basic federated learning algorithm. Perhaps the authors could compare with more advanced federated learning algorithms, such as FedProx, FedOpt, etc.

**Questions:**

1. In Figure 4, it can be observed that the accuracies of the algorithms on CIFAR-10 and CIFAR-100 are very close (the latter only slightly lower). However, the accuracies on CIFAR-100 generally have an observable gap compared to CIFAR-10. Is this due to the small number of clients (only 10) in the experiments? If so, how would the results change if the number of clients is increased?

2. In Figure 7, the losses of SFL-V2 (the hard-sharing scheme) almost tend to zero for the cutting layer = 1 or 2. I wonder if these curves are losses on the training set or the test set. If the losses are on the test set, the results are surprising, since even centralized training cannot achieve zero loss on the test set. Could the authors provide more insights into this phenomenon?

**Limitations:**

The last section discusses the limitations and provides some future research directions.

---

> ### Author Rebuttal · Authors · 2024-08-07
>
> Thank you for your valuable comments! Our response to your comments is as follows.
>
> **1. More comparing methods**
>
> We have included the benchmarks, i.e., FedProx and FedAdam, as suggested. We have used the same hyper-parameters, and trained the models on both CIFAR-10 and CIFAR-100. The results are provided in Figure 1 in the [global rebuttal](https://openreview.net/forum?id=ud0RBkdBfE&noteId=kM9CF1FAdt). From the figures, we observe that SFL-V2 continues to be the best performing algorithm. This is consistent with our observations in the main paper.
>
> **2. Explanation on the experimental results of CIFAR-10 and CIFAR-100**
>
> You are right. On CIFAR-100 we used 10 clients. Hence, after data sampling (with a Dirichlet parameter $\beta=0.1$), each client still has relatively many classes (out of 100), leading to similar performance as in CIFAR-10. We have conducted more experiments with 100 clients on CIFAR-100 using $\beta=0.1$, and the results are given in Figure 2 in the [global rebuttal](https://openreview.net/forum?id=ud0RBkdBfE&noteId=kM9CF1FAdt). The key observation is that the accuracy with 100 clients is generally lower than that with 10 clients, as expected. The reason is that each client has fewer types of classes, which can escalate the client drift issue during training.
>
> **3. Explanation on the experimental results of loss**
>
> The loss plot is based on the training set rather than the test set, which is why, in certain cases, the losses approach zero.  We have also plotted the impact of the choice of cut layer on the SFL test loss, as shown in  Figure 3 in the [global rebuttal](https://openreview.net/forum?id=ud0RBkdBfE&noteId=kM9CF1FAdt). In this figure, we observe that the test loss is higher than the training loss, highlighting the generalization gap and the effect of cut layer selection on model performance.
>
>
>
> **4. Limitation and future work of our paper**
>
> One limitation is that our bounds for SFL achieve the same order (in terms of training rounds) as in FL, yet the experiments showed that SFL outperforms FL under high heterogeneity. This is possibly due to that tighter bounds for SFL are to be derived, which is an important future work. Another limitation is that we did not include the theoretical analysis on how the choice of cut layer affects the convergence.
> For future work, we can apply our derived bounds to optimize SFL system performance, considering model accuracy,  communication overhead, and computational workload of clients.
> We will add more detailed discussions of limitations and future work to the main paper.

---

> > ### Comment · Reviewer_j2yY · 2024-08-13
> >
> > See the reply to the comments of the AC.

---

> > > ### Author Response · Authors · 2024-08-14
> > >
> > > Dear Reviewer j2yY,
> > >
> > > Thanks for your reply. However, we did not see any comments of the AC and related replies. Maybe it is due to the visibility setting. Can you repeat your comments again? Thanks a lot!
> > >
> > > Best regards,
> > >
> > > Authors

---

> > > > ### Comment · Reviewer_j2yY · 2024-08-14
> > > >
> > > > Two points on the numerical experiments:
> > > >
> > > > 1. I read the attached 1 page PDF file but surprisedly found that the FedOpt (FedAdam) performed almost the worst. This contradicts my previous research experience. You can check the FedOpt paper (https://arxiv.org/pdf/2003.00295). I re-implemented 10+ federated learning algorithms and conducted a comprehensive series of numerical experiments using computer vision datasets. The performance of these experiments (relative, not absolute) highly agrees with the FedOpt paper figures.
> > > >
> > > > 2. It's quite weird to draw performance curves only of the training set instead of the test set. It's more common to draw the test set or both.

---

> > > > > ### Author Response · Authors · 2024-08-14
> > > > >
> > > > > Thank you for your reply. Our response is as follows:
> > > > >
> > > > > Regarding the performance of FedOpt, due to time constraints, we only experimented with FedAdam (one of the algorithms in the FedOpt paper) using the same parameter settings as our original setup—specifically, learning rates of 1.0 for the server and 0.01 for the clients. Based on our experimental results, FedAvg outperforms FedOpt at these learning rates. This is consistent with the results in the FedOpt paper (see Figure 5 on Page 31 in the FedOpt paper). we will conduct hyper-parameter tuning for FedOpt in the final version of our paper.
> > > > >
> > > > > Additionally, we will include the results of the test loss, such as those shown in Figure 3 of the global response PDF, in our paper. Thank you again for your valuable comments.

---

### Official Review · Reviewer_b5tv · 2024-07-12

**Soundness:** 2
**Presentation:** 3
**Contribution:** 2
**Rating:** 5
**Confidence:** 3

**Summary:**

The paper proposes the first convergence analysis of the two variants of split federated learning under different set of hypotheses.

**Strengths:**

To the best of my knowledge, as also stated by the authors, there is no theoretical analysis of split federated learning. This is a serious knowledge gap, given the potential practical interest of split FL. The paper proposes the first of such analysis under the classic set of hypotheses: L-smoothness and strong convexity, L-smoothness and convexity, non-convexity.

**Weaknesses:**

I was not able to check the proofs in their completeness, but there are some signs suggesting that the analysis may not have been carried out as carefully as required. First, it seems some hypotheses are missing in the statement of the main theoretical results. For example, convergence to the minimizer in presence of stochastic gradients is not possible unless the learning rate opportunely decreases over time, but this is not mentioned. Second, I have some concerns because 1) the verbal description of the studied algorithms is not very clear, and, more importantly, 2) even their mathematical descriptions are not consistent across the paper. Before moving to the specific comments, I want to stress that a certain vagueness in the actual operation of the algorithm is common to the original papers on split learning and split federated learning, but it is a less serious concern in their case because they do not provide theoretical guarantees.
My main concerns are about how models are aggregated. Now, the two aggregations presented in the main text (equation (3)) and in the algorithm formal presentations in appendix B (line 22 in algorithm 1) do not match. The first one presents an unbiasing effect, the second one not. More importantly, neither one nor the other can lead to asymptotic convergence under partial participation, because there would be a persistent noise due to the variability in the set of participating clients. By a first look at the proof, it seems the proofs do not consider neither of these aggregation rules but rather an aggregation at the level of the gradients (see page 15), which may converge (but I did not check), but there are still some potential problems for SFL-v3 (line 453) because the a_n are ignored.

I have also some concerns about the comparison with FedAvg in the experimental section. My understanding is that the authors study a version of SplitFL where clients communicate after each batch. In this setting, SFL-v1 coincides with FedAvg if 1) \tau=\tilde{\tau} (both the server and the clients average the models at the same time) and 2) FedAvg is allowed to transmit after each gradient computation (as it seems required for a fair comparison). Now, the results presented show different performance for FedAvg and SFL-v1. Unfortunately, there is no information reported about the specific setting (i.e., what are \tau, \tilde{\tau} and how many local gradient steps FedAvg performs before communicating).

The authors have addressed my criticisms above and I have upgraded my score. It is required a significant review of the main text.

**Questions:**

See weaknesses

**Limitations:**

A discussion of the limitations (e.g. the fact that the convergence results hold only when clients are allowed to communicate after each batch) is missing.

---

> ### Author Rebuttal · Authors · 2024-08-07
>
> Thank you for your valuable comments! Our response to your comments is as follows.
>
> **1. Diminishing stepsize for $\mu$-strongly convex loss functions**
>
> We use a diminishing stepsize of $\frac{4}{\mu\tilde{\tau}(\gamma+t)}$ and update the conditions specific to $\mu$-strongly convex loss functions. For these types of loss functions, Theorems 3.5 - 3.8 rely on the hypothesis that $\eta^t = \frac{4}{\mu\tilde{\tau}(\gamma+t)}$ for the client-side model, $\eta^t = \frac{4}{\mu\tau(\gamma+t)}$ for the server-side model, and $\eta^t \leq \frac{1}{2 S\tau_{max}}$. When $\gamma = \frac{8S}{\mu} - 1$, we have $\frac{4}{\mu\tilde{\tau}(\gamma+t)} \leq \frac{1}{2 S\tau_{max}}$ and $\frac{4}{\mu\tau(\gamma+t)} \leq \frac{1}{2 S\tau_{max}}$.
>
> Thus, the hypotheses for the learning rate in different cases are as follows:
>
> - $\mu$-Strongly Convex: $\eta^t = \frac{4}{\mu\tilde{\tau}(\gamma+t)}$ for the client-side model, $\eta^t = \frac{4}{\mu\tau(\gamma+t)}$ for the server-side model.
>
> - General Convex: $\eta^t \leq \frac{1}{2 S\tau_{max}}$.
>
> - Non-convex: $\eta^t \leq \frac{1}{2 S\tau_{max}}$.
>
> Our proofs in the appendix (lines 534, 543, 607, 616, 668, 687, 746, 764) support this analysis.
>
> It is important to note that these hypotheses apply to both SFL-V1 and SFL-V2, under full and partial participation of clients. Additionally, $\tilde{\tau} = \tau$ for SFL-V2. We will revise the presentation in our main paper accordingly.
>
>
> **2. Model aggregation**
>
>  In SFL algorithms, the clients' local model parameters are aggregated according to equations (2) and (3) for full and partial participation, respectively. This process ensures unbiased model updates. Detailed model updates for both the server and clients in SFL-V1 and SFL-V2 can be found in Appendix C.2. Algorithms 1 and 2 describe the process for full participation only. We plan to revise these algorithms to include partial participation.
>
> In our proof, we consider the features of model aggregation in different scenarios, as outlined in Appendix C.2 (page 15). Specifically, in line 453, which describes the model aggregation for SFL-V2 with partial participation on the server side, the parameter $a_n$ is not needed. This is because the main server updates the server-side model parameter sequentially with each client.
>
> **3. Performance comparison between FedAvg and SFL-V1**
>
> SFL characterizes dual-paced model aggregation. Specifically,
> in SFL-V1 and SFL-V2, clients communicate with the main server during every iteration (batch) for sequential model training. Additionally, clients communicate with the fed server every $\tilde{\tau} \geq 1$ iterations (batches) for client-side model aggregation. On the main server side, in SFL-V1, the main server aggregates the server-side models every $\tau \geq 1$ iterations (batches). In contrast, in SFL-V2, the main server does not need to aggregate the server-side model because there is only one server-side model.
>
> In our experiments, we set $\tau = \tilde{\tau}$ to ensure a fair comparison with vanilla FL, i.e., FedAvg. Theoretically, the performance of SFL-V1 should be the same as FedAvg. However, our experiments show that FedAvg outperforms SFL-V1, especially when the data is non-iid distributed across clients, as shown in Fig. 5 (c) and (d).
>
> The reason for this discrepancy lies in the optimizer used in our experiments. We added momentum to the SGD optimizer to stabilize and accelerate model training. In SFL-V1, there are two separate optimizers for the client-side and server-side models. In contrast, FedAvg uses only one optimizer, which provides better global momentum than SFL-V1. As a result, FedAvg performs slightly better than SFL-V1. This phenomenon is also noted in the original SFL paper [1].
>
> We will revise the description of the experimental settings and results in our paper.
>
> [1] Thapa, C., Arachchige, P. C. M., Camtepe, S., \& Sun, L. Splitfed: When federated learning meets split learning. In Proceedings of the AAAI Conference on Artificial Intelligence, 2022, Vol. 36, No. 8, pp. 8485-8493.
>
>
> **4. Limitation of our paper**
>
> One limitation is that our bounds for SFL achieve the same order (in terms of training rounds) as in FL, yet the experiments showed that SFL outperforms FL under high heterogeneity. This is possibly due to that tighter bounds for SFL are to be derived, which is an important future work. Another limitation is that we did not include the theoretical analysis on how the choice of cut layer affects the convergence. We will add the more detailed discussions of limitations to the main paper.

---

> > ### Author Response · Authors · 2024-08-12
> >
> > Dear Reviewer b5tv,
> >
> > We hope that our rebuttal above has addressed your concerns. As the author-reviewer discussion period is
> > ending very soon, please kindly let us know if you have further questions or comments. We will add all the new results and discussions to the final paper. Thank you very much!
> >
> > Best regards,
> > Authors

---

> > ### Comment · Reviewer_b5tv · 2024-08-12
> >
> > I thank the authors for their answers. They confirm that there is too much important information which is relegated to the supplementary material. I trust that, if the paper gets accepted, the authors will make a significant revision of the main text to include the missing information (learning rates, client participation conditions, comparison with FedAvg) and increase my score to 5.

---

> > > ### Author Response · Authors · 2024-08-13
> > >
> > > Thank you very much for the response!
> > >
> > > We will certainly revise our main paper according to the comments if it gets accepted.
> > > Thanks again for your work and positive feedback on our paper!

---

### Official Review · Reviewer_fmCP · 2024-08-01

**Soundness:** 3
**Presentation:** 4
**Contribution:** 3
**Rating:** 7
**Confidence:** 2

**Summary:**

This paper provides theoretical convergence analysis for split federated learning (SFL) methods (SFL-V1 and SFL-V2) under full and partial client participation for strongly convex, generally convex and non-convex settings. These bounds help build fundamental understanding of the settings in which SFL, federated learning (FL) and split learning (SL) are best suited and where SFL suffers (e.g., in terms of data heterogeneity, number of clients, participation rate). The authors provide empirical results that validate their theoretical claims and showcase the performance tradeoffs between SFL, FL and SL.

**Strengths:**

* Well-written and well-structured.
* Theoretical convergence analysis provided for both full and partial client participation across strongly convex, general convex and non-convex objectives.
* Empirical validation of theoretical insights with ablations across datasets on data heterogeneity, number of clients and client participation rate.
* Important and interesting findings as to the most applicable data settings for each algorithm.
* Includes comparison of SL, FL and SFL in terms of accuracy, convergence speed, communication and computation costs.

**Weaknesses:**

* Empirical choice of L_c is not properly swept. Given that the highest value of L_c considered is found to yield the best performance, L_c should be increased until worse performance is observed to evaluate SFL and SL at the optimal choice of cut layer and understand at what point diminishing returns are observed from increasing L_c. Is it always better just to use FL if computational resources allow? What are the tradeoffs?
* Lacking specifics of what FL and SL algorithms are used for comparison. Opportunity to consider other algorithms beyond the most basic implementations of FL and SL.
* Plots are hard to read.

**Questions:**

Where are there diminishing returns in increasing L_c? If a increasing L_c yields better performance, then why not just use FL? What tradeoffs should be considered when selecting L_c beyond just accuracy?

**Limitations:**

While there is no explicit limitations section in the paper, and such discussion could be expanded upon in the text, the authors do mention that the choice of how to set the cut layer (L_c) is unknown and left for future work.

---

> ### Author Rebuttal · Authors · 2024-08-07
>
> Thank you for your valuable comments. Our response to your comments is as follows.
>
> **1. Choice of $L_c$ and tradeoff**
>
> We use the ResNet-18 model in our experiments, which contains four blocks. This means the candidate for $L_c$ is in {$1,2,3,4$}. Thus, $L_c=4$ is already the maximum $L_c$ that can be chosen in SFL with ResNet-18 model. Our experimental results in Figure 2 show that increasing $L_c$ leads to higher model accuracy for SFL. However, the model accuracy of FL is similar to SFL-V1, and both are lower than SFL-V2 under high data heterogeneity. This suggests that even if computational resources allow, it is not always better to use FL than SFL. The main reason is as follows. Under high data heterogeneity, FL suffers from the notorious client drift issue, while SL suffers from the catastrophic forgetting problem. SFL (in particular SFL-V2) combines the training logic of FL (at clients) and SL (at server), which can help mitigate the issues of client drift and forgetting. Hence, SFL can lead to a better performance than both FL and SL.
>
> On the other hand, as suggested in Table 3 of the Appendix, SFL-V1 and SFL-V2 may have higher communication overheads than FL when the number of training data and smashed layer size are large. Therefore, there is a tradeoff between model accuracy and total training time in SFL.
>
> **2. Algorithm details**
>
> The FL algorithm we use is FedAvg, and the SL algorithm we use is given in [Poirot et al. 2019]. We have also included more recent FL algorithms, i.e., FedProx and FedOpt, as additional comparison. The results are given as follows, which show that SFL-V2 outperforms FL and SL under high data heterogeneity. This is consistent with our main conclusion in the submission.
>
> [Poirot et al. 2019] Poirot M G, Vepakomma P, Chang K, et al. Split learning for collaborative deep learning in healthcare. NeurIPs, 2019.
>
>
> **3. Plot presentation**
>
> We have adjusted the fonts and line sizes to make the plots more readable.
>
>
> **4. Limitation of our paper**
>
> One limitation is that our bounds for SFL achieve the same order (in terms of training rounds) as in FL, yet the experiments showed that SFL outperforms FL under high heterogeneity. This is possibly due to that tighter bounds for SFL are to be derived, which is an important future work. Another limitation is that we did not include the theoretical analysis on how the choice of cut layer affects the convergence. We will add the more detailed discussions of limitations to the main paper.

---

> > ### Comment · Reviewer_fmCP · 2024-08-13
> >
> > Thank you for your response with an explanation of the split layer, added clarifications on algorithm details, and statement of limitations. This addresses my concerns.

---

> > > ### Author Response · Authors · 2024-08-13
> > >
> > > Thank you very much for the response. Thanks again for your time and positive feedback on our paper!

---

### Author Rebuttal · Authors · 2024-08-07

We would like to thank all of you for your time and comments on our paper. We have carefully revised our paper and prepared the responses. The added figures are in the attached file.

---

### Author Response · Authors · 2024-08-13

Dear Area Chair and Senior Area Chair,

Thank you very much for handling our submission! We also appreciate the reviewers' efforts in reviewing this paper. As the author-reviewer discussion phase is ending very soon, we would like to bring up the following to your attention:


**1. The contributions of our paper**

We provide **the first comprehensive convergence analysis of SFL for different objective cases on heterogeneous data**, including strong convex, general convex, and non-convex objectives. We further consider the practical **partial participation of clients** for the convergence analysis. The experimental results reveal interesting insights on the impact of the cut layer on SFL and the superiority of SFL-V2 over SFL-V1 and FL. Overall, our paper provides a thorough understanding on the performance of SFL, which potentially guides the implementation and optimization of SFL (e.g., the choice between FL and SFL, the choice of hyper-parameters and cut layers).

**2. Our response to reviewers' comments**

The reviewers raised several concerns regarding the impact of the cut layer $L_c$ on performance, the assumptions and hypotheses in our proof, the details of the algorithms, the theoretical and experimental comparisons between Federated Learning (FL) and Split Federated Learning (SFL), and the limitations of our paper. We have made the following revisions in response to these concerns.

- From a theoretical perspective, we **clarify the assumptions and conditions for learning rates for different objectives**. We also give a qualitative discussion on why SFL-V2 outperforms SFL-V1, FL, and SL.

- Experimentally, we add **additional results using two benchmarks**, FedProx and FedAdam. We also include experiments on CIFAR-100 with more clients under varying data heterogeneity, showing that our key observations hold. We explain phenomena in our results, such as the loss fluctuation and the differences between FL and SFL-V1.

- Additionally, we explain the **choice of $Lc$ and tradeoffs in SFL** by considering the computation and communication overheads alongside model performance, detail our algorithms' model aggregation and partial participation methods, expand the related work section, and enhance clarity throughout the paper. Finally, we add more detailed discussions on **limitations and future work**.

- **Concerns about the inconsistency between theory and experiments**. The comment says that while SFL empirically outperforms FL/SL under high data heterogeneity (i.e., a small value of $\beta$), this improvement is not directly captured in the convergence results. To this end, one needs to improve the bounds for SFL and a potential approach is to improve the results for client-side and server-side models. This would require a different approach and effort, which is left to future work.

If this paper gets accepted, we will include all the new results and discussions in the final version. Thank you again!

Very best regards,

Authors

---

### Decision · Program_Chairs · 2024-09-25

**Decision:**

Accept (poster)

**Comment:**

This paper studies the convergence of distributed split learning.
A theoretical convergence analysis is presented and the performance of the method is empirically studied for various levels of heterogeneity.

The reviewers argued that the presentation of the results must be further improved for the final version:
- currently, too many details are delegated to the appendix. The main text should have a precise mathematical description of (at least one version of)  the algorithm.
- details on the numerical experiments are missing or not well explained. For example, the choice of the hyperparameters for the baseline methods (best practice would be to tune the parameters for each setting separately).

However, the reviewers concluded that the theoretical results presented in this paper are a step toward a better understanding of split learning and could inspire future research in this area.